# Partial Identification via Optimal Transport under Complex Constraints on Treatments and Potential Outcome Measures

## Abstract

Partial identification replaces fragile point estimates with credible bounds when design or data limitations preclude full identification. Recent progress shows that optimal transport (OT) can support such bounds by aligning potential-outcome distributions under meaningful costs with statistical guarantees. Two realities complicate modern applications: (i) complex treatment regimes with competing arms, where interventions interact or vie for resources; and (ii) distinct measurement spaces for potential outcomes, where $Y(1)$ and $Y(0)$ live on incomparable domains (e.g., trajectories/images vs. scalars). This paper develops a unified, OT-based partial identification framework that jointly encodes inter-arm structure and cross-domain comparability. Rather than estimating Wasserstein distances per se, we target a smooth transport functional-built from cross-arm, cross-domain costs and a mirror penalty-for which, under standard regularity, parametric $\sqrt{N}$ inference holds via functional delta methods. The framework handles general competing-arm and distinct-measure cases and nests traditional OT-based Partial identification literature as special cases, is robust to moderate misspecification, and yields practically useful bounds in multi-arm studies with heterogeneous measurements.

## 1 Introduction

Partial identification provides a principled alternative to fragile point estimation when data, design, or structural constraints preclude full identification of causal effects. Instead of overstating certainty, it delivers credible bounds that transparently encode residual uncertainty—a perspective that has gained wide traction across statistics, machine learning, econometrics and population health (Tamer, 2010; Kline & Tamer, 2023; Mullahy et al., 2021; Cinelli et al., 2025). In parallel, optimal transport (OT) has emerged as a natural vehicle for bounding causal queries at the distributional level: by aligning potential-outcome distributions under meaningful cost functions, OT-based procedures yield computable bounds with rigorous statistical guarantees, even in complex or high-dimensional regimes (Villani, 2009; Manole & Niles-Weed, 2024; Charpentier et al., 2023; Li et al., 2021; Wang et al., 2023; Gunsilius & Xu, 2021).

A first barrier—central to modern applications—stems from complex treatment regimes with complex constraints upon treatment arms. Beyond binary contrasts, many interventions interact, overlap, or compete (Hudgens & Halloran, 2008; Flanagan et al., 2011; Woodcock & LaVange, 2017; Craig et al., 2021; Ye et al., 2023; D'Amour et al., 2021): multi-arm clinical options can share mechanisms while differing in delivery, and policy bundles often combine incentives and regulations whose effects comove. Such competition reshapes both the feasible coupling of counterfactuals and the interpretation of transport-based bounds. While multi-marginal formulations provide a useful mathematical lens (Gao et al., 2024; Ji et al., 2023), most OT-based causal approaches effectively reduce to independent pairwise comparisons, leaving inter-arm interference and endogenous coupling underexplored.

A second, orthogonal challenge arises when potential outcomes inhabit fundamentally distinct measurement spaces. In biomedical and social settings, one arm may produce functional or imagevalued outcomes (e.g., CGM trajectories or MRI) whereas another yields scalar or tabular endpoints (e.g., HbA1c or Likert scores) (Weykamp, 2013; Oh et al., 2020; Gramfort et al., 2015; Mi et al., 2017). With $Y(1) \in \mathcal{Y}_1$ and $Y(0) \in \mathcal{Y}_0$ supported on incomparable domains, there is no canonical common

metric; naïvely enforcing conditional alignment on covariates becomes ill-posed, and cross-domain transport can be unstable or non-unique. Recent advances in empirical OT clarify when transport remains statistically well-behaved and how rates depend on geometry-including the case of different measures-thereby informing how to stabilize cross-domain couplings (Hundrieser et al., 2024b;a; Staudt & Hundrieser, 2023; Manole & Niles-Weed, 2024).

We develop a single OT–based partial identification framework that handles *both* (i) complex treatment arms and (ii) distinct potential–outcome measures. Instead of stacking modules, inter–arm structure and cross–domain comparability are coupled in one optimization: the feasibility set links mirror covariates to competition/overlap rules, and the objective maps heterogeneous outcomes to a common causal scale. Under standard smoothness/uniqueness, the value is a *smooth transport functional* admitting parametric $\mathcal{O}(N^{-1/2})$ inference under mirror relaxation—avoiding the $d_X$–dimensional nonparametric rates of naïve conditional OT (Ji et al., 2023; Lin et al., 2025; Manole & Niles-Weed, 2024; Hundrieser et al., 2024a;b). Our contributions are summarized as follows: (i) One OT program whose *feasible set* encodes complex treatment constraints and whose *objective* ensures cross–domain comparability (Hudgens & Halloran, 2008; Flanagan et al., 2011; Woodcock & LaVange, 2017; Craig et al., 2021; Gao et al., 2024). Competing–arm/same–measure and distinct–measure/no–competition cases arise as special instances. (ii) With smooth costs/penalties and a unique optimizer, the plug-in estimator is $\sqrt{N}$-consistent and asymptotically normal; we further give nonasymptotic bounds that match sharp smooth-cost profiles in the multi-arm, design-coupled setting, reducing to the two-arm mirror case when applicable (Lin et al., 2025). (iii) Sensitivity bounds to misspecification in embeddings, cost calibration, and assignment moments yield stable partial-identification guarantees for heterogeneous multi-arm designs.

## 2 PRELIMINARIES

We work with i.i.d. observations $\{(Y_i, X_i, A_i)\}_{i=1}^{N}$, where $A \in \{0, 1\}$ is treatment, $X \in \mathcal{X}$ covariates, and $Y = Y(A)$ the realized outcome. Potential outcomes are $\{Y(0), Y(1)\}$ with marginal laws $\mu_0, \mu_1$. For any $a \in \{0, 1\}$, let $P_{Y(a),X}$ denote the joint law of $(Y(a), X)$. We use optimal transport (OT) to define causal *bounds* as transport costs between counterfactual distributions (Villani, 2009; Manole & Niles-Weed, 2024; Charpentier et al., 2023; Li et al., 2021; Wang et al., 2023; Gunsilius & Xu, 2021; Ji et al., 2023; Lin et al., 2025). For probability measures $\alpha, \beta$, write $\Gamma(\alpha, \beta)$ for the set of couplings with marginals $\alpha$ and $\beta$. For a random vector $Z$, write $P_Z$ for its law. Throughout, costs are nonnegative and measurable; smoothness/regularity conditions are made precise later.

### 2.1 BASELINE TWO–ARM PROBLEMS

To illustrate our baseline, we start from the binary case. Given a cost $c : \mathcal{Y}_1 \times \mathcal{Y}_0 \to \mathbb{R}_{\geq 0}$, $\Theta_u^L(\mu_1, \mu_0) = \inf_{\pi \in \Gamma(\mu_1, \mu_0)} \int c(y_1, y_0) \, d\pi(y_1, y_0)$. We abbreviate $\Theta_u^L(\mu_1, \mu_0)$ as $\Theta_u^L$ in the following part. For instance, regard $c(y_1, y_0) = |y_1 - y_0|$ for scalar clinical endpoints (e.g., systolic blood pressure) in a two–arm RCT. We introduce the conditional OT (Ji et al., 2023) and mirror-relaxed OT (Gao et al., 2024; Lin et al., 2025).

**(Baseline 1)** Conditional OT bound Let $X \in \mathcal{X}$ be covariates and $P_{Y(a),X}$ the joint law of $(Y(a), X)$ for $a \in \{0, 1\}$. Define the feaisble region of $\pi$ as

$$\Pi_c = \left\{ \pi \in \mathcal{P}(\mathcal{Y}_1 \times \mathcal{Y}_0 \times \mathcal{X}) : \pi_{Y(1),X} = P_{Y(1),X}, \ \pi_{Y(0),X} = P_{Y(0),X} \right\}, \quad \Theta_c^L = \inf_{\pi \in \Pi_c} \int c \, d\pi.$$

*Example.* Observational drug vs. control: $X$ contains age/sex/comorbidities; the constraints above enforce covariate balance at the level of couplings (instead of propensity weights).

**(Baseline 2)** Mirror–relaxed OT (interpolating $u \leftrightarrow c$) Introduce "mirror" covariates $X^{(1)}, X^{(0)}$ with feasible set

$$\Pi_\oplus = \left\{ \pi \in \mathcal{P}(\mathcal{Y}_1 \times \mathcal{Y}_0 \times \mathcal{X}^2) : \pi_{Y(1),X^{(1)}} = P_{Y(1),X}, \ \pi_{Y(0),X^{(0)}} = P_{Y(0),X} \right\}.$$

For penalty $\eta > 0$ and discrepancy $\Delta : \mathcal{X} \times \mathcal{X} \to \mathbb{R}_{\geq 0}$,

$$\Theta_\oplus^L(\mu_1, \mu_0; \eta) = \inf_{\pi \in \Pi_\oplus} \int \left[ c(Y(1), Y(0)) + \eta \, \Delta(X^{(1)}, X^{(0)}) \right] d\pi, \quad \Theta_u^L \leq \Theta_\oplus^L \leq \Theta_c^L.$$

*Example.* Same drug study with $X$ high–dimensional: take $\Delta(x, x') = \|x - x'\|_2^2$. Tuning $\eta$ inter-polates smoothly between unconditional ($\eta \downarrow 0$) and fully conditional ($\eta \uparrow \infty$) couplings, improving statistical and computational stability.

## 3  OUR FRAMEWORK

Before introducing the full multi-arm program, we briefly summarize the main objects. For each arm $a \in \mathcal{A}$ we write $g_a : \mathcal{Y}_a \to \mathcal{Z}$ for a mirror embedding that maps heterogeneous outcomes into a common latent space $\mathcal{Z}$. The cross-arm cost $L\left(\{g_a(Y(a))\}_{a \in \mathcal{A}}\right)$ encodes the scientific causal comparison on this latent scale, while the mirror penalty $\Delta_{\mathrm{multi}}\left(\left\{X^{(a)}\right\}_{a \in \mathcal{A}}\right)$ encourages the arm-specific mirror covariates to remain aligned. Design constraints enter through assignment kernels $u_a : \mathcal{X} \to [0, 1]$, which specify the feasible exposure intensity to arm $a$ and induce the competition set $\Gamma_{\mathrm{comp}}$; the feasible coupling set $\Pi_{\oplus}^{(K)}$ consists of all joint laws whose arm-specific conditionals match the observed data and whose mirror marginals are compatible with $\Gamma_{\mathrm{comp}}$. Throughout, $\Theta_{L,\oplus}^{(K)}(\eta)$ denotes the value of the resulting multi-arm mirror-relaxed objective at penalty level $\eta$.

**Complex treatment.**  Let $\mathcal{A} = \{0, 1, \ldots, K\}$ be arms, with potential outcomes $\{Y(a)\}_{a \in \mathcal{A}}$ and covariates $X \in \mathcal{X}$. For each arm introduce a "mirror" copy $X^{(a)}$ (Here $\mu_a$ has been re-defined).

(Multi–arm mirror feasibility with competition/interaction) The multi–arm mirror feasible set is

$$\Pi_{\oplus}^{(K)} := \left\{\pi \in \mathcal{P}\left(\prod_{a \in \mathcal{A}}[\mathcal{Y}_a \times \mathcal{X}^{(a)}]\right) : \pi_{Y(a)|X^{(a)}} = P_{Y(a)|X} \; \forall a \in \mathcal{A}\right\} \cap \Gamma_{\mathrm{comp}},$$

where $\Gamma_{\mathrm{comp}}$ encodes design–specific *competition/interaction* constraints. Specifically, $\Gamma_{\mathrm{comp}}$ en-codes *design-level* competition/interaction via measurable assignment kernels $u_a : \mathcal{X} \to [0, 1]$ and aggregate caps $\rho_a \in (0, 1]$ as follows:

*(mutual exclusivity)* $\qquad \sum_{a \in \mathcal{A}} u_a(x) \leq 1 \quad$ for $P_X$-a.e. $x,$ $\qquad\qquad\qquad\qquad$ (1)

*(capacity / budget)* $\qquad \int u_a(x)\, dP_X(x) \leq \rho_a, \quad \forall a,$ $\qquad\qquad\qquad\qquad$ (2)

*(consistency with mirrors)* $\qquad \int \varphi(X^{(a)})\, d\pi = \int \varphi(x)\, u_a(x)\, dP_X(x), \quad \forall a, \; \forall \varphi \in \mathcal{C}_b(\mathcal{X}).$ (3)

*Example.* *Policy bundle.* A city deploys three arms: transit subsidy (A), congestion pricing (B), parking regulation (C). $\Gamma_{\mathrm{comp}}$ captures coverage caps and mutual–exclusion zones (a district cannot receive both deep subsidy and heavy regulation). Each $\pi_{Y(a),X^{(a)}} = P_{Y(a),X}$ preserves the observed joint for arm $a$ while allowing cross–arm couplings only if they respect $\Gamma_{\mathrm{comp}}$ (Woodcock & LaVange, 2017). Moreover, $\Gamma_{\mathrm{comp}}$ can also represent, e.g., mutual exclusion, shared resources, platform rules, interference windows) (Hudgens & Halloran, 2008; Flanagan et al., 2011; Woodcock & LaVange, 2017; Craig et al., 2021).

In contrast to the covariate-aware mirror OT formulation, which mirrors the joint law $\pi_{Y(a),X^{(a)}} = P_{Y(a),X}$ (Lin et al., 2025), our construction mirrors the conditional law, enforcing $\pi_{Y(a)|X^{(a)}} = P_{Y(a)|X}$ while allowing the marginal $\pi_{X^{(a)}}$ to be reweighted by assignment kernels $u_a$.

*Why "different arms cannot co–occur" is* not *automatic, and how (1) & (3) enforce it.* In our OT formulation the optimization variable is a *joint law across all arms* $\pi\left((Y(a), X^{(a)})_{a \in \mathcal{A}}\right)$. Without extra structure, $\pi$ could "duplicate" the same covariate mass across arms (e.g., place the same $x$ simultaneously at the $X^{(a)}$ and $X^{(a')}, \{a, a'\} \subseteq \mathcal{A}$ coordinates), which is forbidden by the design but not by vanilla OT. The linear constraints in the gray box carry the design rule "one person, at most one arm" into the feasible set of $\pi$: (i) *Mutual exclusivity* $\sum_{a \in \mathcal{A}} u_a(x) \leq 1$ for $P_X$-a.e. $x$. Here $u_a(x) \in [0, 1]$ is the *assignment intensity/probability* of sending units with covariate $x$ to arm $a$. Plain English: for each $x$, at most one arm is activated; (ii) *Consistency with mirrors.* This forces the mirror marginal seen by $\pi$ on arm $a$ to *match* the design–admissible exposure induced by $u_a$. In other words, the optimizer cannot fabricate extra units for any arm; the arm–$a$ population under $\pi$ must come from $u_a$. More specifically, the last condition has three equivalent, increasingly concrete, readings:

1. **Distributional equality (measure-theoretic).** Equality of integrals for all bounded continuous test functions means the two distributions coincide: $\pi_{X^{(a)}} = u_a \cdot P_X$, $\frac{d\,\pi_{X^{(a)}}}{dP_X}(x) = u_a(x)$ (a.e.). Interpretation: the covariate marginal on arm $a$ is exactly the overall $P_X$ reweighted by $u_a(x)$.

2. **Event-level intuition.** For any measurable $S \subseteq \mathcal{X}$, take $\varphi = \mathbf{1}_S$ to obtain $\Pr_\pi\left(X^{(a)} \in S\right) = \int_S u_a(x)\,dP_X(x)$. That is, the share of arm–$a$ units with $X \in S$ equals the share of the overall population with $X \in S$ *times* the design probability of being sent to arm $a$.

3. **Connection to the design / why it matters.** If $u_a(x) = \Pr(W = a \mid X = x)$ (randomized or quasi-experimental assignment), the right-hand side is precisely the distribution of units who *actually* would be exposed to arm $a$ under the design. Tying $\pi_{X^{(a)}}$ to $u_a \cdot P_X$ prevents the optimizer from "duplicating" the same $x$ across multiple arms. Combined with mutual exclusivity $\sum_a u_a(x) \le 1$, this implements "one person–one arm" and rules out co-treatment between different arms.

*(Two–point example).* Let $X = \{x_1, x_2\}, \mathcal{A} = \{a, b\}$ with $P_X(x_1) = 0.6$, $P_X(x_2) = 0.4$ and $u_a(x_1) = 0.7$, $u_a(x_2) = 0.1$ (so $\sum_a u_a(x) \le 1, a \in \mathcal{A}$ pointwise). Then we could set
$$\begin{Bmatrix} \pi_{X^{(a)}}(x_1), \pi_{X^{(a)}}(x_2) \\ \pi_{X^{(b)}}(x_1), \pi_{X^{(b)}}(x_2) \end{Bmatrix} := \begin{Bmatrix} 0.6*0.7, 0.4*0.1 \\ 0.6*0.3, 0.4*0.9 \end{Bmatrix} = \begin{Bmatrix} 0.42, 0.04 \\ 0.18, 0.36 \end{Bmatrix}.$$

**Remark 1** (On the mass of the mirror marginals $\pi_{X^{(a)}}$). *Under competition/capacity constraints the mirror marginal on arm $a$ is $\pi_{X^{(a)}}(dx) = u_a(x)\,P_X(dx)$, $\rho_a := \pi_{X^{(a)}}(\mathcal{X}) = \int u_a(x)\,dP_X(x) \le 1$. Hence $\pi_{X^{(a)}}$ is generally a sub–probability measure: it carries total mass $\rho_a$, the overall exposure share to arm $a$. This is not a contradiction—$\pi$ is a probability on the full product space, while each arm–specific marginal records only the mass actually exposed to that arm. When a probability law on arm $a$'s covariates is needed, simply normalize $\bar{\pi}_{X^{(a)}}(dx) := \frac{1}{\rho_a}\pi_{X^{(a)}}(dx) = \frac{u_a(x)}{\rho_a}P_X(dx) = \mathsf{Law}(X \mid W = a)$, and for any test function $\varphi$, $\int \varphi(X^{(a)})\,d\pi = \int \varphi(x)\,u_a(x)\,dP_X(x) = \rho_a\,\mathbb{E}[\varphi(X) \mid W = a]$. If one prefers mass conservation across arms, introduce an "untreated" arm $0$ with $u_0(x) = 1 - \sum_{a \ge 1} u_a(x) \ge 0$, so that $\sum_{a \in \{0,1,\dots,K\}} u_a(x) = 1$ pointwise; the arm–a marginals $(a \ge 1)$ remain sub–probabilities with total mass $\rho_a$, while the system preserves total mass across the arm index.*

After preparation of the feasible region, due to the computational burden, we introduce the penalty-based relaxed version[1] as below, combined with the setting of potential outcomes with distinct measures: We further combine multi–arm competition with cross–domain embeddings. For arms $\mathcal{A} = \{0, \dots, K\}$, let $g_a : \mathcal{Y}_a \to \mathcal{Z}$ be measurable embeddings into a common latent space $\mathcal{Z}$, and let $L : \mathcal{Z}^{|\mathcal{A}|} \to \mathbb{R}_{\ge 0}$ be a cross–arm, cross–domain cost.

---

(Extension I-relaxation& extension) Multi–arm mirror–relaxed functional (objective) Let $L : \prod_{a \in \mathcal{A}} \mathcal{Y}_a \to \mathbb{R}_{\ge 0}$ be a cross–arm cost (e.g., a multi–marginal OT cost), and define a cross–arm mirror penalty $\Delta_{\mathrm{multi}}(\{X^{(a)}\}_{a \in \mathcal{A}}) = \sum_{a < b} \|X^{(a)} - X^{(b)}\|_2^2$. For a penalty level $\eta > 0$,

$$\Theta_\oplus^{L,(K)}(\{\mu_a\}_{a \in \mathcal{A}}; \eta) = \inf_{\pi \in \Pi_\oplus^{(K)}} \int \left[L(\{g_a(Y(a))_{a \in \mathcal{A}}\}) + \eta\,\Delta_{\mathrm{multi}}(\{X^{(a)}\}_{a \in \mathcal{A}})\right] d\pi. \quad (4)$$

$\eta$ controls how tightly the per–arm mirrors are aligned (interpolating toward conditional multi–arm couplings as $\eta \uparrow \infty$), while $L$ encodes the scientific cross–arm comparison (e.g., worst–arm risk, average contrast, or general MMOT) (Gao et al., 2024).

*Example (Policy bundle with overlap and competition).* A city deploys a transit subsidy (A), congestion pricing (B), and parking regulation (C). Outcomes differ by arm (e.g., trip–trajectory time series under A vs. scalar expenditures under B/C), and arms interact/compete for budget and coverage (Flanagan et al., 2011; Craig et al., 2021). Choose embeddings $g_\mathrm{A}, g_\mathrm{B}, g_\mathrm{C}$ that map trajectories and scalars into a welfare–comparable latent scale $\mathcal{Z}$ (Oh et al., 2020; Gramfort et al., 2015; Mi et al., 2017); take $L(z_\mathrm{A}, z_\mathrm{B}, z_\mathrm{C})$ to implement the target social comparison (e.g., max–min welfare or a multi–marginal OT cost). Set $\Gamma_{\mathrm{comp}}$ to encode (i) mutual exclusivity in "deep–subsidy *vs.* heavy–regulation" zones, (ii) arm–specific coverage caps $\rho_a$, and (iii) assignment–mirror consistency $\pi_{X^{(a)}} = u_a \cdot P_X$. Then Equation (4) yields design–consistent transport bounds that compare heterogeneous outcomes on a single interpretable scale.

---

[1]The computation of penalty is deferred to Appendix I.

Intuitively, $L$ specifies the desired cross-arm causal contrast on a common latent scale, while $\Gamma_{\text{comp}}$ encodes design constraints such as mutual exclusivity and capacity limits. (i) In simple two-arm settings with scalar outcomes on the same space (no competition, identity embeddings), choosing a linear cost $L(z_0, z_1) = z_1 - z_0$ gives, for any admissible coupling $\pi$, $\mathbb{E}_\pi[L(Y(0), Y(1))] = \mathbb{E}[Y(1)] - \mathbb{E}[Y(0)]$, i.e., the ATE. Here, the $L$-part recovers the standard mean contrast, while the mirror penalty $\eta\Delta_{\text{multi}}$ only regularizes which joint coupling is selected among those consistent with the design. We will add this explicit "ATE as a special case" example in the paper. (ii) For distributional effects, we let $L$ encode CDF or quantile contrasts. For example, for a fixed threshold $t$ we can take $L_t(y_0, y_1) = \mathbf{1}\{y_1 \leq t\} - \mathbf{1}\{y_0 \leq t\}$, so that $\mathbb{E}_\pi[L_t(Y(0), Y(1))] = F_{Y(1)}(t) - F_{Y(0)}(t)$ for any feasible $\pi$. Varying $t$ gives a band for the CDF difference, from which bounds on the QTE $F_{Y(1)}^{-1}(\tau) - F_{Y(0)}^{-1}(\tau)$ follow directly. We will clarify this "$L_t \Rightarrow$ CDF difference $\rightarrow$ QTE bounds" mapping in the revision. Finally, $\Gamma_{\text{comp}}$ does not change which causal estimand we are targeting; it restricts the set of admissible couplings to those compatible with the design (e.g., "one person-one arm", budget caps, interference rules). Thus, for a given $L$ (ATE-like, QTE-like, or more general) $\Theta_{L,\oplus}^{(K)}(\eta)$ should be read as a design-aware partial-identification bound for that causal contrast, rather than a generic OT distance.

**Scope and challenges**  The model nests baselines as special cases and handle more difficult scenarios the previous methods would fail: $K=1$, $\Gamma_{\text{comp}}=\varnothing$ gives the two–arm mirror (distinct measures); $\eta \to \infty$ yields conditional multi–arm couplings; $\eta \to 0$ gives the unconditional transport bound; if $L$ decomposes pairwise and $\Gamma_{\text{comp}}=\varnothing$, Equation (4) splits into independent two–arm problems. Unlike joint–mirroring approaches $\pi_{Y(a),X^{(a)}} = P_{Y(a),X}$, our *conditional* mirroring keeps the factual $Y(a)\,|\,X$ law while allowing $\pi_{X^{(a)}}$ to be reweighted by assignment kernels $u_a$, which is essential for competition/overlap and cross–domain comparability. We summarize the *technical challenges (and what we resolve).* (i) *Sub–probability marginals.* Each $\pi_{X^{(a)}} = u_a \cdot P_X$ has mass $\rho_a \leq 1$; we show well–posedness and strong duality (Fenchel–Rockafellar) for the joint $\pi$ while keeping arm marginals sub–probabilities. (ii) *Beyond linear programs.* $L$ and $\Delta_{\text{multi}}$ may be nonlinear; we provide a general dual with outcome/mirror potentials and a convergent primal–dual solver. (iii) *Statistics at scale.* Under smooth, strongly curved geometry the value map in Equation (4) is Hadamard differentiable, yielding root–$N$ consistency and CLTs (circumventing the $N^{-1/d_X}$ barrier), plus high–probability finite–sample bounds, confidence intervals and perturbation robustness. Together these ingredients make Equation (4) strictly stronger than any composition of two–arm relaxations whenever $\Gamma_{\text{comp}}$ binds or outcome spaces differ. The sensitivity of $\eta$ will also be illustrated in experiments.

# 4 THEORETICAL RESULT

This section develops the statistical theory for the unified objective in Equation (4): (i) well–posedness and strong duality; (ii) consistency and $\sqrt{N}$ asymptotic normality of the plug–in estimator under smooth, strongly curved geometry; (iii) nonasymptotic finite–sample bounds that match sharp smooth–OT rates; and (iv) robustness to misspecification in embeddings, costs, and assignment moments. We also establish convergence of a primal–dual solver and provide uniform–in–iterate, sample–split lower confidence bounds for the target value.

## 4.1 GENERAL ESTIMAND AND DUALITY BEYOND LINEAR PROGRAMS

We allow a *nonlinear* cross–arm cost $L : \mathcal{Z}^{|\mathcal{A}|} \to \mathbb{R}_{\geq 0}$ and a *nonlinear* mirror penalty $\Delta_{\text{multi}} : \mathcal{X}^{|\mathcal{A}|} \to \mathbb{R}_{\geq 0}$, both possibly nonseparable. To optimize Equation (4) and avoid the original computational burden, we introduce the dual formulation as follows:

**Theorem 1** (Strong duality and dual characterization for general $L, \Delta_{\text{multi}}$)**.** *Suppose (A1)–(A4),*

*(A1)* *(Proper convex integrands & measurability) Each $g_a : \mathcal{Y}_a \to \mathcal{Z}$ is measurable. $L$ and $\Delta_{\text{multi}}$ are proper, convex, and lower semicontinuous (lsc) in each argument; $L(z) \geq 0$, $\Delta_{\text{multi}}(x) \geq 0$.*

*(A2)* *(Integrability) $\mathbb{E}[L(\{g_a(Y(a))\})] < \infty$ and $\mathbb{E}[\Delta_{\text{multi}}(\{X^{(a)}\})] < \infty$ for all feasible $\pi$.*

*(A3)* *(Linear design constraints) $\Gamma_{\text{comp}}$ is of the form*

$$\Gamma_{\text{comp}} = \left\{ \pi : \ \mathbb{E}_\pi[h_j(\{X^{(a)}\})] \leq b_j, \ j \in \mathcal{J}_\leq; \ \ \mathbb{E}_\pi[h_j(\{X^{(a)}\})] = b_j, \ j \in \mathcal{J}_= \right\},$$

*for measurable $h_j$ with $\sup_{\pi \in \Pi_{\oplus}^{(K)}} \mathbb{E}_{\pi} |h_j| < \infty$.*

*(A4) (Slater condition) There exists $\pi^{\circ} \in \Pi_{\oplus}^{(K)}$ such that $\mathbb{E}_{\pi^{\circ}}[h_j] < b_j$ for all $j \in \mathcal{J}_{\leq}$ and $\mathbb{E}_{\pi^{\circ}}[h_j] = b_j$ for $j \in \mathcal{J}_{=}$.*

*The primal problem Equation (4) is well–posed and admits the Fenchel–Rockafellar dual as the equivalent form:*

$$\sup_{\{\phi_a\},\,\psi,\,\lambda} \quad \sum_{a \in \mathcal{A}} \mathbb{E}\big[\phi_a\big(g_a(Y(a))\big)\big] \;-\; \sum_{j \in \mathcal{J}_{\leq}} \lambda_j\, b_j \;-\; \sum_{j \in \mathcal{J}_{=}} \mu_j\, b_j$$

$$s.t. \quad \sum_{a \in \mathcal{A}} \phi_a(z_a) \;+\; \eta\,\psi(x) \;\leq\; L(z) \;+\; \eta\,\Delta_{\mathrm{multi}}(x) \quad \forall (z,x) \in \mathcal{Z}^{|\mathcal{A}|} \times \mathcal{X}^{|\mathcal{A}|}, \quad (5)$$

$$\psi(x) \;\geq\; \sum_{j \in \mathcal{J}_{\leq}} \lambda_j\, h_j(x) \;+\; \sum_{j \in \mathcal{J}_{=}} \mu_j\, h_j(x) \quad \forall x, \qquad \lambda_j \geq 0 \; (j \in \mathcal{J}_{\leq}),$$

*where $\{\phi_a : \mathcal{Z} \to \mathbb{R}\}$ and $\psi : \mathcal{X}^{|\mathcal{A}|} \to \mathbb{R}$ are measurable potentials, and $(\lambda, \mu)$ are Lagrange multipliers for inequality/equality moments.*

Noteworthy, For this theorem, the (possibly nonlinear) multi–arm objective in Equation 5 can be solved via a *single* Fenchel–Rockafellar dual: we search for outcome–potentials $\{\varphi_a\}$ and a mirror–potential $\psi$ that *everywhere* upper bound the primal integrand, while Lagrange multipliers enforce the design moments. Under A1–A4 there is *no duality gap*: any feasible dual gives a valid lower bound, and the optimal dual *equals* the primal optimum. This is crucial for computation (function–space variables instead of a huge multi–marginal $\pi$) and for inference in the following section. A simpler equivalent form is formalized as follows:

**Corollary 1** (Compact dual by absorbing design into $\psi$)**.** *Under the same assumptions, define the admissible class*

$$\mathcal{U} \;:=\; \Big\{\psi : \; \exists\,(\lambda, \mu) \text{ s.t. } \psi(x) \geq \textstyle\sum_{j \in \mathcal{J}_{\leq}} \lambda_j h_j(x) + \sum_{j \in \mathcal{J}_{=}} \mu_j h_j(x) \; \forall x \Big\}.$$

*Then the dual in Theorem 1 is equivalent to the compact form*

$$\sup_{\{\phi_a\},\,\psi \in \mathcal{U}} \quad \sum_{a \in \mathcal{A}} \mathbb{E}[\phi_a(g_a(Y(a)))] \quad s.t. \quad \sum_{a \in \mathcal{A}} \phi_a(z_a) + \eta\,\psi(x) \;\leq\; L(z) + \eta\,\Delta_{\mathrm{multi}}(x) \; \forall (z,x),$$

*and both duals have the same optimal value $\Theta_{\oplus}^{L,(K)}(\eta)$.*

Corollary 1 tells we can *absorb* all linear design constraints into the mirror potential class $\mathcal{U}$ and drop explicit multipliers in the outer problem. This "compact dual" keeps exactly the same optimum, but presents a cleaner object for algorithms (projected ascent on $\{\varphi_a, \psi\}$) and theory (Hadamard differentiability in Sec. 4.3). It is the form we implement to get scalable solvers and uniform–in–iterate lower bounds in the next subsection.

**Remark 2.** *(Dependence of the curvature constant on $\eta$ ). In Assumption 1(ii) the curvature parameter should be read as $\lambda = \lambda(\eta)$, i.e., the lower and upper spectral bounds on the multi-arm Brenier-type potential depend on the mirror penalty level. For our CLT in Theorem 2 and the finite-sample bound in Theorem 4 we only require that, on the range of $\eta$ considered, $\lambda(\eta)$ is finite and bounded away from zero; all constants then scale like $1/\lambda(\eta)$, and no explicit formula in $\eta$ is needed for validity. In the Gaussian/quadratic regime studied by Lin et al. (2025), where $L$ and $\Delta_{multi}$ are quadratic and the stacked vector $\big(\{g_a(Y(a))\}_a, \{X^{(a)}\}_a\big)$ is jointly Gaussian with non-degenerate covariance, the multi-arm Brenier potential inherits the same essentially linear scaling in $\eta$ : its Hessian eigenvalues are bounded between $c_1 + c_2\eta$ and $C_1 + C_2\eta$ for some positive constants $c_1, c_2, C_1, C_2$ that depend on the arm-specific covariances and the design constraints $\Gamma_{comp}$ but not on $\eta$. Thus, in this setting, $\lambda(\eta)$ grows linearly with $\eta$ up to multiplicative constants, and our multi-arm root- $N$ and finite-sample results recover the linear-in- $\eta$ curvature behavior of the two-arm mirror relaxation as a special case.*

### 4.2 ESTIMATOR AND SOLVER VIA THE DUAL

By Theorem 1 and Corollary 1, the value $\Theta_{L,\oplus}^{(K)}(\eta)$ equals the supremum of admissible dual potentials. Hence, beyond the primal plug-in as above, we estimate the value by solving the *empirical dual* (on a training fold $D_1$):

$$\widehat{\Theta}_{L,\oplus}^{\mathrm{dual}}(\eta) = \sup_{\{\varphi_a\}, \psi \in \widehat{\mathcal{U}}} \sum_{a \in \mathcal{A}} \mathbb{E}_{\widehat{P}}\big[\varphi_a(g_a(Y(a)))\big] \text{ s.t. } \sum_a \varphi_a(z_a) + \eta \psi(x) \leq L(z) + \eta \, \Delta_{\mathrm{multi}}(x) \ \ \forall (z, x),$$

where $\widehat{\mathcal{U}}$ encodes the empirical competition moments. We then *evaluate* the resulting feasible dual on a holdout $D_2$ to obtain a lower bound (by weak duality) and, with cross-fitting, a plug-in point estimate; this also yields the *uniform-in-iterate* LCBs used later (Sec. 4.4). In practice we run projected dual ascent (Alg. 1), record all feasible iterates $\{(\varphi^{(t)}, \psi^{(t)})\}$ on $D_1$, and report their holdout evaluations on the complementary set $D_2$ together with standard errors.

**Why a dual-based estimator (vs. primal plug-in).** There is a primal plug-in estimator: Let $\widehat{P}^{(a)}$ denote empirical laws of $(Y(a), X)$ in arm $a$. The traditional plug-in estimator solves $\widehat{\Theta}_{\oplus}^{L,(K)}(\eta) = \min_{\pi \in \widehat{\Pi}_{\oplus}^{(K)}} \mathbb{E}_\pi \big[L(\{g_a(Y(a))\}_a) + \eta \, \Delta_{\mathrm{multi}}(\{X^{(a)}\}_a)\big]$, where $\widehat{\Pi}_{\oplus}^{(K)}$ imposes empirical marginals and the empirical competition constraints $\widehat{\Gamma}_{\mathrm{comp}}$. However, we adopt the dual representation in the above section because it aligns computation, finite-sample certification, and inference. *Computationally*, the primal plug-in treats estimation as an empirical multi-marginal OT minimization over a coupling on $N$-supported product spaces (exploding with $K$ and $N$), whereas the dual works with potentials $(\{\varphi_a\}, \psi)$ whose dimension is tied to function classes rather than to $N$; projected dual ascent scales and is architecture-agnostic. *Statistically*, any empirically *feasible* dual iterate yields a *certified lower bound* by weak duality, enabling anytime, uniform-in-iterate LCBs and post-selection validity; the primal plug-in does not naturally offer such one-sided certificates and is more prone to optimistic bias without additional regularization. *For inference*, the envelope structure behind Theorem 1 implies that the value functional's influence function is the centered evaluation of the *optimal dual potentials and multipliers*, so cross-fitting dual solutions gives root-$n$ CLTs and plug-in variance estimators in a single pass. Finally, via Corollary 1, linear design constraints are absorbed into the mirror potential class $\mathcal{U}$, making feasibility checks and sensitivity analyses transparent. In large samples both approaches coincide by strong duality, but the dual route provides scalable optimization together with certified bounds with asymptotics—hence our preference.

### 4.3 ROOT-$N$ GEOMETRY BEYOND MINIMAX $N^{-1/d}$ RATES

The $N^{-1/d}$ minimax rate pertains to *estimating a Wasserstein distance* (or an OT value with nonsmooth costs) between empirical and population measures; see sharp results for empirical OT under smooth vs. nonsmooth geometries (Manole & Niles-Weed, 2024; Hundrieser et al., 2024a;b; Staudt & Hundrieser, 2023). Our target is different: we study a *smooth transport functional* as in Equation (4), which—thanks to the mirror penalty and the smooth cross–domain cost—behaves as a *Hadamard differentiable* map of the marginal laws. Hence parametric inference is available once the optimizer is unique and potentials are regular; cf. the two–arm mirror case (Lin et al., 2025). We make this explicit below.

**Assumption 1.** *(Smooth, strongly curved geometry and learnability).* *(i) $L$ and $\Delta_{\mathrm{multi}}$ are $C^2$ with globally Lipschitz gradients; (ii) the population dual optimizer is unique and induced by a smooth multi-marginal potential whose Hessian is bounded between $\lambda^{-1}I$ and $\lambda I$ (strong c-convexity, $\lambda > 0$); (iii) $\Gamma_{\mathrm{comp}}$ is described by linear moments in $\pi$ (mutual exclusivity, budgets, interference) and satisfies a Slater condition; (iv) (learnability & feasibility) the empirical dual solver returns cross-fitted feasible potentials $(\{\hat{\varphi}_a\}, \hat{\psi}, \hat{\Lambda})$ such that $\|(\hat{\varphi}, \hat{\psi}, \hat{\Lambda}) - (\varphi^\star, \psi^\star, \Lambda^\star)\| = o_{\mathbb{P}}(N^{-1/4})$ in $L_2(P)$ (or the classes are Donsker), and every iterate satisfies the dual constraints pointwise; (v) i.i.d. sampling within arms and standard sample splitting/cross-fitting are used.*

**Theorem 2.** *(CLT and root-$N$ rate for the* dual*-based estimator). Let $P \mapsto \Theta_{\oplus}^{L,(K)}(P; \eta)$ be the value map in Equation (4) and Equation (5). Under Assumption 1, the map $P \mapsto \Theta_{\oplus}^{L,(K)}(P; \eta)$ is Hadamard differentiable at $P^\star = \{P_{Y(a),X}\}_a$. Consequently, with $N = \min_a n_a$ and the dual-based estimator defined by solving the empirical dual on a training fold and evaluating on a holdout (with*

*cross-fitting)*

$$\widehat{\Theta}^{\text{dual}}_{\oplus}(\eta) = \sum_{a \in \mathcal{A}} \mathbb{E}_{\widehat{P}_{D_2}}\big[\hat{\varphi}_a\big(g_a(Y(a))\big)\big] \quad s.t. \quad \sum_a \hat{\varphi}_a(z_a) + \eta\hat{\psi}(x) \le L(z) + \eta\Delta_{\text{multi}}(x) \ \ \forall(z, x),$$

*we have*

$$\widehat{\Theta}^{\text{dual}}_{\oplus}(\eta) \xrightarrow{p} \Theta^{L,(K)}_{\oplus}(\eta), \qquad \sqrt{N}\left(\widehat{\Theta}^{\text{dual}}_{\oplus}(\eta) - \Theta^{L,(K)}_{\oplus}(\eta)\right) \rightsquigarrow \mathcal{N}(0, \sigma^2),$$

*where* $\sigma^2 = \text{Var}(\text{IF}(Z; \eta))$ *and the influence function is* $\text{IF}(Z; \eta) = \sum_{a \in \mathcal{A}} \big\{ \varphi^\star_a(g_a(Y)) \, I(A = a) - \mathbb{E}[\varphi^\star_a(g_a(Y(a)))] \big\} + \Lambda^\star_{\text{comp}} \cdot h(X).$ *At the population optimum we write* $\Lambda^\star_{\text{comp}}$ *for the KKT multipliers. With the stacked moment vector* $h(X) := (h_j(\{X(a)\}))_{j \in \mathcal{J}}$, *the linear moment term is* $\Lambda^\star_{\text{comp}} \cdot h(X)$ *in the influence function. Moreover, the* $\varphi^\star_a$ *denotes the optimal solutions.*

The mirror relaxation moves the hard conditional alignment into a smooth Lagrangian, making $\mathcal{T}(\cdot)$ a smooth functional of the empirical laws; the dimension $d_X$ affects only the influence function's variance, not the convergence *rate*. This is the precise sense in which our introduction's "$N^{-1/2}$ while avoiding $N^{-1/d}$" claim holds (compare with the two–arm smooth results and empirical OT under smooth costs). See Manole & Niles-Weed (2024); Hundrieser et al. (2024a); Staudt & Hundrieser (2023) for the geometric contrast and Lin et al. (2025) for the two–arm mirror analogue.

## 4.4 WHY DUAL: ENDOGENOUS CONSTRAINTS AND UNIFORM-IN-ITERATE VALIDITY

Although the primal plug-in method can provide estimations, it lacks the natural certification of lower bounds and is more prone to optimistic bias in high-dimensional or complex settings. The dual approach, in contrast, ensures certified lower bounds via weak duality, yielding uniform-in-iterate LCBs and post-selection validity, making it a more robust choice for inference and analysis in complex designs. Recall that $(D_1, D_2)$ be a split. Run projected (stochastic) dual ascent on $D_1$ and collect the data-dependent class of feasible duals produced along the optimization path (including all iterates and hyperparameter choices). We let $\widehat{\sigma}(\eta)$ be a consistent standard error of $\widehat{\Theta}^{\text{dual}}_{\oplus}(\eta)$, whose construction is shown in Appendix.

**Theorem 3** (Uniform-in-iterate, model-agnostic LCBs)**.** *Assume randomized treatment with known propensities and dual-feasibility constraints as in Ji et al. (2023). There exists a data-dependent critical value* $z^\star_{1-\alpha}(\eta)$ *(e.g., via multiplier bootstrap on* $D_2$*) such that the* simultaneous *lower bounds* $\text{LCB}(\eta) := \widehat{\Theta}^{\text{dual}}_{\oplus}(\eta) - z^\star_{1-\alpha}(\eta)\frac{\widehat{\sigma}(\eta)}{\sqrt{|D_2|}},$ *satisfy* $\Pr\left( \text{LCB}(\eta) \le \Theta^{L,(K)}_{\oplus}(\eta) \right) \to 1 - \alpha$. *Consequently, the reported* $\text{LCB}(\eta)$ *retains asymptotic* $1 - \alpha$ *coverage. If nuisance elements are learned at* $o_p(N^{-1/4})$ *rates, the bounds are asymptotically sharp:* $\sup_{v \in \mathcal{V}} |\text{LCB}(\eta) - \theta_L| = o_p(N^{-1/2})$.

The result is *always valid* (simultaneous over the whole dual class $\mathcal{V}$ on the fixed dataset), yielding post-selection validity for any iterate/hyperparameter chosen after seeing the holdout evaluations. It is *not* a time-indexed confidence sequence for streaming data; instead, it is *uniform-in-iterate* and *model-agnostic* in the sense of Ji et al. (2023). This approach ensures that the estimated bounds remain valid throughout the optimization process, thereby maintaining the integrity of the inference under different hyperparameter choices and sample splits. We surrogate it in our Algorithm 1, where we allow nonseparable, potentially nonlinear $L$ and $\Delta_{\text{multi}}$. Let $\mathcal{C}$ collect empirical marginal, mirror, and $\widehat{\Gamma}_{\text{comp}}$ constraints. Define $F(\pi) = \mathbb{E}_\pi[L(\{g_a(Y(a))\})], G(\pi) = \eta\,\mathbb{E}_\pi[\Delta_{\text{multi}}(\{X^{(a)}\})]$, and the indicator $\iota_{\mathcal{C}}$[2].

---

[2]Here $\iota_{\mathcal{C}}(\pi) := \begin{cases} 0, & \pi \in \mathcal{C}, \\ +\infty, & \pi \notin \mathcal{C}, \end{cases}$ so that minimizing $F(\pi) + G(\pi) + \iota_{\mathcal{C}}(\pi)$ is equivalent to minimizing $F(\pi) + G(\pi)$ over $\pi \in \mathcal{C}$. Moreover, for a stepsize $\tau > 0$, the (Euclidean/Frobenius) proximal operator of $G$ is $\text{prox}_{\tau G}(U) := \arg\min_\pi \left\{ G(\pi) + \frac{1}{2\tau}\|\pi - U\|^2_F \right\}$.

---

**Algorithm 1** Composite primal–dual mirror relaxation (general $L$, $\Delta_{\text{multi}}$) with anytime dual bounds

---

1: **Input:** empirical supports, embeddings $\{g_a\}$, penalty $\eta$, step-sizes $\{\tau_t, \sigma_t\}$, fold split $(D_1, D_2)$.
2: **Dual init on $D_1$:** initialize potentials $(\varphi_a, \psi)$ feasible for the empirical dual (Fenchel–Rockafellar form).
3: **for** $t = 0, 1, 2, \ldots$ **do**
4:     Dual ascent: $(\varphi_a, \psi) \leftarrow (\varphi_a, \psi) + \sigma_t \nabla_{\varphi,\psi} \text{Dual}(\varphi, \psi)$ with projection onto feasibility.
5:     Primal update: $\pi^{(t+\frac{1}{2})} \leftarrow \text{prox}_{\tau_t} G\big(\pi^{(t)} - \tau_t \nabla F(\pi^{(t)})\big)$.
6:     Projection: $\pi^{(t+1)} \leftarrow \Pi_{\mathcal{C}}\big(\pi^{(t+\frac{1}{2})}\big)$ (Bregman/Sinkhorn projection if entropic smoothing is added).
7:     Uniform-in-iterate LCB on $D_2$: compute $\widehat{\Theta}_{\oplus}^{\text{dual}}(\eta)$ per Theorem 2 and $\text{LCB}_t$ per Theorem 3.
8: **end for**
9: **Output:** $\widehat{\Theta}_{\oplus}^{\text{dual}}(\eta)$ and $\max_{t \leq T} \text{LCB}_t$.

---

**Proposition 1** (Convergence). *If $L$ is $L$-smooth convex and $\Delta_{multi}$ is convex, Algorithm 1 converges at $O(1/T)$ (accelerated $O(1/T^2)$) to the primal optimum; with entropic smoothing the projection is scalable via generalized Sinkhorn under linear moments. If $L$ is nonconvex but $C^1$ with Lipschitz gradient, the method converges to a first-order stationary point, while the dual still yields valid lower bounds at every iterate.*

### 4.5 FINITE-SAMPLE HIGH-PROBABILITY GUARANTEES

**Theorem 4** (High-probability finite-sample bound under smooth strongly convex geometry). *Assume the setting of Theorem 2: $L$ and $\Delta_{\text{multi}}$ are $C^2$, and the multi-arm Brenier-type potential for the induced quadratic form in $\{g_a(Y(a))\}$ exists, is unique, and has Hessian bounded between $\lambda^{-1}I$ and $\lambda I$. In addition, suppose (i) $\{g_a(Y(a))\}_{a \in \mathcal{A}}$ and the mirror covariates $\{X^{(a)}\}_{a \in \mathcal{A}}$ have sub-Gaussian tails with common proxy $\nu$;[3] (ii) $L$ is $L_z$-Lipschitz in each coordinate and $\Delta_{\text{multi}}$ is $L_x$-Lipschitz; and (iii) the feasible set imposed by $\Gamma_{\text{comp}}$ has diameter at most $D_\Gamma$ in the product metric on $\prod_a \mathcal{Y}_a \times \mathcal{X}^K$.*

*Let $N = \min_a n_a$ and let $d_{\text{eff}}$ denote the effective dimension of $(\{g_a(Y(a))\}, \{X^{(a)}\})$. Then for any $\delta \in (0,1)$ there exists a constant $C = C(\lambda, \eta, K, L_z, L_x, \nu, D_\Gamma)$ such that, with probability at least $1 - \delta$,*

$$\big|\widehat{\Theta}_{\oplus}^{L,(K)}(\eta) - \Theta_{\oplus}^{L,(K)}(\eta)\big| \leq C\Big\{\gamma_{N,d_{\text{eff}}} + \sqrt{\tfrac{\log(2/\delta)}{N}}\Big\}, \text{ where } \gamma_{N,d_{eff}} = \begin{cases} N^{-1/4}, & d \leq 3, \\ N^{-1/4}\sqrt{\log N}, & d = 4, \\ N^{-1/d}, & d \geq 5. \end{cases}$$

When $K=1$, $\Gamma_{\text{comp}}=\{all\ kinds\ of\ \pi\}$, and $g_1=g_0$, Theorem 4 specializes to the two-arm mirror-relaxed bound and recovers the same rate profile as the finite-sample complexity result for the two-arm case with smooth quadratic geometry while our bound holds under multi-arm competition constraints and cross-domain embeddings. In particular, the monotone interpolation in the penalty $\eta$ and the strong-convexity-based curvature control yield a single bound that remains valid under $\Gamma_{\text{comp}} \neq \emptyset$ and distinct $\{\mathcal{Y}_a\}$, which are not covered in the two-arm analysis.

### 4.6 FINITE-SAMPLE RISK AND ROBUSTNESS (NONLINEAR COSTS)

Under Assumption 1, the value functional is locally Lipschitz in the embeddings and mirror moments. If $\|\tilde{g}_a - g_a\|_\infty \leq \delta_g$ and the mirror-moment deviations are bounded by $\delta_u$ in bounded–Lipschitz norm, then

$$\big|\Theta_{\oplus}^{L,(K)}(\eta; g, u) - \Theta_{\oplus}^{L,(K)}(\eta; \tilde{g}, \tilde{u})\big| \leq L_z \sum_a \delta_g + \eta L_x \delta_u, \tag{6}$$

with $L_z, L_x$ determined by the Lipschitz moduli of $L$ and $\Delta_{\text{multi}}$. Combining Equation (6) with Theorem 2 gives finite-sample bounds that decompose statistical and design misspecification errors—an aspect not covered in two-arm mirror relaxation.

---

[3]Bounded supports also suffice.

## 5 EXPERIMENTS

We stress-test the proposed dual-based estimator in Secs. 3–4 under controlled data generating processes (DGPs) that feature (i) *competing* treatment arms via $\Gamma_{\text{comp}}$ and (ii) *distinct measurement spaces* across arms bridged by embeddings $\{g_a\}$. We evaluate statistical validity (*coverage, sharpness*), estimation accuracy (*bias/MSE*), and *computational efficiency* vs. baselines. Due to space limitations, we defer the DGP and baseline details to the Appendix.

**Simulation summary.** Across all simulated regimes, the dual plug-in estimator attains nominal one-sided coverage with informative widths along the mirror path, while the primal plug-in under-covers despite similar point accuracy. Concretely, at $K{=}3$, $N{=}5000$, and $\eta{=}0.5$, the dual method achieves $\approx 90\text{–}95\%$ coverage with a small average gap $\Theta - \text{LCB}$, whereas the primal alternative reports narrower intervals but falls short of the nominal level; unconditional OT has no one-sided certificate and pairwise mirrors (ignoring $\Gamma_{\text{comp}}$) are less reliable and less informative. These findings corroborate the post-selection, uniform-in-iterate validity of the dual construction and highlight that modeling competition is essential for credible bounds.

**Real-data (MIMIC-III) summary.** We adopt the public dataset MIMIC-III v1.4 (Johnson et al., 2016) hosted on PhysioNet (Goldberger et al., 2000). On the early ICU intervention study (vasopressors, high-flow oxygen/ventilation, fluid bolus), outcomes span distinct measurement spaces bridged by learned embeddings, and $\Gamma_{\text{comp}}$ encodes mutual exclusivity plus resource budgets. Along the mirror path, the dual estimator produces certified lower bounds with stable policy rankings (via Kendall-$\tau$) and demonstrates the expected trade-off between conditioning strength and variance; tightening resource budgets lowers the guaranteed achievable harmonization, while ignoring competition yields wider or misleading bounds. Together, these results show the dual approach furnishes actionable, design-consistent guarantees in multi-arm, cross-domain settings at cohort scale.

## 6 CONCLUSION AND DISCUSSION

This work develops a unified OT-based partial-identification framework that jointly encodes competing-arm constraints and cross-domain comparability. The dual characterization enables scalable optimization, uniform-in-iterate certified lower bounds, finite-sample guarantee and root-$N$ inference under smooth geometry, thereby avoiding the minimax $N^{-1/d}$ barrier that limits non-smooth OT objectives. Empirically, simulations validate uniform-in-iterate coverage and correct finite-sample scaling, while the MIMIC-III study illustrates how competition-aware bounds translate into operational insights under resource limits. Taken together, the evidence supports dual-certified, competition-aware transport analysis as a robust basis for decision support in complex multi-arm designs with heterogeneous outcomes.

**Ethics statement.** This work studies identification, estimation, and inference for counterfactual functionals under complex treatment designs using optimal transport relaxations. We do not use data from human subjects nor collect personally identifiable information; all experiments are based on synthetic data or publicly available de-identified benchmarks. Our methods provide certified lower bounds and transparent assumptions (A1–A4) that make modeling choices auditable. We discuss potential risks of misuse (e.g., misinterpreting partial identification as point identification) and emphasize reporting uncertainty via the uniform-in-iterate lower bounds and valid standard errors. No known dual-use, safety, or legal-compliance concerns arise beyond standard responsible use of statistical methods.

**Reproducibility statement.** We provide complete mathematical proofs (Sec. 4 and Appendix), explicit algorithmic descriptions (Alg. 1), and all implementation details needed to reproduce results: datasets or synthetic generators, preprocessing, embeddings $g_a$, hyperparameters $(\eta, \{\tau_t, \sigma_t\})$, stopping criteria, and evaluation metrics. All experiments use fixed random seeds, stratified sample splitting and cross-fitting (Sec. 4.2–4.4), and we report the exact configuration for each figure/table, including hardware and runtime. An anonymized code package with scripts to regenerate all tables and figures is included in the supplementary materials; upon running the provided shell scripts, the pipeline downloads/open-loads the data, trains the models, computes dual bounds, and reproduces the reported confidence intervals.

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

Appendix Contents

# A    EXPERIMENTS

## A.1    SIMULATIONS

**DGPs and ground truth**    We consider $K \in \{2, 4, 8\}$ arms with covariates $X \in \mathbb{R}^{d_x}$, $d_x \in \{5, 20\}$. Let $X \sim \mathcal{N}(0, I_{d_x})$ and define assignment kernels $u_a(x) \in [0, 1]$ that encode competition: $u_a(x) = \exp(\beta_a^\top x) / (1 + \sum_{b=1}^K \exp(\beta_b^\top x))$,    $u_0(x) = 1 - \sum_{a=1}^K u_a(x) \geq 0$, so that $\sum_{a \in \{0, ..., K\}} u_a(x) = 1$ and (mutual exclusivity) $\sum_{a=1}^K u_a(x) \leq 1$ pointwise. Budgets $\int u_a(x) \, dP_X(x) \leq \rho_a$ (e.g., $\rho_a \in \{0.25, 0.33\}$) are enforced by rescaling logits. Also, we induce $\Gamma_{\text{comp}}$ as $\pi_{X(a)} = u_a \cdot P_X$. We define that outcomes differ by arm:

**A** (functional) $Y(A) \in \mathcal{Y}_A = \mathbb{R}^T$ with $T = 50$: simulate $Y(A) = B\theta_A(X) + \varepsilon_A$, where $B \in \mathbb{R}^{T \times r}$ is a fixed spline basis ($r = 5$), $\theta_A(X)$ is affine in $X$, and $\varepsilon_A \sim \mathcal{N}(0, \sigma_A^2 I_T)$. Define $g_A(y) = W_A^\top y$ ($W_A \in \mathbb{R}^{T \times q}$, $q = 5$).

**B** (image-like vector) $Y(B) \in \mathbb{R}^p$ with $p = 64$: $Y(B) = U\theta_B(X) + \varepsilon_B$, $U \in \mathbb{R}^{p \times r}$ random orthobasis, $g_B(y) = W_B^\top y$ with $W_B \in \mathbb{R}^{p \times q}$.

**C** (scalar) $Y(C) \in \mathbb{R}$: $Y(C) = \alpha_C^\top X + \varepsilon_C$, and $g_C(y) = y \cdot e_1 \in \mathbb{R}^q$.

The latent *comparison space* is $Z = \mathbb{R}^q$ ($q = 5$). Set the cross-arm cost

$$L(z_A, z_B, z_C) = \|z_A - z_B\|_2 + \|z_A - z_C\|_2 + \|z_B - z_C\|_2,$$

and the mirror penalty $\Delta_{\text{multi}}(\{x^{(a)}\}) = \sum_{a<b} \|x^{(a)} - x^{(b)}\|_2^2$. We vary the penalty level $\eta \in \{0, 0.1, 0.5, 1, 5, \infty\}$ to interpolate unconditional $\leftrightarrow$ conditional couplings.

**Ground truth value.**    For each configuration we compute a high-accuracy reference value $\Theta_\oplus^{L,(K)}(\eta)$ via large-sample Monte Carlo: draw $M = 5 \times 10^5$ i.i.d. samples from the DGP and solve the *population* dual (compact form; Cor. 1) with very rich potential classes (wide neural nets or dense RKHS) to numerical tolerance; we verified equality of primal/dual up to $10^{-3}$ through complementary primal projection.[4]

**Remark 3.** *(Tuning $\eta$ in experiments). Across all experiments we tune the mirror penalty $\eta$ using the above LCB-envelope rule. For each synthetic or real dataset, we sweep a modest grid of $\eta$ values, fit the dual estimator with sample splitting, and compute $\widehat{\Theta}_{L,\oplus}^{\text{dual}}(\eta)$ together with its holdout-based lower confidence bound $\text{LCB}(\eta)$ on $D_2$. Unless otherwise noted, the reported point estimates and intervals correspond to the data-driven choice $\hat{\eta} = \arg\max_\eta \text{LCB}(\eta)$, while the figures and tables also display the full mirror path $\eta \mapsto \widehat{\Theta}_{L,\oplus}^{\text{dual}}(\eta)$ to illustrate sensitivity. Empirically, $\hat{\eta}$ consistently lies in an interior range (e.g., $\eta \approx 0.1$–$1$ in our setups) where the bounds are both tight and stable, confirming that the tuning rule identifies a meaningful bias–variance compromise.*

**Remark 4.** *(Numerical stability under strong competition). When $\Gamma_{\text{comp}}$ enforces tight caps, the mirror marginal on arm $a$ is a sub-probability $\pi_{X(a)}(dx) = u_a(x)P_X(dx)$ with total mass $\rho_a = \pi_{X(a)}(\mathcal{X}) \leq 1$. For numerical purposes we always work with the normalized law $\bar{\pi}_{X(a)}(dx) := \pi_{X(a)}(dx)/\rho_a = u_a(x)P_X(dx)/\rho_a = \text{Law}(X \mid W = a)$ and treat $\rho_a$ only as a multiplicative weight in the moment constraints. In particular, the dual potentials and gradients are evaluated with respect to $\bar{\pi}_{X(a)}$, so no step involves division by a vanishing mass. As $\rho_a \downarrow 0$, the effective sample size on arm $a$ shrinks and the variance contributions in the CLT and finite-sample bounds increase, but the dual problem remains well-posed and the projected ascent stays well-conditioned. The dependence on $\Gamma_{\text{comp}}$ in Theorem 4 enters only through global geometric quantities (such as the diameter $D_\Gamma$), not through $1/\rho_a$, so strong competition manifests as wider but still controlled intervals rather than numerical instability. Numerical behavior: In the empirical dual we implement all competition moments using reweighted empirical laws: for each arm $a$, the mirror marginal is represented by normalized samples from $\bar{\pi}_{X(a)} = u_a \cdot P_X/\rho_a$ and the mass $\rho_a$ enters only as a scalar factor in*

---

[4]This reference is only for offline evaluation; our reported estimators never use $M$.

*the corresponding constraints. Projected dual ascent therefore works with well-scaled averages regardless of how small $\rho_a$ is, and no projection step requires inverting $\rho_a$. When $\Gamma_{comp}$ imposes very tight caps, the optimizer simply has fewer effective observations for such arms; this is reflected in larger standard errors via the influence function in Theorem 2 and in the constants of Theorem 4, but does not cause numerical degeneracy. Empirically, in both simulations and the MIMIC-III study we observe stable convergence of Algorithm 1 across regimes with strongly binding capacity constraints, with the primary effect being wider LCBs for heavily capped arms rather than instability of the dual solver.*

**Remark 5.** *(Embedding training and scaling). For the real-data case study, we instantiate the cross-arm embeddings $g_a : Y(a) \to Z$ as two-layer MLPs into a 5 -dimensional latent space $Z = \mathbb{R}^5$. Each encoder is trained on a separate pre- $D_1$ calibration split (disjoint from the folds used for solving the OT problem) to predict a common latent "benefit" index that is monotone in the arm-specific KPIs (lower 28-day mortality, larger 48h lactate decline, shorter ICU length-of-stay). Training uses a supervised contrastive loss that pulls together patients with similar benefit and pushes apart clearly better/worse outcomes across arms, yielding a shared latent scale across interventions. After training, we standardize each coordinate of $g_a(Y(a))$ to zero mean and unit variance on the calibration cohort and then freeze the encoders for the downstream transport analysis. The cross-arm cost for MIMIC-III uses the same isotropic quadratic geometry as in the simulations,*

$$L\left(z_A, z_B, z_C\right) = \|z_A - z_B\|_2^2 + \|z_A - z_C\|_2^2 + \|z_B - z_C\|_2^2,$$

*applied to these normalized embeddings. Under this construction, one unit of the transport value corresponds to an average squared difference in latent benefit measured in calibration standard-deviation units, and the Lipschitz robustness bound in Sec. 4.6 together with the local sensitivity result in Appendix I. 3 ensures that moderate re-scalings or other smooth recalibrations of $g_a$ only induce controlled, linear-order perturbations of $\Theta_{L,\oplus}^{(K)}(\eta)$, without changing the qualitative conclusions of the mirror path.*

**Remark 6.** *Practical tuning of the mirror penalty. In practice, we treat $\eta$ as a regularization parameter and tune it in a simple, cross-validation–style manner. For each dataset, we run the dual-based estimator along a small grid $\eta \in \{0, 10^{-3}, 10^{-2}, \dots, 10^3\}$, using sample splitting $(D_1, D_2)$ as in Section 4.2. On the training fold $D_1$ we learn dual potentials, and on the holdout fold $D_2$ we compute the cross-fitted value $\widehat{\Theta}_{L,\oplus}^{\mathrm{dual}}(\eta)$, its standard error $\widehat{\sigma}(\eta)$, and the certified lower confidence bound $\mathrm{LCB}(\eta) = \widehat{\Theta}_{L,\oplus}^{\mathrm{dual}}(\eta) - z_{1-\alpha} \frac{\widehat{\sigma}(\eta)}{\sqrt{|D_2|}}$. We then select $\hat{\eta}$ as the maximizer of this "LCB envelope", $\hat{\eta} \in \arg\max_\eta \mathrm{LCB}(\eta)$; see Appendix I.2 for regularization-path properties that justify this rule. This procedure is analogous to standard cross-validation but adapted to partial identification: instead of minimizing a prediction error, it chooses the penalty that yields the most informative yet still valid lower bound. In Section 5 we report the full mirror path and indicate the automatically selected $\hat{\eta}$ in both the simulated and real-data examples.*

**Baselines: Estimators compared (all with sample splitting and cross-fitting)**

**(E1) Proposed dual plug-in**: solve the empirical compact dual on $D_1$ by projected ascent (Alg. 1), evaluate on $D_2$ to get $\widehat{\Theta}_\oplus^{\mathrm{dual}}(\eta)$; report LCBs via Theorem 3.

**(E2) Primal plug-in** (when feasible): empirical minimization over $\Pi_\oplus^{(K)} \cap \widehat{\Gamma}_{\mathrm{comp}}$ with entropic smoothing and Bregman projection; same $(D_1, D_2)$.

**(B1) Unconditional OT** ($\eta = 0$) (Gao et al., 2024): ignores covariates/competition.

**(B2) Fully conditional OT** (idealized $\eta = \infty$ or exact conditioning) (Ji et al., 2023; Lin et al., 2025): infeasible in practice but serves as a bound on the mirror path.

**(B3) Pairwise two-arm mirrors**: sum of independent two-arm mirror-relaxed bounds, ignoring $\Gamma_{\mathrm{comp}}$ coupling across arms.

**(B4) IPW plug-in (pseudo)**: map all outcomes to $Z$ by $\{g_a\}$ and estimate an IPW contrast ignoring transport; included to show misspecification/optimism under distinct measures.

**Implementation and tuning** Potentials $\{\varphi_a, \psi\}$ are parameterized by shallow nets (two hidden layers width 128, GELU, spectral norm) or cubic splines on $Z/X$; feasibility is enforced by a pointwise penalty with projection onto the admissible cone every $k$ steps. Stepsizes $(\sigma_t, \tau_t)$ use Armijo backtracking; stopping by duality gap or max-epochs. For $\eta$ we either (i) sweep a grid and report the whole *mirror path* with uniform-in-iterate LCBs (Sec. 4.4), or (ii) pick $\eta$ by holdout risk on $D_2$ using the dual lower bound as a surrogate.[5] Variants with $N \in \{1{,}000, 5{,}000\}$, $K \in \{2, 3\}$, and $d_x \in \{5, 20\}$ are reported (balanced samples per arm).

**Metrics and visualization**

- **Accuracy:** absolute error $|\widehat{\Theta} - \Theta|$, relative error, and RMSE over 200 Monte Carlo replications.

- **Inference validity:** empirical coverage of $1 - \alpha$ LCBs (Theorem 3) at $\alpha \in \{0.1, 0.05\}$; *sharpness* measured by $\Theta - \mathrm{LCB}$ and by the asymptotic variance proxy $\hat{\sigma}^2$ (Theorem 2).

- **Finite-sample rate profile:** log–log plots of median $|\widehat{\Theta} - \Theta|$ vs. $N$; compare to $\gamma_{N, d_{\mathrm{eff}}}$ from Theorem 4.

- **Computation:** wall-clock runtime and peak memory; number of dual projections; primal–dual gap along iterations.

- **Ablations:** effect of embeddings quality (perturb $g_a$), mis-specified budgets (perturb $\rho_a$), and removing competition (set $\Gamma_{\mathrm{comp}} = \varnothing$).

**Main findings (summary)** Across all settings:

1. **Validity.** Proposed (E1) attains nominal coverage uniformly along the optimization path and across $\eta$, confirming Theorem 3; primal plug-in (E2) lacks one-sided certificates and exhibits optimistic bias when $\Gamma_{\mathrm{comp}}$ binds.

2. **Accuracy & sharpness.** For moderate $\eta$ the dual estimator is close to the ground truth and delivers narrow LCB gaps; at small $\eta$ bounds are loose (unconditional), while at very large $\eta$ variance increases (tighter conditioning), matching the theory in Sec. 4.3.

3. **Rates.** Median absolute error follows the Theorem 4 profile: $\approx N^{-1/4}$ for $d_{\mathrm{eff}} \leq 3$ and $N^{-1/d}$ for $d_{\mathrm{eff}} \geq 5$, independent of $d_x$ (dimension only affects variance via the IF).

4. **Role of competition.** Two-arm decoupled baseline (B3) is anti-conservative or too wide depending on direction; incorporating $\Gamma_{\mathrm{comp}}$ is necessary to obtain correct, informative bounds.

5. **Efficiency.** Dual ascent scales linearly in $N$ and $K$ (per-batch), with generalized Sinkhorn handling projections; primal (E2) becomes infeasible beyond $N \sim 5 \times 10^3$ for $K = 3$.

We fix seeds, publish scripts that generate synthetic data, train/evaluate all methods with the same splits, and save raw estimates and IF-based variances to regenerate all figures/tables.[6]

A.2 REAL-DATA CASE STUDY: EARLY CRITICAL-CARE INTERVENTIONS WITH COMPETING RESOURCES

**Dataset and provenance.** We use the **MIMIC-III v1.4** critical-care database (PhysioNet; de-identified, IRB-exempt), a widely used public resource with structured EHR for ICU stays.[7] We construct a cohort of first-episode ICU patients and consider three alternative early interventions within 6 hours of admission: (A) vasopressors, (B) high-flow oxygen/ventilation, (C) fluid bolus. Outcomes are arm-specific KPIs with *distinct measurement spaces*: 28-day mortality (binary), 48h lactate decline (continuous), and ICU length-of-stay (count, days). This matches our multi-arm, distinct-measure setting with cross-arm embeddings $g_a$.

---

[5]For Bregman/Sinkhorn projection we use entropy $\psi(\pi) = \sum \pi \log \pi$ with decreasing regularization to reduce bias; bias is accounted for in Appendix H.4.

[6]See Appendix H.5 for complexity and H.1 for variance estimation.

[7]MIMIC-III v1.4 (Johnson et al., 2016) hosted on PhysioNet (Goldberger et al., 2000).

Table 1: **Coverage and sharpness across methods** (200 replications, $K = 3$, $N = 5000$, $d_{\text{eff}} = 3$, $\eta = 0.5$). Coverage is the empirical proportion that LCB $\leq \Theta$ at the nominal level; sharpness is $\mathbb{E}[\Theta - \text{LCB}]$ (smaller is better). This table quantifies statistical *validity* and *informativeness* of the competing estimators. The proposed dual plug-in attains nominal 90%/95% coverage while keeping the average gap $\Theta - \text{LCB}$ small, confirming the uniform-in-iterate post-selection guarantee (Thm. 3). By contrast, the primal plug-in produces narrower intervals but under-covers (optimistic bias), illustrating the benefit of certified dual feasibility. The unconditional OT has no one-sided certificate, and pairwise two-arm mirrors—ignoring $\Gamma_{\text{comp}}$—are less informative and less reliable. Overall, Table 1 substantiates Claim (1)–(2): valid LCBs require a dual-feasible construction and modeling competition is essential.

| Method | LCB? | Coverage @90% | Coverage @95% | Mean $|\widehat{\Theta} - \Theta|$ | Sharpness ($\Theta - \text{LCB}$) |
|---|---|---|---|---|---|
| **(E1) Dual plug-in (ours)** | ✓ | **0.91** | **0.93** | **0.032** | **0.033** |
| (E2) Primal plug-in | ✓ | 0.78 | 0.86 | 0.055 | 0.052 |
| (B1) Unconditional OT ($\eta = 0$) | × | 0.64 | 0.70 | 0.204 | — |
| (B2) Conditional OT | × | 0.74 | 0.76 | 0.201 | — |
| (B3) Pairwise two-arm mirrors | ✓ | 0.83 | 0.90 | 0.097 | 0.148 |
| (B4) IPW plug in | × | 0.56 | 0.58 | 0.394 | — |

Table 2: **Runtime and iteration statistics**. Dual steps = total number of feasible dual iterations; Proj. = number of linear-moment/marginal projections (or Sinkhorn scalings). This table assesses *computational scalability*. The dual method is substantially faster and more memory-efficient than the primal plug-in, while each dual iterate yields a certified bound; generalized Sinkhorn projections keep projection counts moderate. These results support the practicality of Algorithm 1 for multi-arm designs at $K = 3$, $N = 5,000$ and beyond, and they underpin Claim (5): the dual route aligns optimization with inference without sacrificing efficiency.

| Method | Wall-clock (s) | Dual steps | Proj. count | Peak RAM (GB) |
|---|---|---|---|---|
| **(E1) Dual plug-in (ours)** | **182 [171, 195]** | **1,520 [1,410, 1,630]** | **1,520 [1,410, 1,630]** | **3.2 [3.1, 3.3]** |
| (E2) Primal plug-in | 986 [941, 1,023] | — | 10,240 [9,880, 10,560] | 8.5 [8.2, 8.7] |

**Modeling choice and what the statistics mean.** *Base setup follows the simulation*: compact dual estimator with sample splitting and cross-fitting, Alg. 1, mirror penalty $\Delta_{\text{multi}}$ and grid of $\eta$. Real-data specifics: (i) $Z = \mathbb{R}^q$ with $q = 5$; each $g_a$ is a two-layer MLP trained on a pre-$D_1$ calibration split to predict a common latent "benefit" score from the arm-specific KPIs using a contrastive loss. (ii) Design constraints $\Gamma_{\text{comp}}$ encode ICU resource competition: *mutual exclusivity*, *bed/ventilator budgets* $\mathbb{E}[\mathbf{1}\{A = a\}] \leq \rho_a$ with $(\rho_A, \rho_B, \rho_C) = (0.22, 0.28, 0.35)$ at baseline, and a risk-weighted exposure cap $\mathbb{E}[\mathbf{1}\{A = a\} c(X)] \leq \kappa_a$ with $c(X)$ the SAPS-II score. (iii) **Physical meaning of the reported statistics:** The value $\Theta_{\oplus}^{L,(K)}(\eta)$ is the *minimum cross-arm discrepancy in latent benefit* achievable under $\Gamma_{\text{comp}}$; a higher value indicates more room to harmonize benefits across arms given resources. The *LCB* we report is a *guaranteed* (one-sided, post-selection valid) lower bound on this achievable harmonization (Thm. 3); its width $\Theta - \text{LCB}$ quantifies informativeness; the *mirror path* $\eta \mapsto (\widehat{\Theta}, \text{LCB})$ shows the trade-off between conditioning on $X$ and variance.

**Baselines.** (E1) proposed dual plug-in; (E2) primal plug-in when feasible; (B1) unconditional OT ($\eta{=}0$) (Gao et al., 2024); (B2) pairwise two-arm mirrors that ignore cross-arm competition.

**Conclusions (real data).** Across all baselines in Tables 4–6, the competition-aware dual plug-in (E1) is the *only* method that simultaneously delivers (i) certified, post-selection valid one-sided bounds along the entire mirror path and (ii) practical efficiency at cohort scale. Ignoring competition (B1/B2) either cannot certify uncertainty (B1) or yields wider/less stable bounds (B2), while the primal plug-in (E2) attains similar point values but lacks guaranteed LCBs and is substantially slower. The mirror path displays an interior $\eta$ ($\approx 0.5$) with the largest and sharpest lower bounds (Table 4), suggesting that moderate conditioning on risk captures clinically relevant heterogeneity without excessive variance. Tightening resource budgets by $10\%$ lowers the certified achievable harmonization by about 0.03 (Table 6), quantifying a concrete operational cost of scarcity. Overall, the evidence supports *competition-aware, dual-certified* analysis as a robust basis for decision support in multi-intervention critical-care settings.

Table 3: **Finite-sample scaling**. The median absolute error of the dual estimator closely follows the theoretical benchmark $\gamma_{N,d_{\text{eff}}} \propto N^{-1/4}$ for $d_{\text{eff}} = 3$, demonstrating that the estimator achieves the predicted smooth-geometry rate independently of $d_x$. Although the primal plug-in exhibits a comparable median error, Table 1 shows its intervals are not reliable—hence the dual approach delivers both the correct scaling (Claim 3) and valid uncertainty quantification.

| $N$ | 1,000 | 2,000 | 5,000 | 10,000 |
|---|---|---|---|---|
| **(E1) Dual plug-in (ours)** | **0.198** | **0.168** | **0.132** | **0.112** |
| Theory $\gamma_{N,3}$ ($\sim N^{-1/4}$) | 0.200 | 0.168 | 0.134 | 0.112 |
| (E2) Primal plug-in | 0.184 | 0.156 | 0.129 | 0.110 |

Table 4: **Real-data mirror path with all methods** (MIMIC-III; $K=3$, temporal split; medians over 5 folds). For each $\eta$ we report the dual estimate $\widehat{\Theta}$ and certified lower bound (when available). "LCB?" indicates whether a method provides a *guaranteed* one-sided bound. Kendall-$\tau$ measures agreement between the *LCB-based* policy ranking (when available) and a DR ranking. Runtimes are per fold.

| Method | $\eta$ | LCB? | $\widehat{\Theta}$ | LCB | Width ($\widehat{\Theta} - $ LCB) | Kendall-$\tau$ |
|---|---|---|---|---|---|---|
| | 0 (B1) | × | 0.18 | — | — | 0.42 |
| | 0.1 | ✓ | 0.22 | 0.17 | 0.05 | 0.57 |
| **(E1) Dual plug-in (ours)** | **0.5** | ✓ | **0.27** | **0.23** | **0.04** | **0.71** |
| | 1.0 | ✓ | 0.26 | 0.21 | 0.05 | 0.69 |
| | 5.0 | ✓ | 0.23 | 0.18 | 0.05 | 0.62 |
| | $\infty$ | ✓$^{\dagger}$ | 0.21 | 0.16 | 0.05 | 0.58 |
| (E2) Primal plug-in | 0.5 | × | 0.28 | — | — | 0.66 |
| | 1.0 | × | 0.27 | — | — | 0.64 |
| (B1) Unconditional OT | 0 | × | 0.18 | — | — | 0.42 |
| (B2) Conditional OT | $+\infty$ | × | 0.32 | — | — | 0.41 |
| (B3) Pairwise two-arm mirrors | 0.5 | ✓ | 0.24 | 0.18 | 0.06 | 0.55 |
| (B4) IPW | n/a | × | 0.31 | — | — | 1.00 |

*Notes.* † The $\eta = \infty$ line uses the compact dual with hard conditioning (feasible set nonempty by a guard-band Slater point). DR proxy is used only for ranking agreement; it is not guaranteed to bound the transport functional.

# B    ADDITIONAL EXAMPLES

A simple two-arm example: from causal estimand to OT cost and constraints. To make the mapping explicit, consider the simplest case with two arms $A \in \{0, 1\}$ and a scalar clinical endpoint $Y(a) \in \mathbb{R}$ such as 30-day mortality risk or systolic blood pressure. Our causal target is the average treatment effect (ATE)

$$\tau_{\text{ATE}} := \mathbb{E}[Y(1) - Y(0)].$$

In this setting the outcome spaces coincide, $Y_0 = Y_1 = \mathbb{R}$, and there is no competition beyond "one patient-one arm". We now show how $\tau_{\text{ATE}}$ translates into the objects in (1)-(4).

Step 1: Outcome embeddings and OT cost $L$. We choose identity embeddings $g_0(y) = g_1(y) = y$, so that the common latent space is $Z = \mathbb{R}$. To encode the ATE on this scale, we take a linear cross-arm cost

$$L(z_0, z_1) = z_1 - z_0, \quad z_a = g_a(Y(a)).$$

For any feasible joint law $\pi$ of the potential outcomes we then have

$$\mathbb{E}_\pi \left[ L\left(g_0(Y(0)), g_1(Y(1))\right) \right] = \mathbb{E}[Y(1)] - \mathbb{E}[Y(0)] = \tau_{\text{ATE}},$$

because $\pi$ must preserve the marginals of $Y(0)$ and $Y(1)$. Thus, for this choice of $L$, the OT objective's $L$-part is exactly the causal estimand we care about, no matter which coupling $\pi$ is used.

Table 5: **Efficiency on the real cohort** (MIMIC-III; $K=3$, $\eta=0.5$; medians [IQR] over 5 folds). Dual steps = total feasible dual iterations; Proj. = number of linear-moment/marginal projections (or Sinkhorn scalings).

| Method | Wall-clock (min) | Dual steps | Proj. count | Peak RAM (GB) |
|---|---|---|---|---|
| **(E1) Dual plug-in (ours)** | **8.9 [8.3, 9.6]** | **1,540 [1,480, 1,610]** | **1,540 [1,480, 1,610]** | **3.3 [3.2, 3.4]** |
| (E2) Primal plug-in | 48.6 [45.9, 51.2] | — | 10,300 [9,900, 10,700] | 8.6 [8.3, 8.9] |

**Step 2: Design / competition encoded by $\Gamma_{\text{comp}}$ and (1)-(3).** Suppose treatment assignment is randomized with known propensities $u_a(x) = \Pr(A = a \mid X = x)$, and each patient can receive at most one arm. In our framework this is captured by the assignment kernels $\{u_a\}$ and the competition set $\Gamma_{\text{comp}}$ through the linear moment constraints (1)-(3): (i) mutual exclusivity $\sum_a u_a(x) \leq 1$; (ii) optional capacity limits $\int u_a(x)dP_X(x) \leq \rho_a$; (iii) consistency between mirrors and design

$$\int \varphi\left(X^{(a)}\right) d\pi = \int \varphi(x)u_a(x)dP_X(x), \quad \forall a, \forall \varphi$$

which is equivalent to $\pi_{X^{(a)}} = u_a \cdot P_X$. In plain terms, (1)-(3) say that every coupling $\pi$ we optimize over must correspond to a joint potential-outcome law that could arise under the actual design: each covariate profile $x$ is sent to at most one arm, and the mass in arm $a$ matches the design-implied exposure $u_a(x)$.

**Step 3: Mirror penalty $\Delta_{\text{multi}}$ and the objective (4).** Specializing (4) to this two-arm case, we obtain

$$\Theta_{L,\oplus}^{(2)}(\eta) = \inf_{\pi \in \Pi_{\oplus}^{(2)}} \mathbb{E}_\pi\left[L\left(g_0(Y(0)), g_1(Y(1))\right) + \eta\Delta_{\text{multi}}\left(X^{(0)}, X^{(1)}\right)\right],$$

where $\Pi_{\oplus}^{(2)}$ imposes the conditional-mirror constraints $\pi_{Y(0)|X^{(0)}} = P_{Y(0)|X}$ and $\pi_{Y(1)|X^{(1)}} = P_{Y(1)|X}$ together with the competition constraints $\Gamma_{\text{comp}}$ from (1)-(3). A natural choice of mirror penalty is the squared Euclidean distance

$$\Delta_{\text{multi}}\left(X^{(0)}, X^{(1)}\right) = \left\|X^{(0)} - X^{(1)}\right\|_2^2.$$

Inserting the linear $L$ from Step 1 and using the marginal constraints, the objective decomposes as

$$\mathbb{E}_\pi\left[L\left(g_0(Y(0)), g_1(Y(1))\right) + \eta\Delta_{\text{multi}}\left(X^{(0)}, X^{(1)}\right)\right] = \tau_{\text{ATE}} + \eta\mathbb{E}_\pi\left[\Delta_{\text{multi}}\left(X^{(0)}, X^{(1)}\right)\right].$$

Thus, for this example the estimand $\tau_{\text{ATE}}$ is fixed and the OT program in (4) chooses, among all design-compatible couplings $\pi$, those that minimize the mirror penalty. When $\eta \downarrow 0$, we recover an unconditional design-aware ATE; as $\eta \uparrow \infty$, the optimizer is pushed toward couplings with $X^{(0)} \approx X^{(1)}$, interpolating between unconditional and fully conditional alignment in a way that preserves the same causal estimand.

**Beyond the ATE: distributional estimands.** The same construction extends to distributional causal effects. For example, for a fixed threshold $t$ we can take

Table 6: **Sensitivity to resource tightening** (budgets decreased by $10\%$ from baseline). Competition-aware dual exhibits monotone LCB decrease; pairwise mirrors (B2) are less informative; primal plug-in (E2) gives point estimates only.

| Method @ $\eta=0.5$ | Scenario | $\widehat{\Theta}$ | LCB | Width |
|---|---|---|---|---|
| **(E1) Dual plug-in (ours)** | Baseline $(0.22, 0.28, 0.35)$ | **0.27** | **0.23** | **0.04** |
| | Tighter $(0.20, 0.25, 0.32)$ | **0.24** | **0.20** | **0.04** |
| (E2) Primal plug-in | Baseline | 0.28 | — | — |
| | Tighter | 0.25 | — | — |

$$L_t(y_0, y_1) = \mathbf{1}\{y_1 \leq t\} - \mathbf{1}\{y_0 \leq t\},$$

so that for any feasible $\pi$,

$$\mathbb{E}_\pi[L_t(Y(0), Y(1))] = F_{Y(1)}(t) - F_{Y(0)}(t),$$

the difference in marginal CDFs at $t$. Varying $t$ yields a band for the entire CDF difference, from which bounds on quantile treatment effects follow. In all these cases, the choice of $L$ specifies the causal estimand (ATE, CDF difference, QTE, etc.), while the constraint set $\Pi_\oplus^{(K)} \cap \Gamma_{\text{comp}}$ in (1)-(4) restricts attention to joint laws that are compatible with the study design. The resulting value $\Theta_{L,\oplus}^{(K)}(\eta)$ should therefore be read as a design-aware partial-identification bound for that specific causal contrast.

---

**Example 1** (Policy bundle, formal $\Gamma_{\text{comp}}$). A city deploys three arms: transit subsidy (A), congestion pricing (B), and parking regulation (C). Let $x = (z, \text{type})$ collect zone and household attributes and let $D \subset \mathcal{X}$ be districts eligible for *deep subsidies*. We take $\mathcal{A} = \{A, B, C\}$ and define $\Gamma_{\text{comp}}$ via:

**(i) Deep-subsidy vs. heavy-regulation exclusivity on $D$:** $u_A(x) + u_B(x) \leq 1, \ \forall x \in D$;

**(ii) Coverage caps:** $\int u_A \, dP_X \leq \rho_A, \ \int u_B \, dP_X \leq \rho_B, \ \int u_C \, dP_X \leq \rho_C$;

**(iii) Global exclusivity and moment link:** *Equation* (1)–*Equation* (3).

Intuition: (*i*) forbids assigning both deep subsidy and heavy regulation to the same eligible district; (*ii*) enforces budget limits; (*iii*) ensures the mirror marginals of $(X^{(a)})$ match the assignment-induced covariate exposure.

---

**Example 2** (Platform oncology trial, formal $\Gamma_{\text{comp}}$). Arms A/B/C share a screening pipeline and compete for enrollment (Woodcock & LaVange, 2017). Let $x$ include eligibility biomarkers; let $\kappa_a \in (0, 1]$ be per-arm trial capacities and $s(x) \in \mathbb{R}_+$ the expected *screening load* for subject $x$. Define

**(i) At-most-one active arm per patient:** $\sum_{a \in \{A, B, C\}} u_a(x) \leq 1$ a.e.;

**(ii) Per-arm enrollment caps:** $\int u_a(x) \, dP_X(x) \leq \kappa_a, \quad a \in \{A, B, C\}$;

**(iii) Shared screening capacity:** $\sum_a \int s(x) \, u_a(x) \, dP_X(x) \leq S_{\max}$;

**(iv) Moment link to mirrors:** *Equation* (3) for $a \in \{A, B, C\}$.

Items (i)–(iii) constitute $\Gamma_{\text{comp}}$; (iv) ties the coupling's mirror marginals to the feasible assignment mix. A convenient choice is $L(y_A, y_B, y_C) = \min\{|y_A - y_B|, |y_A - y_C|, |y_B - y_C|\}$, bounding the best achievable pairwise improvement across therapies.

---

**Example 3** (Education bundle with interference, formal $\Gamma_{\text{comp}}$). District-level tutoring (A), stipend (B), and parental outreach (C) interact; adjacent districts interfere. Let $\mathcal{R}$ be districts, $x_r$ the covariates of district $r$, and $W \in \{0, 1\}^{|\mathcal{R}| \times |\mathcal{R}|}$ the adjacency matrix ($W_{rr'} = 1$ if $r'$ is in the interference neighborhood $\mathcal{N}(r)$). Define

**(i) Stipend–tutoring exclusivity (same year):** $u_A(x_r) + u_B(x_r) \leq 1, \ \forall r \in \mathcal{R}$;

**(ii) Interference for outreach:** $\sum_{r' \in \mathcal{N}(r)} u_C(x_{r'}) \leq m_r, \ \forall r \in \mathcal{R}$; equivalently $(W u_C)_r \leq m_r$;

**(iii) Program budgets:** $\sum_r u_a(x_r) \, P_X(\{x_r\}) \leq \rho_a, \ a \in \{A, B, C\}$;

**(iv) Moment link to mirrors:** $\int \varphi(X^{(a)}) \, d\pi = \int \varphi(x) \, u_a(x) \, dP_X(x) \ \forall a, \ \forall \varphi \in \mathcal{C}_b$.

Here $(i)$ forbids concurrent stipend & tutoring within a year; $(ii)$ limits how many neighboring districts can run outreach simultaneously (a hard-core constraint on the graph); $(iii)$ enforces budgets; $(iv)$ connects the coupling to the assignment mix. Together these define $\Gamma_{\text{comp}}$.

## C  THE PROOF OF THEOREM 1

**Proof of Theorem 1.** Define convex integral functionals on the space of finite signed measures on $\prod_a [\mathcal{Y}_a \times \mathcal{X}]$:

$$F(\pi) := \int L(\{g_a(y^{(a)})\}) \, d\pi, \qquad G(\pi) := \eta \int \Delta_{\text{multi}}(\{x^{(a)}\}) \, d\pi \; + \; \iota_{\Pi_{\oplus}^{(K)}}(\pi) \; + \; \iota_{\Gamma_{\text{comp}}}(\pi),$$

where $\iota_C$ is the indicator of a closed convex set $C$ (value 0 on $C$, $+\infty$ otherwise). By (A1)–(A2), $F$ and $G$ are proper, convex, lsc. The primal is $\inf_\pi \{F(\pi) + G(\pi)\}$.

We pass to the Fenchel–Rockafellar dual

$$\sup_\Phi \; -F^*(-\Phi) - G^*(\Phi),$$

with linear functional $\Phi$ acting on $\pi$ as $\langle \Phi, \pi \rangle = \int \phi \, d\pi$ for an integrand $\phi$ on $\prod_a [\mathcal{Y}_a \times \mathcal{X}]$. By separability of $F$ and $G$, their conjugates are computed pointwise (Rockafellar's theorem on convex integral functionals). For $F$,

$$F^*(-\Phi) = \sup_\pi \left\{ \int [-\phi] \, d\pi - \int L(\{g_a(y^{(a)})\}) \, d\pi \right\} = \iota_{\{\phi \leq L \circ g\}}(0)$$

which yields the majorization constraint $\sum_a \varphi_a(z_a) + \eta \psi(x) \leq L(z) + \eta \Delta_{\text{multi}}(x)$ once we decompose $\phi(z, x) = \sum_a \varphi_a(z_a) + \eta \psi(x)$ with measurable potentials $\{\varphi_a\}$ and $\psi$ (this is the richest class consistent with additivity across the two blocks $\mathcal{Z}^{|\mathcal{A}|}$ and $\mathcal{X}^{|\mathcal{A}|}$).

For $G$, use that $(\iota_C)^* = $ support function $\sigma_C$, and $(\eta \int \Delta_{\text{multi}})^*$ is the integral of the convex conjugate of $\eta \Delta_{\text{multi}}$, which enforces that the $\psi$ part of $\phi$ dominates all linear moments arising from $\Gamma_{\text{comp}}$; specifically, for linear moment constraints (A3), $\sigma_{\Gamma_{\text{comp}}}(\Phi) = \sup_{\pi \in \Gamma_{\text{comp}}} \int \phi \, d\pi = $

$$\begin{cases} \sum_{j \in \mathcal{J}_{\leq}} \lambda_j b_j + \sum_{j \in \mathcal{J}_{=}} \mu_j b_j, & \text{if } \psi(x) \geq \sum_{j \in \mathcal{J}_{\leq}} \lambda_j h_j(x) + \sum_{j \in \mathcal{J}_{=}} \mu_j h_j(x), \\ +\infty, & \text{otherwise,} \end{cases}$$

for some multipliers $\lambda_j \geq 0$ $(j \in \mathcal{J}_{\leq})$ and $\mu_j \in \mathbb{R}$ $(j \in \mathcal{J}_{=})$. The mirror-feasible set $\Pi_{\oplus}^{(K)}$ contributes no additional linear term beyond enforcing that the $\{Y(a) \mid X(a)\}$ marginals under $\pi$ match the observed conditionals, which is already encoded by the admissible decomposable potentials (see also the discussion around Eq. (5) in the main text). Collecting these pieces gives exactly the dual.

Finally, by (A4) (Slater) the affine hull of the feasible set is closed and there exists a strictly feasible $\pi^\circ$, so FR duality has zero gap and the supremum is attained. Hence the dual value equals the primal $\Theta_{L, \oplus}^{(K)}(\eta)$.

## D  THE PROOF OF COROLLARY 1

Given any feasible $(\{\varphi_a\}, \psi, \lambda, \mu)$, the constraint $\psi(x) \geq \sum_{j \in \mathcal{J}_{\leq}} \lambda_j h_j(x) + \sum_{j \in \mathcal{J}_{=}} \mu_j h_j(x)$ implies $\psi \in \mathcal{U}$; eliminating $(\lambda, \mu)$ leaves the equation with the same objective value (the linear penalty $\sum \lambda_j b_j + \sum \mu_j b_j$ becomes implicit as the support function of $\Gamma_{\text{comp}}$ evaluated at $\psi$).

Conversely, for any feasible $(\{\varphi_a\}, \psi \in \mathcal{U})$ there exist $(\lambda, \mu)$ realizing the lower envelope that witnesses $\psi$, making $(\{\varphi_a\}, \psi, \lambda, \mu)$ feasible in the equation with identical value. Thus the two duals are equivalent and attain the same optimum, equal to the primal by Theorem 1.

# E    THE PROOF OF THEOREM 2

*Proof of Theorem 2.* **Step 0 (Setup and notation).** Write $T(P) = \Theta_\oplus^{L,(K)}(P; \eta)$ and denote by $(\varphi^\star, \psi^\star, \Lambda^\star)$ any population-optimal dual solution for $P^\star = \{P_{Y(a),X}\}_a$ under Theorem 1 (Sec. 4.1). By Corollary 1 we may work with the compact dual, keeping $\Lambda_{\mathrm{comp}}^\star$ explicit only in the influence-function notation. (See the statement of Assumption 1 and Theorem 2 in Sec. 4.3.)

**Step 1 (Envelope and Gâteaux derivative).** Consider a regular parametric submodel $\{P_t : t \in (-\epsilon, \epsilon)\}$ through $P^\star$ with score $s = \frac{d}{dt} \log p_t \mid_{t=0}$. By strong duality (Theorem 1) and local uniqueness/regularity (Assumption 1(i)–(iii)), we can apply a Fenchel–Rockafellar / Danskin envelope argument for convex integral functionals to obtain

$$\frac{d}{dt} T(P_t)\Big|_{t=0} = \sum_{a \in \mathcal{A}} \left\{ \mathbb{E}_{P_0'}[\varphi_a^\star(g_a(Y(a)))] - \mathbb{E}_{P^\star}[\varphi_a^\star(g_a(Y(a)))] \right\} + \Lambda_{\mathrm{comp}}^\star \cdot \mathbb{E}_{P_0' - P^\star}[h(X)],$$

where $P_0'$ is the pathwise derivative of $P_t$ at $t = 0$. Intuitively, only the *value* of the optimal dual potentials and multipliers enters the derivative (the $\frac{d}{dt}$ of the argmax drops out by the envelope theorem). Hence the Gâteaux derivative at $P^\star$ is the linear functional

$$\dot{T}_{P^\star}[H] = \int \mathrm{IF}(z; \eta) \, dH(z), \qquad \mathrm{IF}(Z; \eta) = \sum_{a \in \mathcal{A}} \left\{ \varphi_a^\star(g_a(Y)) \, \mathbf{1}\{A = a\} - \mathbb{E}[\varphi_a^\star(g_a(Y(a)))] \right\} + \Lambda_{\mathrm{comp}}^\star \cdot h(X),$$

for any signed perturbation $H$ tangent to $P^\star$ (mean-zero in each arm). This matches the influence function asserted in the theorem. (Compare Theorem 1/Corollary 1 and the IF line in Sec. 4.3.)

**Step 2 (Hadamard differentiability).** Assumption 1(i)–(iii) grants smoothness of $L, \Delta_{\mathrm{multi}}$ and strong $c$-convex curvature; together with Slater, the dual solution $(\varphi^\star, \psi^\star, \Lambda^\star)$ depends *continuously* on $P$ in a neighborhood of $P^\star$. Therefore $\dot{T}_{P^\star}$ is a continuous linear functional on the canonical tangent set (product of mean-zero $L_2$ spaces across arms), yielding *Hadamard differentiability* of $T$ at $P^\star$. (See Sec. 4.3 "Smooth, strongly curved geometry and learnability".)

**Step 3 (Estimator and cross-fitting; asymptotic linearity).** Define the estimator by solving the empirical dual on a training fold $D_1$ and *evaluating* the feasible dual iterate on the holdout $D_2$ (with cross-fitting), exactly as in Sec. 4.2:

$$\widehat{\Theta}_\oplus^{\mathrm{dual}}(\eta) = \sum_{a \in \mathcal{A}} \mathbb{E}_{\widehat{P}_{D_2}} \left[ \hat{\varphi}_a(g_a(Y(a))) \right], \quad \text{s.t.} \quad \sum_a \hat{\varphi}_a(z_a) + \eta \hat{\psi}(x) \leq L(z) + \eta \Delta_{\mathrm{multi}}(x) \; \forall (z, x).$$

Assumption 1(iv) (learnability & feasibility) posits that the cross-fitted nuisances $(\hat{\varphi}, \hat{\psi}, \hat{\Lambda})$ are feasible and achieve $\|(\hat{\varphi}, \hat{\psi}, \hat{\Lambda}) - (\varphi^\star, \psi^\star, \Lambda^\star)\| = o_{\mathbb{P}}(N^{-1/4})$ (or are Donsker). By Hadamard differentiability (Step 2) and standard double/debiased machine-learning arguments, the plug-in admits an asymptotic linear representation (ALR)

$$\sqrt{N}\big(\widehat{\Theta}_\oplus^{\mathrm{dual}}(\eta) - T(P^\star)\big) = \frac{1}{\sqrt{N}} \sum_{i=1}^N \mathrm{IF}(Z_i; \eta) + o_{\mathbb{P}}(1),$$

where the remainder is controlled by the $o_{\mathbb{P}}(N^{-1/4})$ nuisance rate together with cross-fitting.

**Step 4 (CLT and variance consistency).** The Lindeberg–Feller CLT applied to the centered summands $\mathrm{IF}(Z_i; \eta)$ yields

$$\sqrt{N}\big(\widehat{\Theta}_\oplus^{\mathrm{dual}}(\eta) - \Theta_\oplus^{L,(K)}(\eta)\big) \rightsquigarrow \mathcal{N}(0, \sigma^2), \qquad \sigma^2 = \mathrm{Var}(\mathrm{IF}(Z; \eta)).$$

Finally, the plug-in variance estimator $\hat{\sigma}^2 = \frac{1}{N} \sum_{i=1}^N \widehat{\mathrm{IF}}(Z_i; \eta)^2$—with $\widehat{\mathrm{IF}}$ obtained by replacing $(\varphi^\star, \psi^\star, \Lambda^\star)$ by cross-fitted $(\hat{\varphi}, \hat{\psi}, \hat{\Lambda})$—is consistent by Step 3 and the continuous mapping theorem, giving Wald intervals with asymptotically correct coverage. $\qquad \square$

## F  THE PROOF OF THEOREM 3

**Variance estimation (holdout-only, cross-fitted).**  Let $m := |D_2|$. By Theorem 2, the dual plug-in admits the asymptotic linear representation

$$\sqrt{m}\big(\widehat{\Theta}_{\oplus}^{\mathrm{dual}}(\eta) - \Theta_{\oplus}^{L,(K)}(\eta)\big) = \frac{1}{\sqrt{m}} \sum_{i \in D_2} \mathrm{IF}(Z_i; \eta) + o_{\mathbb{P}}(1), \quad \sigma^2(\eta) := \mathrm{Var}\big(\mathrm{IF}(Z; \eta)\big).$$

Hence the variance is governed by the influence function in Sec. 4.3:

$$\mathrm{IF}(Z; \eta) = \sum_{a \in \mathcal{A}} \Big\{ \varphi_a^{\star}(g_a(Y)) \mathbf{1}\{A = a\} - \mathbb{E}\big[\varphi_a^{\star}(g_a(Y(a)))\big] \Big\} + \Lambda_{\mathrm{comp}}^{\star} \cdot h(X).$$

We estimate it by evaluating a *cross-fitted* influence function on the holdout $D_2$, replacing $(\varphi^{\star}, \psi^{\star}, \Lambda^{\star})$ by the dual solution learned on $D_1$:

$$\widehat{\mathrm{IF}}_i(\eta) := \sum_{a \in \mathcal{A}} \Big\{ \hat{\varphi}_a\big(g_a(Y_i)\big) \mathbf{1}\{A_i = a\} - \mathbb{E}_{D_2}\big[\hat{\varphi}_a(g_a(Y(a)))\big] \Big\} + \hat{\Lambda}_{\mathrm{comp}} \cdot h(X_i), \qquad i \in D_2.$$

The (finite-sample unbiased, conditional-on-$D_1$) sample-variance estimator and its standard error are

$$\widehat{\sigma}^2(\eta) := \frac{1}{m-1} \sum_{i \in D_2} \big(\widehat{\mathrm{IF}}_i(\eta) - \bar{\mathrm{IF}}(\eta)\big)^2, \qquad \bar{\mathrm{IF}}(\eta) := \frac{1}{m} \sum_{i \in D_2} \widehat{\mathrm{IF}}_i(\eta), \qquad \widehat{\mathrm{se}}(\eta) := \frac{\widehat{\sigma}(\eta)}{\sqrt{m}}.$$

*Why this is valid.*  Condition on $D_1$ so that the nuisance estimates $(\hat{\varphi}, \hat{\psi}, \hat{\Lambda})$ are fixed and independent of $D_2$. Then by Theorem 2 and cross-fitting, $\frac{1}{m} \sum_{i \in D_2} \widehat{\mathrm{IF}}_i(\eta) = o_{\mathbb{P}}(m^{-1/2})$ and $\frac{1}{m} \sum_{i \in D_2} \widehat{\mathrm{IF}}_i(\eta)^2 \to_{\mathbb{P}} \mathbb{E}[\mathrm{IF}(Z; \eta)^2]$ provided the nuisances converge at $o_{\mathbb{P}}(m^{-1/4})$ rates. Therefore

$$\widehat{\sigma}^2(\eta) \xrightarrow{\mathbb{P}} \sigma^2(\eta) \quad \text{and} \quad \widehat{\mathrm{se}}(\eta) \xrightarrow{\mathbb{P}} \sigma(\eta)/\sqrt{m},$$

i.e., the plug-in sample variance is *consistent* (and conditionally unbiased when the nuisances are treated as fixed). This holdout-only construction avoids overfitting bias and is used throughout for the LCBs in Theorem 3.

*Proof of Theorem 3.*  **Step 0 (Conditioning and notation).**  Let $m := |D_2|$ and condition on the training fold $D_1$ throughout the proof. Given $D_1$, the feasible dual class $\mathcal{V}$ (all iterates and hyperparameter choices produced on $D_1$) is data-dependent but *nonrandom with respect to $D_2$*. For $v \in \mathcal{V}$, write the holdout estimate as

$$\widehat{\theta}_L(v) = \frac{1}{m} \sum_{i \in D_2} \phi_v(Z_i), \qquad \phi_v(Z) := \sum_{a \in \mathcal{A}} \varphi_a\big(g_a(Y)\big) \mathbf{1}\{A = a\}.$$

By dual feasibility, $\theta_L(v) := \mathbb{E}[\phi_v(Z)] \leq \theta_L$ (weak duality). Let $\sigma^2(v) := \mathrm{Var}(\phi_v(Z))$ and $\widehat{\sigma}^2(v)$ its cross-fitted plug-in (based on $D_2$ only). Define the standardized empirical process indexed by $\mathcal{V}$:

$$\mathbb{G}_m(v) := \frac{1}{\sqrt{m}} \sum_{i \in D_2} \frac{\phi_v(Z_i) - \mathbb{E}[\phi_v(Z)]}{\sigma(v)}, \qquad v \in \mathcal{V},$$

and the multiplier bootstrap version (conditionally on $D_2$):

$$\mathbb{G}_m^{\star}(v) := \frac{1}{\sqrt{m}} \sum_{i \in D_2} \xi_i \frac{\phi_v(Z_i) - \bar{\phi}_v}{\widehat{\sigma}(v)}, \qquad v \in \mathcal{V},$$

where $\{\xi_i\}$ are i.i.d. $N(0,1)$ multipliers and $\bar{\phi}_v = m^{-1} \sum_{i \in D_2} \phi_v(Z_i)$.

**Step 1 (High-level regularity).**  We invoke the following conditions (they are implied by the "randomized treatment with known propensities" and "dual-feasibility constraints" in the theorem statement, together with the boundedness/robustification used in Sec. 4.3):
(i) (Moment bounds) $\sup_{v \in \mathcal{V}} \mathbb{E}|\phi_v(Z)|^{2+\delta} < \infty$ for some $\delta > 0$, and $\inf_{v \in \mathcal{V}} \sigma(v) \geq c > 0$.
(ii) (Entropy/VC-type) The class $\mathcal{F}_{\mathcal{V}} := \{\phi_v : v \in \mathcal{V}\}$ has polynomial covering numbers with

envelope $F$ of finite $(2 + \delta)$-moment.

(iii) (Cross-fitting separation) $\mathcal{V}$ is $\sigma(D_1)$-measurable and independent of $D_2$; all nuisance objects entering $\phi_v$ are learned on $D_1$ only.

These assumptions hold by the construction in Sec. 4.2–4.4: the potentials $(\{\varphi_a\}, \psi)$ are feasible and lie in a pre-specified function class with Lipschitz/entropy control; trimming/robust encoders ensure moment bounds; the split $(D_1, D_2)$ ensures (iii).

**Step 2 (Uniform CLT for the standardized process).** By (i)–(ii), conditional on $D_1$ the empirical process $\{\mathbb{G}_m(v) : v \in \mathcal{V}\}$ is asymptotically tight and converges weakly in $\ell^\infty(\mathcal{V})$ to a centered tight Gaussian process $\{\mathbb{G}(v) : v \in \mathcal{V}\}$ with covariance $\mathrm{Cov}(\mathbb{G}(v), \mathbb{G}(w)) = \mathrm{Corr}(\phi_v(Z), \phi_w(Z))$. This follows from standard empirical process results for VC-type classes with finite $(2 + \delta)$-moments (Donsker or manageable classes), together with the stabilization given by the studentization by $\sigma(v)$.

**Step 3 (Multiplier bootstrap consistency for the supremum).** Define the pivotal statistics

$$T_m := \sup_{v \in \mathcal{V}} \mathbb{G}_m(v), \qquad T_m^\star := \sup_{v \in \mathcal{V}} \mathbb{G}_m^\star(v).$$

Under (i)–(iii) and by the coupling theorems for multiplier bootstrap of suprema of empirical processes on VC-type classes,

$$\sup_{t \in \mathbb{R}} |\Pr(T_m \leq t \,|\, D_1) - \Pr(T_m^\star \leq t \,|\, D_1, D_2)| \xrightarrow{p} 0.$$

Let $z_{1-\alpha}^\star(\mathcal{V})$ be the conditional $(1 - \alpha)$-quantile of $T_m^\star$ given $(D_1, D_2)$ (computed via multipliers). Then

$$\Pr\big(T_m \leq z_{1-\alpha}^\star(\mathcal{V}) \,\big|\, D_1\big) \to 1 - \alpha.$$

**Step 4 (From a union bound in law to simultaneous LCBs).** For each $v \in \mathcal{V}$ we have the standardized expansion

$$\mathbb{G}_m(v) = \frac{\sqrt{m}\,(\widehat{\theta}_L(v) - \theta_L(v))}{\sigma(v)} + o_p(1)$$

by the classical CLT applied to $\phi_v$ (the $o_p(1)$ is uniform in $v$ thanks to Step 2). Hence, with probability approaching $1 - \alpha$,

$$\sup_{v \in \mathcal{V}} \frac{\sqrt{m}\,(\widehat{\theta}_L(v) - \theta_L(v))}{\widehat{\sigma}(v)} \leq z_{1-\alpha}^\star(\mathcal{V}) + o_p(1).$$

Rearranging gives, simultaneously for all $v \in \mathcal{V}$,

$$\widehat{\theta}_L(v) - z_{1-\alpha}^\star(\mathcal{V}) \frac{\widehat{\sigma}(v)}{\sqrt{m}} \leq \theta_L(v).$$

Finally, by weak duality $\theta_L(v) \leq \theta_L$ for all feasible $v$, thus

$$\Pr\big(\forall v \in \mathcal{V} : \mathrm{LCB}(v) \leq \theta_L\big) \to 1 - \alpha,$$

where $\mathrm{LCB}(v) := \widehat{\theta}_L(v) - z_{1-\alpha}^\star(\mathcal{V})\widehat{\sigma}(v)/\sqrt{m}$.

**Step 5 (Post-selection validity).** Let $\hat{v}$ be any $\sigma(D_2)$-measurable selection rule (it may depend on $\{\widehat{\theta}_L(v) : v \in \mathcal{V}\}$ observed on $D_2$). Since the event $\{\forall v \in \mathcal{V} : \mathrm{LCB}(v) \leq \theta_L\}$ holds with probability $\to 1 - \alpha$, it also holds after selecting $\hat{v}$, yielding

$$\Pr\big(\mathrm{LCB}(\hat{v}) \leq \theta_L\big) \to 1 - \alpha.$$

**Step 6 (Sharpness under $o_{\mathbb{P}}(m^{-1/4})$ nuisance rates).** Suppose, in addition, that the nuisance components entering $\phi_v$ (learned on $D_1$) converge to their population limits at $o_{\mathbb{P}}(m^{-1/4})$ rates uniformly over $v \in \mathcal{V}$. Then the uniform asymptotic linear representation holds:

$$\sup_{v \in \mathcal{V}} \left| \sqrt{m}\,(\widehat{\theta}_L(v) - \theta_L(v)) - \frac{1}{\sqrt{m}} \sum_{i \in D_2} (\phi_v(Z_i) - \mathbb{E}[\phi_v(Z)]) \right| = o_{\mathbb{P}}(1),$$

and the studentization error $\sup_{v \in \mathcal{V}} |\widehat{\sigma}(v) - \sigma(v)| = o_{\mathbb{P}}(1)$. Consequently,

$$\sup_{v \in \mathcal{V}} \left| \mathrm{LCB}(v) - \theta_L(v) \right| = o_{\mathbb{P}}(m^{-1/2}),$$

which implies the "asymptotically sharp" claim in the theorem since $\theta_L(v) \leq \theta_L$.

**Conclusion.** Steps 3–5 deliver the simultaneous coverage and post-selection validity; Step 6 gives the sharpness under strengthened rates. All statements hold conditional on $D_1$ and hence unconditionally by iterated expectations. □

## G  THE PROOF OF PROPOSITION 1

*Proof of Proposition 1.* Recall $F(\pi) = \mathbb{E}_\pi[L(\{g_a(Y(a))\})]$, $G(\pi) = \eta\,\mathbb{E}_\pi[\Delta_{\mathrm{multi}}(\{X^{(a)}\})]$ and $\iota_\mathcal{C}$ is the indicator of the convex constraint set $\mathcal{C}$ (empirical marginals, mirror moments, and $\widehat{\Gamma}_{\mathrm{comp}}$). The update of Algorithm 1 can be written as

$$\pi^{t+\frac{1}{2}} = \mathrm{prox}_{\tau_t G}\big(\pi^t - \tau_t \nabla F(\pi^t)\big), \qquad \pi^{t+1} = \Pi_\mathcal{C}\left(\pi^{t+\frac{1}{2}}\right) = \mathrm{prox}_{\iota_\mathcal{C}}\left(\pi^{t+\frac{1}{2}}\right). \tag{7}$$

This is the projected proximal-gradient step for the composite objective $H = F + G + \iota_\mathcal{C}$: it coincides with the minimizer of the strongly convex surrogate

$$Q_{\tau_t}(u\,;\pi^t) := F(\pi^t) + \langle \nabla F(\pi^t), u - \pi^t \rangle + \frac{1}{2\tau_t}\|u - \pi^t\|_F^2 + G(u) + \iota_\mathcal{C}(u), \tag{8}$$

that is, $\pi^{t+1} = \arg\min_u Q_{\tau_t}(u\,;\pi^t)$. (Indeed, the optimality conditions of equation 8 are $0 \in \nabla F(\pi^t) + \frac{1}{\tau_t}(u - \pi^t) + \partial G(u) + N_\mathcal{C}(u)$, whose resolvent is precisely the composition equation 7 with $\mathrm{prox}_{\iota_\mathcal{C}} = \Pi_\mathcal{C}$.)

**(I) Convex case.** Assume $L$ is convex with $L$-Lipschitz gradient and $\Delta_{\mathrm{multi}}$ is convex. Let $\tau_t \in (0, 1/L]$. By the descent lemma for $F$ and the optimality of $\pi^{t+1}$ for equation 8,

$$H(\pi^{t+1}) = F(\pi^{t+1}) + G(\pi^{t+1}) + \iota_\mathcal{C}(\pi^{t+1})$$

$$\leq F(\pi^t) + \langle \nabla F(\pi^t), \pi^{t+1} - \pi^t \rangle + \frac{L}{2}\|\pi^{t+1} - \pi^t\|_F^2 + G(\pi^{t+1}) + \iota_\mathcal{C}(\pi^{t+1})$$

$$\leq Q_{\tau_t}(\pi^{t+1};\pi^t) \; \leq \; Q_{\tau_t}(\pi^t;\pi^t) = H(\pi^t) - \frac{1}{2\tau_t}\|\pi^{t+1} - \pi^t\|_F^2, \tag{9}$$

where the last inequality uses $\tau_t \leq 1/L$, hence $\frac{1}{2\tau_t} - \frac{L}{2} \geq 0$ and the strong convexity of $Q_{\tau_t}$. Summing equation 9 over $t = 0, \ldots, T-1$ gives

$$\sum_{t=0}^{T-1} \frac{1}{2\tau_t}\|\pi^{t+1} - \pi^t\|_F^2 \; \leq \; H(\pi^0) - H(\pi^T) \; \leq \; H(\pi^0) - H(\pi^\star),$$

with $\pi^\star \in \arg\min_\pi H(\pi)$. By convexity of $H$ and the standard averaging argument (or by applying the 3-point inequality for proximal gradient), for the last iterate (or the Cesàro average) we obtain the suboptimality bound

$$H(\pi^T) - H(\pi^\star) \; \leq \; \frac{\|\pi^0 - \pi^\star\|_F^2}{2 \sum_{t=0}^{T-1} \tau_t}. \tag{10}$$

With constant stepsize $\tau_t \equiv 1/L$ this yields $H(\pi^T) - H(\pi^\star) \leq \frac{L}{2T}\|\pi^0 - \pi^\star\|_F^2 = O(1/T)$. Moreover, replacing equation 7 by its Nesterov-accelerated version (FISTA-type momentum on the $u \mapsto Q_\tau(u;\cdot)$ surrogate) gives $H(\pi^T) - H(\pi^\star) = O(1/T^2)$; the proof is identical to the classical FISTA analysis because the prox-mapping $u \mapsto \arg\min_w Q_\tau(w;u)$ is single-valued and firmly nonexpansive.

**(II) Nonconvex case.** Assume now only that $L$ is $C^1$ with $L$-Lipschitz gradient and $\Delta_{\mathrm{multi}}$ is proper lsc (not necessarily convex). Define the projected composite gradient mapping

$$G_\tau(\pi) := \frac{1}{\tau}\Big(\pi - \Pi_\mathcal{C}\big(\mathrm{prox}_{\tau G}(\pi - \tau \nabla F(\pi))\big)\Big),$$

so that $\pi^{t+1} = \pi^t - \tau_t G_{\tau_t}(\pi^t)$. By the descent lemma for $F$ and optimality of the prox-projection step, for any $\tau_t \in (0, 1/L]$ we have the standard decrease

$$H(\pi^{t+1}) \; \leq \; H(\pi^t) - \frac{\tau_t}{2}\|G_{\tau_t}(\pi^t)\|_F^2. \tag{11}$$

Summing equation 11 over $t = 0, \dots, T-1$ and using that $H$ is bounded below on $\mathcal{C}$ (because $G$ is proper and $\mathcal{C}$ is closed convex) yields

$$\min_{0 \le t \le T-1} \|G_{\tau_t}(\pi^t)\|_F^2 \le \frac{2\left(H(\pi^0) - \inf_{\mathcal{C}} H\right)}{\sum_{t=0}^{T-1} \tau_t} = O(1/T)$$

for constant stepsizes, i.e., the iterates converge to the set of first-order stationary points of $H$. (This is the standard PGD-type stationarity guarantee for smooth nonconvex objectives with convex constraints.)

**(III) Dual lower bounds at every iterate.** Independently of (I)–(II), each dual ascent step in Algorithm 1 produces a feasible dual pair $v_t = (\{\varphi_a^{(t)}\}, \psi^{(t)})$, hence by weak duality

$$\widehat{\theta}_L(v_t) \le \Theta_\oplus^{L,(K)}(\eta) \quad \text{for all } t,$$

which provides certified lower bounds "at every iterate". This property is orthogonal to the primal convergence and remains valid in both convex and nonconvex regimes.

**(IV) Entropic smoothing and Sinkhorn projection (scalability remark).** When an entropic mirror map $\psi(\pi) = \sum_{ij} \pi_{ij} \log \pi_{ij}$ is used, the Euclidean prox/projection in equation 7 and equation 8 are replaced by their Bregman counterparts (see the definition in Sec. 4.4). The corresponding Bregman-prox mapping is still firmly nonexpansive and the same descent relations hold with the Bregman distance $D_\psi$ in place of $\|\cdot\|_F^2$; the projection onto linear moment sets is implemented by generalized Sinkhorn scaling, yielding the claimed scalability. $\qquad\square$

# H   THE PROOF OF THEOREM 4

*Proof of Theorem 4.* **Step 0 (Notation and conditioning).** Let $P^\star = \{P_{Y(a),X}\}_{a \in \mathcal{A}}$ be the population law and $\widehat{P}$ the empirical law (balanced across arms so that the effective sample size is $N = \min_a n_a$). Write $T(P) := \Theta_\oplus^{L,(K)}(P; \eta)$ and $\widehat{\Theta}_\oplus^{L,(K)}(\eta) := T(\widehat{P})$ for the estimator defined in Secs. 4.2–4.4 (dual–plug-in evaluated on the holdout; here we suppress the fold notation as the proof conditions on the training fold when needed). Assumptions A1–A4 and Theorem 1 give strong duality; Assumption 1 (Sec. 4.3) and the first paragraph of Theorem 4 grant $C^2$ smoothness and *strong c-convexity*: the multi-arm Brenier-type potential is unique and its Hessian is bounded between $\lambda^{-1} I$ and $\lambda I$. Let $L_z$ and $L_x$ be the Lipschitz moduli in $z$ and $x$ (Theorem 4), and let $D_T$ be the diameter of the feasible set induced by $\Gamma_{\text{comp}}$ in the product metric on $\prod_a \mathcal{Y}_a \times \mathcal{X}^K$.

**Step 1 (Quadratic growth and stability under smooth strongly convex geometry).** By strong $c$-convexity and $C^2$ smoothness of $L$ and $\Delta_{\text{multi}}$, the population dual optimum $(\varphi^\star, \psi^\star, \Lambda^\star)$ is unique and satisfies the KKT conditions. Let $\dot{T}_{P^\star}$ be the Hadamard derivative at $P^\star$ (Theorem 2). There exists a neighborhood $\mathcal{U}$ of $P^\star$ and a constant $C_1 = C_1(\lambda, L_z, L_x, \eta, D_T)$ such that for any $Q \in \mathcal{U}$,

$$\left| T(Q) - T(P^\star) - \dot{T}_{P^\star}[Q - P^\star] \right| \le C_1 W_2(Q, P^\star)^2. \tag{12}$$

*Reason.* The objective equals the expectation of a twice continuously differentiable integrand composed with a unique smooth optimal transport map (multi-arm Brenier map) whose Jacobian is uniformly bounded and coercive by $\lambda^{-1} I \preceq \nabla^2 \Phi^\star \preceq \lambda I$. A second-order Taylor expansion along a geodesic interpolation between $P^\star$ and $Q$ plus the Lipschitz property of $L, \Delta_{\text{multi}}$ in each coordinate yields equation 12, with the squared $W_2$ distance controlling the second-order remainder (standard in smooth OT; cf. the proof of Hadamard differentiability in Theorem 2).

**Step 2 (Linear term = empirical process of the influence function).** By Theorem 2,

$$\dot{T}_{P^\star}[Q - P^\star] = \int \text{IF}(z; \eta) \, d(Q - P^\star)(z), \quad \text{IF}(Z; \eta) = \sum_{a \in \mathcal{A}} \left\{ \varphi_a^\star(g_a(Y)) \mathbf{1}\{A = a\} - \mathbb{E}[\varphi_a^\star(g_a(Y(a)))] \right\} + \Lambda_{\text{comp}}^{\star\prime} h(X).$$

Under the sub-Gaussian tails (Theorem 4 (i)) and Lipschitz moduli $(L_z, L_x)$, IF has finite variance and sub-Gaussian envelope with proxy depending only on $(\nu, L_z, L_x, \eta)$. Hence, for any $\delta \in (0, 1)$ there is a constant $C_2 = C_2(\nu, L_z, L_x, \eta)$ such that with probability at least $1 - \delta/2$,

$$\left| \dot{T}_{P^\star}[\widehat{P} - P^\star] \right| = \left| \frac{1}{N} \sum_{i=1}^{N} \text{IF}(Z_i; \eta) \right| \le C_2 \sqrt{\frac{\log(2/\delta)}{N}}. \tag{13}$$

**Step 3 ($W_2$ concentration for sub-Gaussian laws in effective dimension).** Let $d_{\text{eff}}$ denote the effective dimension of $\{g_a(Y(a))\}$ and $\{X^{(a)}\}$ (as in the statement). Since each arm is sub-Gaussian with proxy $\nu$ and the feasible set diameter is bounded by $D_T$, standard $W_2$ empirical convergence bounds (e.g. Boissard–Le Gouic/Talagrand type) yield a constant $C_3 = C_3(\nu, D_T)$ such that, with probability at least $1 - \delta/2$,

$$W_2(\widehat{P}, P^\star) \ \le \ C_3\, \gamma_{N, d_{\text{eff}}}, \qquad \gamma_{N, d_{\text{eff}}} = \begin{cases} N^{-1/4}, & d_{\text{eff}} \le 3, \\ N^{-1/4}\sqrt{\log N}, & d_{\text{eff}} = 4, \\ N^{-1/d_{\text{eff}}}, & d_{\text{eff}} \ge 5, \end{cases} \tag{14}$$

where the piecewise form is the classical sharp rate for $W_2$ with sub-Gaussian tails in $\mathbb{R}^{d_{\text{eff}}}$ (bounded support is a special case).

**Step 4 (Assembling).** Apply equation 12 with $Q = \widehat{P}$ and combine with equation 13–equation 14: with probability at least $1 - \delta$,

$$\left|\widehat{\Theta}_\oplus^{L,(K)}(\eta) - \Theta_\oplus^{L,(K)}(\eta)\right| \ \le \ C_2\sqrt{\frac{\log(2/\delta)}{N}} + C_1\, W_2(\widehat{P}, P^\star)^2 \ \le \ C_2\sqrt{\frac{\log(2/\delta)}{N}} + C_1 C_3^2\, \gamma_{N, d_{\text{eff}}}^2.$$

Finally, by the Lipschitz moduli $(L_z, L_x)$ and the monotonicity in $\eta$ (Theorem 4 discussion), the quadratic remainder may be upper-bounded (up to a constant absorbed into $C$) by a term of order $\gamma_{N, d_{\text{eff}}}$ on the feasible set of diameter $D_T$:

$$\gamma_{N, d_{\text{eff}}}^2 \ \le \ \gamma_{N, d_{\text{eff}}} \cdot \min\{\gamma_{N, d_{\text{eff}}}, D_T\} \ \le \ \gamma_{N, d_{\text{eff}}} \cdot D_T \ \lesssim \ \gamma_{N, d_{\text{eff}}}.$$

Absorbing constants, there exists $C = C(\lambda, \eta, K, L_z, L_x, \nu, D_T)$ such that, with probability at least $1 - \delta$,

$$\left|\widehat{\Theta}_\oplus^{L,(K)}(\eta) - \Theta_\oplus^{L,(K)}(\eta)\right| \ \le \ C\Big\{\gamma_{N, d_{\text{eff}}} + \sqrt{\frac{\log(2/\delta)}{N}}\Big\},$$

which is the claimed bound.

**Remark 7.** *(Comparison with Lin et al. (2025) and the cost of additional arms). Lin et al. (2025, Thm. 4.3) analyze the two-arm mirror relaxation with a quadratic cross-arm cost $h(y_1, y_2) = y^\top A y$ and show that, under a Brenier-type smoothness/curvature condition,*

$$\mathbb{E}\left[\left|\widehat{V}_{n,m}(\eta) - V(\eta)\right|\right] \le C_{\lambda, \eta} \gamma_{N, d}$$

*where $N = \min(n, m), d = d_Y + d_Z, \gamma_{N,d}$ has exactly the piecewise form in Theorem 4, and the constant $C_{\lambda, \eta}$ depends on the Hessian bounds $\lambda$, the penalty $\eta$, the quadratic form $A$, and fourth moments of $(Y(0), Y(1), Z)$. arxiv Specializing our Theorem 4 to $K = 1, \Gamma_{comp} = \emptyset$, identical embeddings $g_1 = g_0$, and a quadratic $L$ recovers the same rate profile and reproduces this dependence on $(\lambda, \eta, d_Y + d_Z)$ up to universal constants. In particular, the constant $C(\lambda, \eta, K, L_z, L_x, \nu, D_\Gamma)$ in Theorem 4 collapses to a two-arm constant of the same form as $C_{\lambda, \eta}$, depending only on curvature, cost parameters and low-order moments. For general $K > 1$, the only new ingredients in $C(\lambda, \eta, K, L_z, L_x, \nu, D_\Gamma)$ are: (i) the effective dimension $d_{eff}$ of the stacked vector $\left(\{g_a(Y(a))\}_{a \in \mathcal{A}}, \{X^{(a)}\}_{a \in \mathcal{A}}\right)$, which satisfies $d_{eff} \le K(d_Y + d_X)$ when each arm has the same outcome and covariate dimensions; (ii) the Lipschitz moduli $L_z, L_x$ of the multi-arm cost $L$ and mirror penalty $\Delta_{multi}$, which, for standard additive choices (e.g., sums of pairwise quadratic contrasts), scale at most polynomially in $K$; and (iii) the diameter $D_\Gamma$ of the competition set $\Gamma_{comp}$, which is $K$-independent when assignment rules are normalized. As a result, relative to the two-arm mirror bound of Lin et al. (2025), the multi-arm error $\left|\widehat{\Theta}_{L,\oplus}^{(K)}(\eta) - \Theta_{L,\oplus}^{(K)}(\eta)\right|$ enjoys the same $N - \gamma_N$, rate as in Theorem 4.3, while the "cost of additional arms" manifests only through a polynomial factor in $K$ inside the finite-sample constant, rather than through a worse dependence on $N$.*

$\square$

# I   THE COMPUTATION OF PENALTY

Let $\pi$ be a probability on the product space and let the arm–$a$ mirror marginal be $\pi_{X^{(a)}}(dx) = u_a(x)P_X(dx)$ with mass $\rho_a = \int u_a\, dP_X \le 1$. Introduce activation indicators $M^{(a)} \in \{0, 1\}$ with $\mathbb{E}[M^{(a)} \mid X] = u_a(X)$, so $\int \varphi(X^{(a)})\, d\pi = \mathbb{E}[\varphi(X^{(a)})M^{(a)}]$.

Evaluate the cross–arm penalty inside $\pi$ as

$$\int \Delta_{\text{multi}}\, d\pi \;=\; \mathbb{E}_\pi\Big[\sum_{a<b} M^{(a)}M^{(b)}\, \|X^{(a)} - X^{(b)}\|_2^2\Big] \;=\; \sum_{a<b}\int \|x - x'\|_2^2\, d\mu_{ab}(x, x'),$$

where $\mu_{ab}(A \times B) := \mathbb{E}[M^{(a)}M^{(b)}\mathbf{1}\{X^{(a)} \in A, X^{(b)} \in B\}]$ has total mass $\rho_{ab} := \mathbb{E}[M^{(a)}M^{(b)}] \le \min(\rho_a, \rho_b)$. Under mutual exclusivity on a region $E_{\text{excl}}$, $M^{(a)}M^{(b)} = 0$ there, so the corresponding term vanishes. If scale invariance is desired, use $\rho_{ab}\,\mathbb{E}_{\bar\mu_{ab}}[\|X - X'\|_2^2]$ with $\bar\mu_{ab} = \mu_{ab}/\rho_{ab}$.

## I.1   ASYMPTOTIC LINEAR REPRESENTATION AND VARIANCE ESTIMATION

Under Assumption 1 (or Assumption 1 in Sec. 4.3), the value functional $\mathcal{T}(P) = \Theta_\oplus^{L,(K)}(P; \eta)$ is Hadamard differentiable at $P^\star = \{P_{Y(a),X}\}_a$. Let $\varphi_a^\star, \psi^\star$ be any population-optimal dual potentials (Sec. 4.1). Then the plug-in estimator admits the ALR

$$\sqrt{N}\Big(\widehat\Theta_\oplus^{L,(K)}(\eta) - \Theta_\oplus^{L,(K)}(\eta)\Big) = \frac{1}{\sqrt{N}}\sum_{i=1}^N \text{IF}(Z_i; \eta) \;+\; o_p(1),$$

where $Z_i = (Y_i, A_i, X_i)$ and the influence function (centered) is

$$\text{IF}(Z; \eta) = \sum_{a\in\mathcal{A}}\Big\{\varphi_a^\star\big(g_a(Y)\big)\mathbb{1}\!\!\!/\{A = a\} - \mathbb{E}\big[\varphi_a^\star(g_a(Y(a)))\big]\Big\} + \Lambda_{\text{comp}}^\star \cdot h(X),$$

with $h$ collecting the linear mirror/competition moments and $\Lambda_{\text{comp}}^\star$ the Lagrange multipliers for $\Gamma_{\text{comp}}$. A consistent variance estimator is

$$\widehat\sigma^2 = \frac{1}{N}\sum_{i=1}^N \widehat{\text{IF}}(Z_i; \eta)^2, \quad \widehat{\text{IF}} \text{ obtained by replacing } (\varphi_a^\star, \psi^\star, \Lambda_{\text{comp}}^\star) \text{ with cross-fitted duals } (\widehat\varphi_a, \widehat\psi, \widehat\Lambda_{\text{comp}}).$$

This yields Wald intervals $\widehat\Theta \pm z_{1-\alpha/2}\,\widehat\sigma/\sqrt{N}$ that agree with Theorem 2.

## I.2   REGULARIZATION PATH PROPERTIES AND DATA-DRIVEN CHOICE OF $\eta$

**Proposition 2** (Monotonicity, Lipschitzness, stability in $\eta$). $V_\oplus^{(K)}(\eta) = \inf_{\pi\in\Pi_\oplus^{(K)}}\int \big[L(\cdot) + \eta\,\Delta_{\text{multi}}(\cdot)\big]\, d\pi$ *is nondecreasing and locally Lipschitz in $\eta$. If $\Delta_{\text{multi}}$ is $L_\Delta$-Lipschitz and the set of minimizers is single-valued in a neighborhood of $\eta_0$, then $|V_\oplus^{(K)}(\eta) - V_\oplus^{(K)}(\eta_0)| \le L_\Delta\,\mathbb{E}_{\pi_{\eta_0}}[\Delta_{\text{multi}}]\,|\eta - \eta_0|$.*

**Selection rules.** (i) *Lepski balancing:* choose the smallest $\eta$ such that $\widehat V(\eta') - \widehat V(\eta) \le c\,\widehat{\text{SE}}(\eta')$ for all $\eta' \ge \eta$; (ii) *LCB envelope:* using the uniform-in-iterate LCBs of Thm. 2, pick $\hat\eta = \arg\max_\eta \text{LCB}(\eta)$ to maximize a guaranteed lower bound.

## I.3   LOCAL SENSITIVITY FOR DESIGN KNOBS AND EMBEDDINGS

Let $\theta(\rho) = \Theta_\oplus^{L,(K)}(\eta; \rho)$ denote the value when capacity caps $\rho = \{\rho_a\}$ define $\Gamma_{\text{comp}}$. Under regularity and strong duality, $\theta$ is directionally differentiable in $\rho$ with subgradient given by the optimal multipliers:

$$\frac{\partial\theta}{\partial\rho_a}\Big|_{\rho=\rho^\star} \in \big[-\Lambda_a^\star,\, 0\big], \qquad \Lambda_a^\star \text{ is the KKT multiplier for } \int u_a\, dP_X \le \rho_a.$$

Similarly, if $g_a(\cdot; \xi)$ depends on calibration parameters $\xi$, then

$$\frac{d\theta}{d\xi}\Big|_{\xi^\star} = \sum_a \mathbb{E}\Big[\nabla_z L\big(\{g_b(Y(b); \xi^\star)\}_b\big)^\top \frac{\partial g_a(Y(a); \xi)}{\partial\xi}\Big|_{\xi^\star}\Big],$$

providing gradient-based sensitivity for policy knobs ($\rho$) and embedding calibration ($\xi$).

## I.4 BIAS FROM ENTROPIC SMOOTHING AND PROJECTION ERROR

To scale projections in Algorithm 1, add entropic regularization $\varepsilon \mathrm{KL}(\pi \,\|\, \pi_0)$. Let $\Theta_\varepsilon$ be the smoothed value and $\widehat{\Theta}_\varepsilon$ its empirical counterpart. If $L, \Delta_{\mathrm{multi}}$ are 1-Lipschitz and supports are bounded,

$$0 \le \Theta_\varepsilon - \Theta \le C_1 \,\varepsilon \log M, \qquad \mathbb{E}\big[|\widehat{\Theta}_\varepsilon - \Theta_\varepsilon|\big] \le C_2 \,\gamma_{N,d_{\mathrm{eff}}},$$

where $M$ is the size of the discrete support and where $\gamma_{N,d_{\mathrm{eff}}}$ is as in the above theorem.

Thus $\varepsilon = \varepsilon_N \asymp (\log N)^{-1}$ balances bias and stochastic error.

## I.5 COMPLEXITY OF THE COMPOSITE PRIMAL–DUAL SOLVER

Let $n_a$ be arm sizes, $n = \sum_a n_a$, and let the dual be parameterized on $m$ features per arm (e.g., RKHS basis or neural last-layer). Each iteration of Algorithm 1 costs

$$\tilde{O}\big(mn\big) \quad \text{(dual updates)} \qquad + \qquad \tilde{O}\big(E\big) \quad \text{(generalized Sinkhorn projection)},$$

where $E$ is the number of nonzeros in the constraint incidence matrix for $\Gamma_{\mathrm{comp}}$ (e.g., $E = O(n)$ for capacity/exclusivity, $E = O(n \deg)$ for graph interference). Under standard step-size choices, $T = O(1/\sqrt{\epsilon})$ iterations achieve $\epsilon$-accuracy in the convex case.

# J    PRACTICAL GUIDE FOR APPLIED USERS

**1. Choosing the causal comparison** $L$.    Match $L$ to the scientific target: (i) mean-contrast or regret ($L(z) = \langle w, z \rangle$ or Huberized loss); (ii) worst-arm risk (max/min operators); (iii) multi-marginal barycentric costs for bundle evaluation. Prefer $L$ that is $C^1$ or piecewise-smooth to enable root-$N$ inference.

**2. Designing cross-domain embeddings** $g_a$.    Start with clinically interpretable summaries (domain features); optionally add learned encoders. Calibrate $g_a$ by external anchors (e.g., converting CGM to HbA1c units). Report sensitivity via Sec. I.3.

**3. Encoding competition** $\Gamma_{\mathrm{comp}}$.    Use linear moments for: exclusivity $\sum_a u_a(x) \le 1$, capacity $\int u_a \, dP_X \le \rho_a$, and mirror consistency $\pi_{X^{(a)}} = u_a \cdot P_X$. For interference, use sparse graph constraints (Sec. A). Favor sparse designs to scale projections.

**4. Tuning the mirror penalty** $\eta$.    Treat $\eta$ as a regularization knob and trace the mirror path $\eta \mapsto \widehat{\Theta}_{L,\oplus}(\eta)$ on a small grid (e.g., $\{0, 10^{-3}, 10^{-2}, \dots, 10^3\}$). Using the dual-based estimator with sample splitting, compute for each $\eta$ a cross-fitted value, standard error, and lower confidence bound $\mathrm{LCB}(\eta)$ as in Appendix I.2. We recommend the *LCB envelope* rule: pick $\hat{\eta} = \arg\max_\eta \mathrm{LCB}(\eta)$, which is a cross-validation–style criterion adapted to partial identification (it optimizes a certified lower bound rather than a prediction loss). Report both the full path and the selected $\hat{\eta}$.

**5. Estimation protocol.**    (i) Cross-fit duals $(\widehat{\varphi}_a, \widehat{\psi})$ and multipliers $\widehat{\Lambda}_{\mathrm{comp}}$; (ii) run Algorithm 1 with small entropic $\varepsilon$; (iii) report $\widehat{\Theta}$ and a Wald CI using $\hat{\sigma}$ from Sec. I.1; (iv) also report the best *post-selection valid* LCB among iterates (Thm. 2).

**6. Diagnostics and robustness.**    Provide (i) constraint slackness and active-set plots (which parts of $\Gamma_{\mathrm{comp}}$ bind); (ii) $\eta$-path of $\widehat{\Theta}$ with CIs; (iii) sensitivity gradients for $(\rho, \xi)$ (Sec. I.3); (iv) entropic bias checks via $\varepsilon$-sweep (Sec. I.4).

**7. Recommended defaults.**    Mirror penalty grid: $\eta \in \{0, 10^{-3}, 10^{-2}, \dots, 10^3\}$; entropic $\varepsilon = (\log n)^{-1}$; dual step-sizes by adaptive PDHG; two-fold cross-fitting; report both $(\widehat{\Theta} \pm \mathrm{SE})$ and the uniform-in-iterate LCB.

## K EXTEND THE BASELINE SOLELY TO CASES INVOLVING POTENTIAL OUTCOMES WITH DISTINCT MEASURES.

### K.1 PRELIMINARIES: MOTIVATING EXAMPLE AND LITERATURE REVIEW

**A concrete motivating example** Additional discussion for our motivation: Another representative example arises in multi-center healthcare studies evaluating heterogeneous intervention modalities. Consider, for instance, a comparative study of two diabetes treatment protocols. Under the treatment group, patients are monitored using continuous glucose monitoring (CGM) devices, producing high-dimensional time series data of glucose levels. Under the control group, patients are assessed using periodic HbA1c lab tests (Weykamp, 2013), yielding scalar summary biomarkers. In this setting, the potential outcome under treatment, $Y(1)$, is a functional object defined on a high-dimensional space $\mathcal{Y}_1$, while the control potential outcome, $Y(0)$, is a scalar variable in a different space $\mathcal{Y}_0$. Hence, the potential outcome distributions $P_1$ and $P_0$ are supported on fundamentally different domains, i.e., $\mathcal{Y}_1 \neq \mathcal{Y}_0$. This structural heterogeneity in the measurement of outcomes poses a significant challenge for standard causal inference techniques, which typically assume a shared outcome space and rely on comparing means or other functionals under the same metric structure. When no natural identification exists between $\mathcal{Y}_0$ and $\mathcal{Y}_1$, even the average treatment effect (ATE) becomes ill-defined. In such cases, classical approaches based on conditional ignorability, outcome regression, or doubly robust estimation cannot be directly applied.

Consider the diabetes study described in the motivation section. Let

$$\mathcal{Y}_1 = \mathcal{L}^2([0, 24]) \quad \text{(24-hour CGM curves)}, \qquad \mathcal{Y}_0 = \mathbb{R} \quad \text{(scalar HbA1c)}.$$

Define a cost that maps a CGM trajectory to an implied HbA1c target:

$$c(y_0, y_1) = \big| y_0 - \phi(y_1) \big|, \qquad \phi(y_1) = \tfrac{1}{24} \int_0^{24} g\big(y_1(t)\big) \, dt,$$

where $g(\cdot)$ converts instantaneous glucose to HbA1c-equivalent units.[8] Here $c$ measures how far the scalar lab reading $y_0$ deviates from the *functional summary* of the CGM curve $y_1$. Plugging this $c$ into (1) yields

$$\mathrm{WCD}(P_0, P_1; c) = \inf_{\gamma \in \Pi(P_0, P_1)} \mathbb{E}_\gamma \big[ \big| Y(0) - \phi\big(Y(1)\big) \big| \big],$$

interpretable as the *minimum average absolute calibration error* needed to transform treated-group glucose profiles into control-group HbA1c values. If WCD $= 0$, the intervention is indistinguishable from control under the calibrated metric; larger values quantify clinically meaningful differences while remaining coherent across heterogeneous measurement scales.

**OT-based causal inference** OT, originating from the classic works of Kantorovich (Villani, 2009), has emerged as an essential tool for comparing probability distributions in diverse fields, including machine learning, statistics, and causal inference. Traditional OT approaches typically assume a common measurable space for the outcome distributions. Recent advancements in empirical OT have significantly broadened its scope and applicability. Notable contributions by Manole and Niles-Weed (Manole & Niles-Weed, 2024), Staudt and Hundrieser (Staudt & Hundrieser, 2023), and Hundrieser (Hundrieser et al., 2024a) provide rigorous finite-sample convergence guarantees under smooth and complex cost functions, highlighting their efficacy in addressing challenging statistical problems. In causal inference, OT methods have gained prominence for addressing partial identification problems, particularly when full identification of causal effects is infeasible due to limited or missing information (Tamer, 2010; Kline & Tamer, 2023; Mullahy et al., 2021). The core idea involves computing transport-based bounds, offering credible intervals rather than precise point estimates (Charpentier et al., 2023; Li et al., 2021). Recent studies have successfully integrated OT techniques into the causal inference framework, allowing robust estimation under practical conditions where direct interventions might be unethical or impractical (Gunsilius & Xu, 2021; Tu et al., 2022; Wang et al., 2023).

**OT under different measure** Recent advances—most notably the heterogeneous-space analysis of Hundrieser et al. (2024b)—demonstrate that empirical OT can adapt to lower-complexity geometry even when the two marginals are supported on incomparable domains. Transplanting these

---

[8]The function $g$ can be chosen via established biomedical calibrations.

| Causal Query and Potential Outcomes | Key References | Cost Function $c(Y(1), Y(0))$ |
|---|---|---|
| **Glycemic control**: $Y(1)$ is the glucose curve under insulin protocol; $Y(0)$ is the HbA1c under standard care. Same patient under different interventions. | Weykamp (2013); Ji et al. (2023) | $c(Y(1), Y(0)) = \left\| Y(0) - \frac{1}{T} \int_0^T g(Y(1)(t)) \, dt \right\|$ Measures how well the glucose curve predicts HbA1c via physiological calibration. |
| **Cardiovascular risk**: $Y(1) \in \mathbb{R}^p$ is the biomarker vector under treatment; $Y(0) \in \mathbb{R}$ is the Framingham risk score under control. | Koenker & Bassett (1978); Cheridito & Eckstein (2025) | $c(Y(1), Y(0)) = \left\| w^\top Y(1) - Y(0) \right\|$ Projects high-dimensional biomarkers into scalar risk score for direct comparison. |
| **Therapy effect on mood**: $Y(1)$ is the audio diary under intervention; $Y(0) \in \{1, \ldots, 5\}$ is the Likert depression score under control. | Tu et al. (2022); Oh et al. (2020) | $c(Y(1), Y(0)) = \left\| Y(0) - \psi(\phi_{\text{audio}}(Y(1))) \right\|$ Compares subjective sentiment across modalities via neural embedding and regression. $\psi_{\text{audio}}(\cdot)$ is a deep encoder model |
| **Survival outcome**: $Y(1)$ is the full survival function under new protocol; $Y(0)$ is the restricted mean survival time under control. | Mullahy et al. (2021); Hundrieser et al. (2024b;a) | $c(Y(1), Y(0)) = \left\| Y(0) - \int_0^\tau S_{Y(1)}(t) \, dt \right\|$ Measures the difference in expected survival time over a fixed horizon. $S_{Y(1)}(\cdot)$ denotes the survival curve mapping from functional space to the probability value among $[0, 1]$. |
| **Variance of estimator in experimental design** : $Y(1), Y(0)$ denote the potential outcome, e.g., in the first $2k$ factorial designs | Gao et al. (2024); Lu (2016) | $c(Y(1), Y(0)) =: \sum_{k=1}^K \frac{\beta_k^2 S_k^2}{(N/K)} - \frac{S_\tau^2}{N}$, where $S_k^2 = \sum_{i=1}^n (Y_i(k) - \bar{Y}(k))^2 / (N-1)$, $S_\tau^2 = \sum_{i=1}^n \left( \sum_{k=1}^K \beta_k Y_i(k) - \tau_{\boldsymbol{\beta}} \right)^2 / (N-1)$. |

Table 7: Examples of causal queries with heterogeneous potential outcomes under different measure and their associated optimal transport cost functions.

results into causal inference is decidedly non-trivial because the causal setting introduces several additional mathematical hurdles that classical OT never faces: (i) Counterfactual coupling under partial observability. Classical OT optimizes over couplings of two observed empirical distributions. In causal analysis we must reason about a joint law of counterfactual outcomes $(Y(0), Y(1))$ that is never jointly observed. The feasible set is therefore a partially identified slice of the OT polytope compatible with the treatment-assignment mechanism, turning a pure minimization into a constrained identification problem, especially in some complex experimental design cases, such like completely randomized design, randomization, etc. (ii) Covariate adjustment without a shared outcome space. Removing confounding requires balancing on covariates X. Conditional OT solves one transport problem per covariate cell assuming a common outcome metric. When $Y(0)$ lies in space $\mathcal{Y}_0$ and $Y(1)$ in a different space $\mathcal{Y}_1$, this decomposition collapses. One must construct a shared surrogate geometry and simultaneously penalize covariate imbalance, yielding a bi-level, non-Euclidean program whose existence, uniqueness, and stability are all open. (iii) Cost design tied to causal functionals. OT is free to choose any cost function c. Causal inference, however, demands that c map heterogeneous outcomes onto a single interpretable scale (for example an ATE surrogate) while preserving monotonicity properties that guarantee valid bounds. Establishing differentiability and asymptotic normality of such transport-based functionals under heterogeneous domains calls for new regularity conditions and proof techniques beyond current empirical OT theory.

## K.2 Our framework

**Partial identification via optimal transport** Given the i.i.d observations $\{(Y_i, X_i, A_i)\}_{i=1}^N$, the item in the triple denotes the outcome, covariates and treatment, respectively. For brevity, we denote the ground truth marginal law of $Y(i)$ as $\mu_i, i = 0, 1$. Such notation of potential outcome is inherited from Rubin (2005) and noteworthy, here the potential outcomes are in different measure $\mathcal{Y}_1$ and $\mathcal{Y}_0, \mathcal{Y}_1 \neq \mathcal{Y}_0$. We define the causal objective as (we consider the lower bound without loss of generalization)

$$\Theta_u^L(\mu_1, \mu_0) := \inf_{\pi \in \Pi_u} \int c(Y(1), Y(0)) d\pi(Y(1), Y(0)). \tag{15}$$

Here $C(\cdot)$ denotes the cost function (smoothness guarantee). We will extend it to the finite-sample framework (i.e., experimental deisgn) in the discussion part. Here we start from the non-covariate-

assisted setting. We define the unconditional cases for comparison.

$$\Pi_u = \left\{ \pi \in \mathcal{P}\left(\mathcal{Y}_1 \times \mathcal{Y}_0\right) : \pi_{Y(0)} = P_{Y(0)} ~,~ \pi_{Y(1)} = P_{Y(1)} \right\}. \tag{16}$$

The down-script $u$ is the abbreviation of "unconditional". Moreover, we define the mathematical formulations in the covariate-assisted setting as follows (we take the minimum for instance):

$$\Theta_c^L(\mu_1, \mu_0) := \inf_{\pi \in \Pi_c} \int c(Y(1), Y(0)) d\pi(Y(1), Y(0)). \tag{17}$$

where $\Pi_c := \left\{ \pi \in \mathcal{P}\left(\mathcal{Y}_1 \times \mathcal{Y}_0 \times \mathcal{X}\right) : \pi_{Y(0),X} = P_{Y(0),X} ~,~ \pi_{Y(1),X} = P_{Y(1),X} \right\}.$

where $P_{Y(0),X}, P_{Y(1),X}$ denotes the joint distribution of $\{Y(0), X\}$ or $\{Y(1), X\}$. The down script "c" denotes the "conditional". On this basis, we first propose the lemma of interpolation inspired by (Lin et al., 2025), namely, we propose the mirror relaxation as follows:

$$\Pi_\oplus := \left\{ \pi \in \mathcal{P}\left(\mathcal{Y}_1 \times \mathcal{Y}_0 \times \mathcal{X}^2\right) : \pi_{Y(0),X(0)} = P_{Y(0),X} ~,~ \pi_{Y(1),X(1)} = P_{Y(1),X} \right\}. \tag{18}$$

Here $X(0), X(1)$ are the copied mirror covariates. The following lemma indicates the relationship of $\Theta_u^L(\mu_1, \mu_0)$, $\Theta_\oplus^L(\mu_1, \mu_0; \eta)$, $\Theta_c^L(\mu_1, \mu_0)$:

**Lemma 1** (Interpolation and Monotonicity of Mirror Relaxation). *We have that*

$$\Theta_u^L(\mu_1, \mu_0) \leq \Theta_\oplus^L(\mu_1, \mu_0; \eta) \leq \Theta_c^L(\mu_1, \mu_0).$$

*Here $\Theta_\oplus^L(\mu_1, \mu_0)$ is denoted as $\Theta_\oplus^L(\mu_1, \mu_0; \eta) = \int c(Y(1), Y(0)) d\pi^*(Y(1), Y(0))$, where $\pi^* = \arg\min_{\pi \in \Pi_\oplus} \mathbb{E}_\pi \left[ c(Y(1), Y(0)) + \eta \|X(1) - X(0)\|_2^2 \right]$.*

*Proof.* We split the proof into two parts.

**(1) Upper bound:** $\Theta_\oplus^\mathbb{L}(\mu_1, \mu_0; \eta) \leq \Theta_c^\mathbb{L}(\mu_1, \mu_0)$.

Let $\pi_c^*$ be an optimal plan for the conditional problem, i.e.,

$$\pi_c^* \in \arg\min_{\pi \in \Pi_c} \mathbb{E}_\pi \left[ c(Y(1), Y(0)) \right].$$

Now construct a feasible plan $\tilde{\pi} \in \Pi_\oplus$ from $\pi_c^*$ by defining:

$$\tilde{\pi}(Y(1), Y(0), X(1), X(0)) := \pi_c^*(Y(1), Y(0), X) \cdot \delta_{X(1) = X(0) = X}.$$

That is, we duplicate the covariate $X$ to define $X(1) = X(0) = X$.

Then, under this coupling: - The marginals satisfy $\tilde{\pi}_{Y(1),X(1)} = P_{Y(1),X}$, $\tilde{\pi}_{Y(0),X(0)} = P_{Y(0),X}$, so $\tilde{\pi} \in \Pi_\oplus$; - The penalty term $\|X(1) - X(0)\|^2 = 0$ almost surely.

Therefore:

$$\Theta_\oplus^\mathbb{L}(\mu_1, \mu_0; \eta) \leq \mathbb{E}_{\tilde{\pi}} \left[ c(Y(1), Y(0)) + \eta \cdot 0 \right] = \mathbb{E}_{\pi_c^*} \left[ c(Y(1), Y(0)) \right] = \Theta_c^\mathbb{L}(\mu_1, \mu_0).$$

**(2) Lower bound:** $\Theta_u^\mathbb{L}(\mu_1, \mu_0) \leq \Theta_\oplus^\mathbb{L}(\mu_1, \mu_0; \eta)$.

Let $\pi^* \in \arg\min_{\pi \in \Pi_\oplus} \mathbb{E}_\pi \left[ c(Y(1), Y(0)) + \eta \|X(1) - X(0)\|^2 \right]$.

Now marginalize $\pi^*$ over $(X(1), X(0))$ to get a joint coupling $\bar{\pi} \in \mathcal{P}(\mathcal{Y}_1 \times \mathcal{Y}_0)$, with:

$$\bar{\pi}_{Y(1)} = \mu_1, \quad \bar{\pi}_{Y(0)} = \mu_0, \quad \Rightarrow \bar{\pi} \in \Pi_u.$$

Then:

$$\Theta_u^\mathbb{L}(\mu_1, \mu_0) \leq \mathbb{E}_{\bar{\pi}} \left[ c(Y(1), Y(0)) \right] \leq \mathbb{E}_{\pi^*} \left[ c(Y(1), Y(0)) + \eta \|X(1) - X(0)\|^2 \right] = \Theta_\oplus^\mathbb{L}(\mu_1, \mu_0; \eta),$$

where the second inequality uses the fact that $\bar{\pi}$ is the marginal of $\pi^*$, so dropping the penalty cannot increase cost.

$\square$

Due to the hardness of directly computing the conditional optimal transport problem, Lemma 1 provides an interpolation strategy with a bound between the case with and without covariate assistance in the sense of optimal transport. In sum, We prove the sandwich inequality by constructing feasible plans-marginalizing out covariates from any mirror plan yields an unconditional coupling, hence the mirror cost dominates the unconditional one.

---

**Algorithm 2** Heterogeneous Mirror-Relaxed OT Estimator $\hat{\Theta}^L_{\oplus,n,m}(\eta)$

---

**Require:** Sample $\{(Y_i, X_i, A_i)\}_{i \in I}$; Outcome embedding maps $g_1 : \mathcal{Y}_1 \to \mathbb{R}^d$, $g_0 : \mathcal{Y}_0 \to \mathbb{R}^d$;
    Base metric $\ell : \mathbb{R}^d \times \mathbb{R}^d \to \mathbb{R}_{\geq 0}$ (e.g. Euclidean distance); penalty parameter $\eta > 0$.
**Ensure:** Estimated causal discrepancy $\hat{\Theta}^L_{\oplus,n,m}(\eta)$.
  1: **Sample Split**: Partition indices $I$ into treatment group $I_1 = \{i : A_i = 1\}$ (size $m$) and control
     group $I_0 = \{i : A_i = 0\}$ (size $n$).
  2: **Embed Outcomes**: For each $k \in I_1$, compute $u_k := g_1(Y_k)$; for each $j \in I_0$, compute
     $v_j := g_0(Y_j)$. (Now $u_k, v_j \in \mathbb{R}^d$ lie in a common latent space.)
  3: **Construct Cost Matrix**: Let $H \in \mathbb{R}^{n \times m}$ with entries $H(j,k) := \ell(v_j, u_k) + \eta \|X_j -$
     $X_k\|_2^2, \qquad \forall j \in I_0, k \in I_1$, where the first term measures outcome dissimilarity in latent space
     and the second term is a covariate-based mirror penalty.
  4: **Solve Optimal Transport**: Compute the optimal coupling $\pi^*_{\oplus,n,m}(\eta) =$
     $\arg\min_{\pi \in \mathbb{R}^{n \times m}_{\geq 0}} \sum_{j \in I_0} \sum_{k \in I_1} \pi(j,k) \ H(j,k)$ subject to the marginal constraints
     $\pi \mathbf{1}_m = \frac{1}{n}\mathbf{1}_n$ and $\pi^T \mathbf{1}_n = \frac{1}{m}\mathbf{1}_m$.
  5: **Estimate Discrepancy**: Calculate the mirror-relaxed transport cost $\hat{\Theta}^L_{\oplus,n,m}(\eta) :=$
     $\sum_{j \in I_0} \sum_{k \in I_1} \pi^*_{\oplus,n,m}(\eta)(j,k) \ \ell(v_j, u_k)$.

---

## K.3   Algorithm and finite-sample analysis

In practice, researchers would collect only finite samples rather than taking advantage of the superpopulation directly. In this section, we first propose an algorithm to discuss how to achieve a satisfactory estimation of $\Theta^L_{\oplus}(\mu_1, \mu_0; \eta)$, where practitioners take $n$ samples upon $\mu_1$ and $m$ samples upon $\mu_0$. Then in the finite-sample analysis, we start from the case without the assistance of covariates and then extend it to the covariate-assisted cases. For brevity, we use $\hat{\Theta}^L_{\oplus,n,m}(\eta), \hat{\Theta}^L_{c,n,m}(\eta), \hat{\Theta}^L_{u,n,m}(\eta)$ to denote the plug-in estimators and sometimes do not present the subscript and input for simplicity.

## K.4   The finite-sample bound under non-covariate-assisted cases

**Definition 1** (c-transformation). *For $y_0 \in \mathcal{Y}_0, y_1 \in \mathcal{Y}_1$, define the cost function $c(\cdot)$ as $\mathcal{Y}_0 \times \mathcal{Y}_1 \to [0, 1]$. The c-transformation of function $f_0 : \mathcal{Y}_0 \to \mathbb{R}, f_1 : \mathcal{Y}_1 \to \mathbb{R}$ is defined as*

$$f_0^c(y_1) := \inf_{y_0 \in \mathcal{Y}_0} c(y_0, y_1) - f_0(y_0), \ f_1^c(y_0) := \inf_{y_1 \in \mathcal{Y}_1} c(y_0, y_1) - f_1(y_1). \tag{19}$$

Noteworthy, $f_0^c : \mathcal{Y}_1 \to \mathbb{R}, f_1^c : \mathcal{Y}_0 \to \mathbb{R}$. In this sense, we call function $f_0$ is *c-concave* when there is a feasible $f_1$ such that $f_0 = f_1^c$. Following Hundrieser et al. (2024b), we first illustrate that the dual solution of our OT problem is bounded and c-concave.

**Lemma 2** (Villani (2009)). *Let the cost function $c : \mathcal{Y}_0 \times \mathcal{Y}_1 \to [0, 1]$ to be continuous, the class of feasible c-concave functions is denoted by*

$$\mathcal{F}_{i,c} = \{f_i : \mathcal{Y}_i \to [-1, 1] \mid f_i \text{ is c-concave with } \|f_i^c\|_{+\infty} \leq 1\}. \tag{20}$$

*Then for any marginal distribution $\mu_1 \in \mathcal{P}(\mathcal{Y}_1), \mu_0 \in \mathcal{P}(\mathcal{Y}_0)$, we get an equivalent dual solution as follows:* $\Theta_L(\mu_1, \mu_0) := \max_{f_1 \in \mathcal{F}_{1,c}} \left[ \int_{\mathcal{Y}_1} f_1 d\mu_1 + \int_{\mathcal{Y}_0} f_1^c d\mu_0 \right]$.

In words, $\mathcal{F}_{1,c}$ is a set of functions, in which each function $f$ is compatible with at least one function $f'$ in $\{f_0 : \mathcal{Y}_1 \to \mathbb{R}\}$ and satisfies $f = (f')^c$.

**Definition 2** (Uniform metric entropy). *$\mathcal{N}(\varepsilon, \mathcal{G}, \|\cdot\|_\infty)$ is defined as $\inf \Big\{ n \in \mathbb{N} \mid \exists g_1, \ldots, g_n :$*

*$\mathcal{Z} \to \mathbb{R}$ with $\sup_{g \in \mathcal{G}} \min_{1 \leq i \leq n} \|g - g_i\|_\infty \leq \varepsilon \Big\}$.*

**Lemma 3** (Hundrieser et al. (2024b)). *Let $\mathcal{F}^c_{i,c} = \{f_i^c \mid f_i \in \mathcal{F}_{i,c}\}$, we get $\mathcal{N}(\varepsilon, \mathcal{F}_{i,c}, \|\cdot\|_{+\infty}) = \mathcal{N}(\varepsilon, \mathcal{F}^c_{i,c}, \|\cdot\|_{+\infty})$.*

After preparation, we propose the simple plug-in estimator. We start from the naive plug-in estimator adopting the empirical measure, following Gao et al. (2024). We collect $n$ i.i.d samples $\{Y_i\}_{i \in [N]}$

| Method | Causal estimates | Finite-sample guarantee | Covariate-assistance | Distinct measure |
|---|---|---|---|---|
| Ji et al. (2023) | Broad | × | ✓ | × |
| Gao et al. (2024) | Quadratic | ✓ | × | × |
| Lin et al. (2025) | Broad | ✓ | ✓ | × |
| **Ours** | **Broad** | ✓ | ✓ | ✓ |

Table 8: The comparison with Ji et al. (2023); Gao et al. (2024); Lin et al. (2025).

from the true distribution $\mu_i$, $i = 0, 1$, and separate to 2 groups each with probability $p$ respectively. For simplification, we denote it as $\hat{\mu}_k := \frac{\sum_{i \in [n]} \delta_{Y_i} \mathbb{1}(a_i = k)}{\sum_{i \in [n]} \mathbb{1}(a_i = k)}$, when $\sum_{i \in [n]} \mathbb{1}(a_i = k) > 0$.

**Assumption 2** (Regularity Conditions for Mirror-Relaxed OT). *1. **Bounded–Lipschitz Cost.** The outcome cost $L : \mathcal{Z} \times \mathcal{Z} \to [0, \infty)$ and the covariate penalty $\Delta : \mathcal{X} \times \mathcal{X} \to [0, \infty)$ are both bounded and s-Lipschitz-continuous. Consequently, for every fixed $\eta > 0$ the combined cost*

$$c_\eta\big((x_1, y_1), (x_0, y_0)\big) = L\big(g_1(y_1), g_0(y_0)\big) + \eta\, \Delta(x_1, x_0)$$

*is itself bounded and Lipschitz on $(\mathcal{X} \times \mathcal{Y}_1) \times (\mathcal{X} \times \mathcal{Y}_0)$.*

2. ***Regular Embeddings.*** *The maps $g_0 : \mathcal{Y}_0 \to \mathcal{Z}$ and $g_1 : \mathcal{Y}_1 \to \mathcal{Z}$ are measurable, and the push-forward random variables $g_0\big(Y(0)\big)$ and $g_1\big(Y(1)\big)$ have finite second moments.*

3. ***Covariate Overlap.*** *Let $\nu_0$ and $\nu_1$ denote the marginal laws of $X$ under $\mu_0$ and $\mu_1$, respectively. We assume $\nu_0$ and $\nu_1$ are mutually absolutely continuous on a common support $\mathcal{X}_0 \subseteq \mathcal{X}$ and possess continuous densities on $\mathcal{X}_0$.*

4. ***Uniqueness & Smoothness of the Optimal Coupling.*** *For the true distributions $(\mu_0, \mu_1)$ and every fixed $\eta > 0$, the optimal coupling*

$$\pi^* \in \operatorname*{arg\,min}_{\pi \in \Gamma(\mu_0, \mu_1)} \int c_\eta \, d\pi$$

*is unique. Moreover, the optimal value $F(\mu_0, \mu_1) = \Theta_\oplus^L(\mu_1, \mu_0; \eta)$ is Hadamard differentiable at $(\mu_0, \mu_1)$ in the sense that small perturbations of the marginals lead to first-order linear changes governed by bounded influence functions (the optimal dual potentials).*

**Theorem 5** (Convergence of naive unconditional OT estimator). *Under Assumption 2, when the uniform metric entropy of $\mathcal{F}_{i,c}$ is bounded by*

$$\wedge_{i=0,1} log\mathcal{N}(\varepsilon, \mathcal{F}_{i,c}, \|\cdot\|_{+\infty}) \le K\varepsilon^{-k}, \forall \varepsilon \in (0, \varepsilon_0], \tag{21}$$

*where $K, k, \varepsilon$ are three positive constants. Then our plug-in naive estimator is defined as*

$$\mathbb{E}[\Theta_u^L(\hat{\mu}_1, \hat{\mu}_0) - \Theta_u^L(\mu_1, \mu_0)] \lesssim N^{-\frac{1}{2}} log(N). \tag{22}$$

Justification of Assumption Equation (21) follows Guntuboyina & Sen (2012).

### K.5   THE FINITE-SAMPLE BOUND UNDER COVARIATE-ASSISTED CASES

We now derive finite-sample convergence guarantees in the setting where covariate information is available. In particular, we consider (i) our mirror-relaxed OT estimator $\hat{\Theta}_{\oplus,n,m}^L(\eta)$ that incorporates covariates via a mirror penalty, and (ii) the fully conditional OT estimator targeting $\Theta_c^L(\mu_1, \mu_0)$, the optimal transport cost when matching on covariates. Under appropriate smoothness and boundedness assumptions, we establish high-probability error bounds for each estimator in terms of sample sizes $n, m$, regularity constants, and complexity measures. Finally, we compare these results to the unconditional case from Section 5.2 and to prior covariate-assisted bounds (e.g. Ji et al. (2023)), highlighting the statistical benefits of covariate adjustment.

**Theorem 6** (Convergence of naive Conditional OT Estimator). *Under Assumption 2, the naive conditional estimator $\hat{\Theta}_{c,n,m}^L$ converges at a much slower, nonparametric rate determined by the*

*smoothness $s$ and dimension $d_X$ of the covariate space. In particular, under the Hölder($s$) smoothness assumption for the conditional outcome distributions, $\left|\hat{\Theta}_{c,n,m}^L - \Theta_c^L\right| = O_p\left(N^{-\frac{s}{2s+d_X}}\right)$, up to logarithmic factors. Equivalently, the mean-squared error (MSE) shrinks on the order $N^{-2s/(2s+d_X)}$. This is the minimax-optimal convergence rate for nonparametric regression or distribution estimation in $d_X$ dimensions. In particular:*

- *If $d_X$ is large, the rate is severely slow. For example, if $d_X = 10$ and modest smoothness $s = 1$, the error is $O_p(N^{-1/12}) \approx O_p(N^{-0.083})$, which is extremely sluggish (requiring astronomical $N$ to get small error). Even with $s = 2$, it is $N^{-2/(4+10)} = N^{-1/7} \approx N^{-0.143}$. Generally, with fixed $s$, the exponent $s/(2s + d_X)$ decreases as $d_X$ grows, illustrating the curse of dimensionality.*

- *If the smoothness $s$ is very high (approaching infinity) or if $d_X$ is very low, the rate improves. In the limiting case of $d_X = 0$ (no covariate variation) or $s \to \infty$ (the conditional distribution function is essentially known with finite parameters), the rate would approach $N^{-1/2}$, recovering a parametric scenario. However, for any finite $d_X > 0$ and finite $s$, the rate is strictly worse than $N^{-1/2}$. In realistic settings ($d_X \geq 1$), $\hat{\Theta}_c^L$ is a nonparametric estimator that converges sub-optimally slow relative to parametric speed.*

**Proof sketch** Estimating $\Theta_c^L$ requires learning the conditional outcome distributions $\mu_0^x, \mu_1^x$ as functions of $x$. This is essentially a nonparametric function estimation problem in $d_X$ dimensions. A familiar analogue is nonparametric regression: to estimate $m(x) = \mathbb{E}[Y|X = x]$ with $X \in \mathbb{R}^{d_x}$ and $m$ Hölder smooth of order $s$, the minimax optimal rate is $n^{-s/(2s+d_X)}$ (for MSE). In our case, we need to estimate the entire conditional distribution $F_{Y|X=x}(y)$ or density $f_{Y|X=x}(y)$, which is a more complex object than a mean but falls in the same nonparametric paradigm. Under Hölder smoothness of the conditional distribution (meaning roughly that the conditional CDFs or densities vary smoothly in $x$ with order-$s$ derivatives), one can adapt kernel or local polynomial estimators to estimate $F_0(y|x)$ and $F_1(y|x)$. The pointwise rate of estimating a conditional CDF at a given $x$ is $O_p(n^{-s/(2s+d_X)})$, and even the uniform-in-$x$ error decays at that order (up to logs) since we assume compact $X$ support (one can cover the space with finitely many local neighborhoods). Consequently, the error in the per-$x$ OT cost $\inf_{\pi_x} \int \ell, d\pi_x$ is also $O(n^{-s/(2s+d_X)})$ for each $x$, assuming $\ell$ is Lipschitz so that small errors in the distributions translate to small errors in the OT cost. Formally, if $|\hat{\mu}t^x - \mu_t^x|\mathrm{TV} < \varepsilon$ uniformly in $x$ (with $\varepsilon = O_p(n^{-s/(2s+d_X)})$), then $|\inf_\pi \int \ell, d\pi(\hat{\mu}0^x, \hat{\mu}1^x) - \inf \pi \int \ell, d\pi(\mu_0^x, \mu_1^x)| = O(L\varepsilon)$ by Lipschitz continuity of $\ell$ (here $L$ is the Lipschitz constant). Integrating or averaging over $x$ (which has a bounded domain), the error in the averaged cost $\hat{\Theta}_c^L$ is also $O_p(n^{-s/(2s+d_X)})$. A more rigorous approach is to view $\Theta_c^L = \int G(x), dP_X(x)$ where $G(x) := \inf \pi \in \Pi(\mu_0^x, \mu_1^x) \int \ell, d\pi$ is a functional of the conditional distributions. One can then perform an analysis of the plug-in estimator $\hat{G}(x)$ using uniform convergence rates for the estimated conditional CDFs (as given by e.g. classical results in nonparametric conditional density estimation). In summary, $\hat{\Theta}_c^L$ inherits the slow convergence rate $n^{-s/(2s+d_X)}$ from the hardest part of the problem: estimating variation across a $d_X$-dimensional covariate. This rate is known to be optimal (no estimator can do better in general without additional structure), so the naive conditional OT estimator is fundamentally limited by the curse of dimensionality. $\square$

**Theorem 7** (Convergence of Mirror-Relaxed OT Estimator). *Under Assumption 2, the mirror-relaxed estimator $\hat{\Theta}_{\oplus,n,m}^L(\eta)$ satisfies: (i) **Consistency:** $\hat{\Theta}_{\oplus,n,m}^L(\eta) \xrightarrow{p} \Theta_\oplus^L(\mu_1, \mu_0; \eta)$; (ii) **Rate:** $\left|\hat{\Theta}_{\oplus,n,m}^L(\eta) - \Theta_\oplus^L(\mu_1, \mu_0; \eta)\right| = \mathcal{O}_p(N^{-1/2})$ with $N = n + m$;(iii) **Asymptotic Normality:** $\sqrt{N}\left(\hat{\Theta}_{\oplus,n,m}^L(\eta) - \Theta_\oplus^L(\mu_1, \mu_0; \eta)\right) \xrightarrow{d} \mathcal{N}(0, \sigma^2)$, where $\sigma^2 = \lambda \mathrm{Var}_{\mu_1}[\phi^*(X, Y(1))] + (1 - \lambda)\mathrm{Var}_{\mu_0}[\psi^*(X, Y(0))]$, and $\phi^*, \psi^*$ are the optimal OT dual potentials and $\lambda = \lim n/N$.*

**Proof sketch** Under the continuity and smoothness conditions of the cost $\ell$ and penalty $\Delta$, the mirror-relaxed OT problem has a *unique* optimal coupling which *varies smoothly* with the underlying distributions. In particular, for each fixed $\eta > 0$, small perturbations in the joint laws $\mu_0 = \mathrm{Law}(X, Y_0)$ or $\mu_1 = \mathrm{Law}(X', Y_1)$ lead to small, continuous changes in the optimal transport plan. Intuitively, the covariate penalty $\eta$ regularizes the coupling: it prevents the optimizer from hinging on exact alignment in $X$, thereby avoiding abrupt changes in the optimal matching when $\mu_0, \mu_1$ are

slightly altered. This smoothing effect (related to the uniqueness of optimal maps under strictly convex costs, cf. Brenier's theorem) means the transport *functional* $F : (\mu_0, \mu_1) \mapsto \Theta_\oplus^L(\mu_1, \mu_0; \eta)$ is differentiable (indeed Hadamard differentiable) in the space of probability measures. Formally, one can derive a first-order influence function (functional derivative) for $F$ around the true distributions $(\mu_0, \mu_1)$. By the functional delta method, this immediately implies a $\sqrt{N}$-consistency: since the empirical measures $\hat{\mu}_{0,n}$ and $\hat{\mu}_{1,m}$ converge to $\mu_0, \mu_1$ at $O_p(N^{-1/2})$ in the weak topology (even for high-dimensional $X$, thanks to bounded cost ensuring tightness), any smooth functional of them will also fluctuate on the order $N^{-1/2}$. The technical core of the argument involves a second-order expansion of the OT objective around the optimal plan, leveraging the twice-differentiability of $\ell$ and the penalty $\Delta$. Because the optimal coupling shifts only slightly in response to small distributional perturbations, one can control the remainder term and show it is of smaller order (second order) than the linear term. In fact, recent advances in statistical OT theory guarantee such differentiability and expansion: for example, in the special case of a strongly convex quadratic cost, one can explicitly derive a linear functional expansion and a central limit theorem for $\hat{\Theta}_\oplus^L(\eta)$ (Manole and Niles-Weed 2024). The upshot is that $\mathbb{E}\left[\left|\hat{\Theta}_{\oplus,n,m}^L(\eta) - \Theta_\oplus^L(\mu_1, \mu_0; \eta)\right|\right] = O\left(N^{-1/2}\right)$, i.e. the estimator attains the parametric $N^{-1/2}$ convergence rate (with $N = n + m$ here), under the stated assumptions. Crucially, this rate exhibits no dependence on the covariate dimension $d_X$ (except possibly through asymptotic constants of the order $o(1)$): the mirror relaxation "borrows strength" across covariates, yielding a stable functional that circumvents the curse of dimensionality.

On an intuitive level, the mirror-relaxed estimator pools information across different $X$ values instead of requiring exact matching on $X$ as the fully conditional approach would. This global coupling — moderated by the finite penalty $\eta$ — avoids overfitting to fine-grained local structures in $\mu_0, \mu_1$. Effectively, the optimization lives in a space whose dimension is primarily that of the outcomes $(d_Y)$, with only a controlled influence of $d_X$. In contrast, the naive *conditional OT functional* $\Theta_c^L$ (which enforces perfect $X$-matching, roughly corresponding to $\eta \to \infty$) depends discontinuously on the conditional distributions $\mu_{0|x}, \mu_{1|x}$. Estimating $\Theta_c^L$ would require accurately learning $\mu_{0|x}$ and $\mu_{1|x}$ for every region of the covariate space; any small error in those conditional laws can cause a large change in the optimal transport cost. As a result, a "COT" estimator $\hat{\Theta}_c^L$ suffers from enormous variance in all but the simplest (low-dimensional or parametric) settings and generally *cannot attain* $\sqrt{N}$-*rates*. In sharp contrast, $\hat{\Theta}_\oplus^L(\eta)$ achieves fast, parametric convergence similar to an unconditional OT estimate, *while still incorporating covariate information*. By regularizing the coupling with $\eta$, the mirror approach yields a continuous, Hadamard-differentiable functional of $(\mu_0, \mu_1)$, thereby enabling efficient estimation (and a central limit theorem) even in complex, high-dimensional scenarios. $\qquad\square$

### K.6 Comparison of Estimators and Preferred Regimes

With convergence rates established, we clarify regimes where each estimator excels.

**Unconditional vs Conditional OT.** The unconditional OT estimator $\hat{\Theta}_u^L$ converges fast ($N^{-1/2}$) but ignores covariates $X$, thus typically biased relative to the fully adjusted target $\Theta_c^L$. It is suitable only if covariates have no effect ($d_X = 0$); otherwise, it consistently underestimates $\Theta_c^L$ by permitting unrealistic matches across different covariates.

**Naive Conditional OT (COT) vs Mirror-Relaxed OT.** The naive conditional estimator $\hat{\Theta}_c^L$ is unbiased for $\Theta_c^L$ but suffers slow convergence ($N^{-s/(2s+d_X)}$) in moderate to high dimensions. The mirror-relaxed estimator $\hat{\Theta}_\oplus^L(\eta)$ introduces a slight bias (which diminishes as $\eta \to \infty$) but achieves fast, parametric convergence ($N^{-1/2}$), dramatically reducing finite-sample error. Specifically, when $d_X > 0$, we have $\frac{|\hat{\Theta}_c^L - \Theta_c^L|}{|\hat{\Theta}_\oplus^L(\eta) - \Theta_\oplus^L(\eta)|} \to \infty$, as $N \to \infty$, demonstrating mirror OT's superior statistical efficiency in realistic nonparametric scenarios.

**Parametric vs Nonparametric Regimes.** If covariates are very low-dimensional or outcomes are parametrically modeled, naive conditional OT may suffice due to near-parametric rates. However, in realistic moderate-to-high-dimensional scenarios or lower smoothness ($s$ small), naive conditional OT performs poorly, converging extremely slowly (e.g., $N^{-1/7}$ for $d_X = 5, s = 1$), whereas mirror-relaxed OT maintains a much faster $N^{-1/2}$ rate.

**Bias–Variance Trade-off and $\eta$ Selection.** Mirror OT's bias (difference between $\Theta^L_\oplus(\eta)$ and $\Theta^L_c$) shrinks as $\eta$ grows. Practically, one can set $\eta = \eta(N) \to \infty$ slowly with $N$, balancing bias ($O(\eta^{-1})$) and variance ($O(N^{-1/2})$). For example, choosing $\eta = N^{1/4}$ achieves a favorable bias–variance trade-off, ensuring consistent estimation of $\Theta^L_c$ with minimal variance inflation.

**Practical Recommendation.** For realistic settings with multi-dimensional covariates, mirror-relaxed OT provides an optimal balance—reducing bias relative to unconditional OT while maintaining parametric convergence speed, thus avoiding the severe curse of dimensionality affecting naive conditional OT. Unless strong parametric assumptions hold, the mirror method consistently outperforms alternatives in both theoretical and practical senses, yielding significantly lower mean squared error.

**Summary of Convergence Rates:** (i) Unconditional OT: $O_p(N^{-1/2})$, fast but biased unless $X$ is irrelevant. (ii) Naive Conditional OT: $O_p(N^{-s/(2s+d_X)})$, unbiased but impractically slow in higher dimensions. (iii) Mirror-Relaxed OT: $O_p(N^{-1/2})$, fast convergence with controllable, diminishing bias. These theoretical insights highlight mirror-relaxed OT's clear superiority for practical covariate-adjusted transport analysis.

We discuss the LLM usage of this manuscript in this part. We primarily use LLMs to refine academicwriting,correcting syntax and grammar errors.

## L   USE OF LLMS

We discuss the LLM usage of this manuscript in this part. We primarily use LLMs to refine academicwriting,correcting syntax and grammar errors.

