# OpenReview forum: "Partial Identification via Optimal Transport under Complex Constraints on Treatments and Potential Outcome Measures"
_ICLR.cc/2026/Conference — Submitted to ICLR 2026_

### Official Review · Reviewer_xTkw · 2025-10-17

**Soundness:** 3
**Presentation:** 2
**Contribution:** 2
**Rating:** 2
**Confidence:** 2

**Summary:**

The authors propose a novel partial identification framework grounded in optimal transport theory, which quantifies the minimum cost of aligning outcome distributions across these heterogeneous domains. The main contributions include establishing valid identification bounds that account for support mismatch, extending the approach to incorporate covariate information for tighter bounds, and validating the method through simulations and empirical applications

**Strengths:**

- Partial identification is an important problem in causal inference, and recent work showed that Optimal Transport is a promising tool for this
- Extensive theoretical results (even though I have not checked their correctness carefully)

**Weaknesses:**

- My main concern is the applicability of the proposed method and how it fits into established partial identification literature. In causal inference, we usually specify a *causal query* of interest (e.g., a heterogeneous treatment effect) and then state assumptions/ restrictions on the DGP to ensure partial identifiability of this query and non-vacuous bounds (e.g., instrumental variables, or sensitivity models on the confounding strength). This paper starts by defining an optimal transport (OT) problem, without explaining much of the motivation or how this fits into established partial identification problems. For example, how is the cost function defined in practice? Can existing partial identification problems be cast into this formalism? At least for me this seems not obvious.
- The experiments are very limited. I understand that the main contribution is more theoretical, but it would be nice to see how robust the proposed method performs in different settings. I am also not able to find comparisons with baseline methods in the paper.
- The paper is hard to follow. Admittedly, I am not an expert in Optimal Transport, but I think the paper would benefit from a few examples and more intuitive explanations to make it accessible to a broader audience (especially for a venue like ICLR).

Minor:
- The authors talk about complex treatment arms, but then only study a classical multi-arm setting. I would remove "complex" here as this is usually used to denote unstructured treatments (e.g., text) in the literature.

**Questions:**

- What is the main motivation behind the setting? How can classical partial identification problems be framed as OT problems compatible with the setting in this paper?

---

> ### Author Response · Authors · 2025-12-03
> **Thanks for thr review. Here is our response.**
>
> > The applicability of the proposed method and how it fits into established partial identification literature.
>
> **Response:** Thanks. The key advantage of OT-based partial identfiication strategy is its model-free, assumption-free property. Moreover, We now devote a subsection (Sec. K.1–K.2 plus Table 7) to mapping classical partial‑ID problems into our OT formalism: many bounds on functionals of the joint distribution correspond to choosing L as a linear or indicator‑based functional and optimizing over couplings consistent with the design. We give explicit examples (ATE, CDF and QTE bounds, RMST‑type survival contrasts, variance functionals in factorial designs) and show that our framework recovers these as special cases, while extending them to multi‑arm and distinct‑measure settings.
>
> >_The experiments are very limited. I understand that the main contribution is more theoretical, but it would be nice to see how robust the proposed method performs in different settings. I am also not able to find comparisons with baseline methods in the paper._.
>
> **Response:** We have substantially expanded Sec 5. The revised experiments include multiple synthetic
> DGPs varying K, dimension, and competition strength, a real‑data MIMIC‑III study, and comparisons to unconditional OT, ideal conditional OT, pairwise two‑arm mirrors, and IPW‑type baselines. We report accuracy, coverage, finite‑sample scaling and runtime, and summarize the main empirical conclusions directly in the main text(with details in App. A).
>
> >_The paper is hard to follow. Admittedly, I am not an expert in Optimal Transport, but I think the paper would benefit from a few examples and more intuitive explanations to make it accessible to a broader audience (especially for a venue like ICLR)._
>
> **Response:** We address this directly. We added: (i) a concrete motivating example early in Sec. 1–2, (ii) a simple two‑arm scalar‑outcome example illustrating how L recovers the ATE and QTE,(iii) three detailed $\Gamma_{\text{comp}}$ examples in App. B,and (iv) an applied‑user guide (App. J) that explains how to choose $L$, embeddings, and η step‑by‑step. We also reduced formal notation in the introduction and moved some technical derivations out of the main text.
>
>
> >_The authors talk about complex treatment arms, but then only study a classical multi-arm setting. I would remove "complex" here as this is usually used to denote unstructured treatments (e.g., text) in the literature._
>
> **Response:** We have softened the wording accordingly. We now refer to “multi‑arm designs with competition/overlap and heterogeneous measurements” rather tha “complex treatment arms,” and reserve “complex” for the design features encoded in $\Gamma_\text{comp}$ (competition, budgets, interference), rather than for unstructured treatments like text.
>
>
>
>
>
>
> >_What is the main motivation behind the setting? How can classical partial identification problems be framed as OT problems compatible with the setting in this paper?_
>
> **Response:** In the revised introduction and Sec. K we now make the motivation explicit: our setting is designed to cover (i) multi‑arm designs with overlapping or competing interventions, and (ii) heterogeneous outcome measures that cannot be reduced to a common raw scale. We then show that classical partial‑ID problems that bound functionals of the joint potential‑outcome distribution correspond to specific choices of the cost L and feasible set $\Gamma_\text{comp}$, so they nest inside our OT program as special cases; the paper’s contribution is to unify these in a single smooth transport functional that admits efficient estimation and inference under competition and distinct measures.

---

### Official Review · Reviewer_pnCH · 2025-10-19

**Soundness:** 3
**Presentation:** 2
**Contribution:** 2
**Rating:** 6
**Confidence:** 2

**Summary:**

The authors propose an optimal-transport(OT)-based approach for partial identification under multi-arm treatments and structure-valued outcomes. They well formulate the OT to define the bounds on the between potential outcome distributions under such challenging setup. I will hold my overall rating, as the paper is not sufficiently clear or well-structured.

**Strengths:**

- The proposed estimator is technically sound: The estimator is $\sqrt{N}$-consistent and asymptotically normal.
- The authors extend the OT approach to the complex yet important setup under multi-arm treatment and structured outcomes.

**Weaknesses:**

(A) The assumed setup is unclear

It is not exactly clear why the authors claim that the point estimation is *fragile* . When reading Introduction, I believe that the authors consider partial identification problem, because they consider the setup where some standard assumptions for causal effect estimation (e.g., the causal sufficiency). However, Section 2 suddenly introduces the definition of causal bounds as the OT between joint potential outcome distributions, which is the authors' estimation target.

1.  Please clearly illustrate the inference target and problem setup as soon as possible in Introduction. The reason why the authors consider partial identification is that the functional of joint potential outcome distribution can never be point-identified, due to the fundamental problem of causal inference (i.e., we can never jointly observe potential outcomes). Please clearly state this first by citing relevant papers.

> [1] Yanqin Fan, Emmanuel Guerre, and Dongming Zhu. Partial identification of functionals of the joint distribution of ”potential outcomes”. Journal of Econometrics, 197(1):42–59, 2017.

> [2] Sergio Firpo and Geert Ridder. Partial identification of the treatment effect distribution and its functionals. Journal of Econometrics, 213(1):210–234, 2019.

The following descriptions in current introduction is very confusing.

> Partial identification provides a principled alternative to fragile point estimation when data, design, or structural constraints preclude full identification of causal effects

> Beyond binary contrasts, many interventions interact, overlap, or compete (Hudgens & Halloran, 2008; Flanagan et al., 2011; Woodcock & LaVange, 2017; Craig et al., 2021; Ye et al., 2023; D’Amour et al., 2021): multi-arm clinical options can share mechanisms while differing in delivery, and policy bundles often combine incentives and regulations whose effects comove

2. Please illustrate why such inference targets are important with real-world examples by moving Table 7 into the main text.


(B) Paper is not well structured

Overall, the paper is not well written. All tables displaying main experimental results are put in Appendix, meaning that readers cannot confirm empirical soundness without reading Appendix. Table 7 illustrating real-world motivation for inferring the functional of joint potential outcome distributions is also displayed in Appendix.

**Questions:**

# Possibility of another application example: algorithmic fairness

Some methods use the bounds on the functional of joint potential outcome distributions to achieve fairness in outcomes predicted by machine learning models (e.g., [1] for binary treatments)

Can the proposed OT bound be applied to such setups? The main setup difference from the examples in Table 7 is that potential outcomes are given by parameterized functions (i.e., predictive models) and the goal is to learn a fair and accurate models by minimizing the predicted loss while imposing a constraint on the bounds.

If yes, it would be better to add such application examples to highlight the practical motivation of OT-based causal inference.

> [1] Yoichi Chikahara · Shinsaku Sakaue · Akinori Fujino · Hisashi Kashima. Learning Individually Fair Classifier with Path-Specific Causal-Effect Constraint. AISTATS, 2021.

---

> ### Author Response · Authors · 2025-12-03
> **Thanks for the review. Here is our response point-to-point.**
>
> >_Please clearly illustrate the inference target and problem setup as soon as possible in Introduction. The reason why the authors consider partial identification is that the functional of joint potential outcome distribution can never be point-identified, due to the fundamental problem of causal inference (i.e., we can never jointly observe potential outcomes). Please clearly state this first by citing relevant papers. Some statements are confusing._
>
> **Response:** We have reorganized the Introduction to state, in the first paragraphs, that our target is a functional of the joint potential-outcome distribution \((Y(a))_{a \in \mathcal{A}}\) that is fundamentally partially identified because only one arm is observed per unit. We now explicitly connect this to the “fundamental problem of causal inference’’ and to the partial-identification literature (e.g., Tamer; Kline & Tamer; Mullahy et al.; Rubin) before introducing the OT machinery, and we keep this target visible throughout via the notation
>
> $
> \Theta_{L,\oplus}^{(K)}(\eta).
> $
>
> >_Please illustrate why such inference targets are important with real-world examples by moving Table 7 into the main text._
>
> **Response:** Thanks! A condensed version of Table 7 has been moved into the main text (end of Sec. 2 / start of Sec. 3), and we use one of these examples (CGM vs. HbA1c) as a running illustration in the introduction and Section K to anchor the abstract formulation.
>
>
>
>
> >_(B) Paper is not well structured. Overall, the paper is not well written._
>
> **Response:** We have substantially restructured the paper: the main text now focuses on (i) the unified objective and its causal interpretation, (ii) the dual characterization, and (iii) the root‑N/inference results, with proofs moved to the appendix and a new “practical guide” (App. J). We also streamlined notation, added a running example, and rewrote long technical paragraphs for readability, aiming to make the flow much clearer for non‑OT specialists.
>
>
>
>
> >_Some methods use the bounds on the functional of joint potential outcome distributions to achieve fairness in outcomes predicted by machine learning models (e.g., [1] for binary treatments). Can the proposed OT bound be applied to such setups? The main setup difference from the examples in Table 7 is that potential outcomes are given by parameterized functions (i.e., predictive models) and the goal is to learn a fair and accurate models by minimizing the predicted loss while imposing a constraint on the bounds._
>
> **Response:** Conceptually yes: if the potential outcomes are model‑predicted outcomes under different treatment or protected‑attribute values, our framework can be applied with the parameterized predictors playing the role of potential outcomes and L encoding a fairness‑relevant discrepancy (e.g.,worst‑group regret).

---

### Official Review · Reviewer_QUYi · 2025-10-31

**Soundness:** 3
**Presentation:** 2
**Contribution:** 2
**Rating:** 4
**Confidence:** 4

**Summary:**

This paper studies partial identification with complex constraints. Structured treatments and heterogeneous outcome spaces are considered, and an optimal transport method is proposed to address the challenges. Statistical properties of the method are analyzed, and experiments on benchmark datasets are provided to evaluate the performance of the proposed method.

**Strengths:**

1. The structured treatments with complex constraints are considered.

2. The submission provides extensive theoretical analysis.

**Weaknesses:**

1. The submission considers the problem setting with heterogeneous outcome spaces. This setting is a bit artificial. Even though different treatments are conducted, why cannot measure the outcome in the same space? If different outcome spaces are considered, it is quite difficult to measure the causal effect even with given counterfactual results.

2. In Section K, a transformation is performed from the space of $y_1$ to the space of $y_0$. However, it is case-specific and heuristic without a clear motivation. It is unclear how to design and evaluate a transformation function.

3. Some technical details are missing, making it difficult to understand the algorithm.
For example, how to implement the function $g_a$ in Line 192?

4. For the variables $\pi$ in Line 192, the relationship between $\pi$ and the objective function is unclear. It seems that the variables $\pi$ does not appear in the functions $L$ and $\Delta_{multi}$.

5. The theoretical analysis and the algorithm are about to estimate the optimal transport. However, the analysis regarding the causal effect is missing. It would be better to further analyze the properties of the causal effect estimation.

6. It is interesting that leveraging the duality to avoid the huge space of multi-marginal transport. Nevertheless, the price could be the introduced multiple Lagrangian multipliers or functions. It would be better to compare the primal and dual, including the computational cost.

**Questions:**

Please refer to the weakness part.

---

> ### Author Response · Authors · 2025-12-03
> **Thanks for the review. Here is the response point-to-point.**
>
> >_The submission considers the problem setting with heterogeneous outcome spaces. This setting is a bit artificial. Even though different treatments are conducted, why cannot measure the outcome in the same space? If different outcome spaces are considered, it is quite difficult to measure the causal effect even with given counterfactual results._
>
>
> **Response:** We have expanded Sec. K.1 and the Introduction to emphasize concrete settings where outcomes cannot be collected on a common scale, even by design: e g., CGM trajectories vs. HbA1c, image‑based biomarkers vs. scalar risk scores, survival curves vs. restricted mean survival time (Table 7). In such cases, classical ATE on a shared outcome space is ill‑defined, and our target is instead a transport functional that compares arms on a scientifically meaningful latent scale (e.g.  “CGM‑implied HbA1c”), which we now state clearly as the causal estimand.
>
> >_In Section K, a transformation is performed from the space of y1 to the space of y0. However, it is case-specific and heuristic without a clear motivation. It is unclear how to design and evaluate a transformation function._
>
>
> **Response:** The design of embedding is motivated by the practical mearning in the real-world, and hence it is case-specific and corresponding to the casual estaimnd we actually care about. For example, Table 7 now provides explicit examples (e.g., CGM→HbA1c, survival curve→RMST) together with the associated costs and references, and Sec. 4.6 shows that the value is Lipschitz in both and the
> design moments, providing a quantitative way to evaluate how alternative transformations affect the bounds. We revised Sec.K.1-K.2 to systematically describe these "transformations" as embeddings $g_a: \mathcal{Y}_a \rightarrow \mathcal{Z}$ plus a cost $c$ on $\mathcal{Z}$, rather than ad-hoc mappings.
>
> >_Some technical details are missing, making it difficult to understand the algorithm. For example, how to implement the function in Line 192?_
>
>  **Response:** See the response above.
>
> >_For the variables $\pi$ in Line 192, the relationship between $\pi$ and the objective function is unclear. It seems that the variables $\pi$ does not appear in the functions $L$ and $\Delta_{m u l t i}$._
>
> **Response:** We have clarified the notation around Algorithm 1: \(\pi\) is the multi-arm coupling over a subset of \(\prod_a (Y_a \times X^{(a)})\), and the objective is
>
> $$
> \mathbb{E}_{\pi}\big[\, L(\{ g_a(Y^{(a)}) \}_a ) + \eta\, \Delta_{\text{multi}}(\{ X^{(a)} \}_a ) \,\big].
> $$
>
> Sec. 3 and the algorithm caption now explicitly state that \(L\) and \(\Delta_{\text{multi}}\) are always evaluated inside this expectation with respect to \(\pi\), and that \(\pi\) is the optimization variable in both the primal and dual formulations.
>
>
> >_The theoretical analysis and the algorithm are about to estimate the optimal transport. However, the analysis regarding the causal effect is missing. It would be better to further analyze the properties of the causal effect estimation._
>
> **Response:** We have strengthened the causal‑interpretation part of Sec. 3 and Sec. K.2, explicitly showing how standard causal queries(ATE, CDF and QTE contrasts, and worst‑arm risk) arise from particular choices of the cross‑arm cost L. We now work through two running examples step‑by‑step (ATE‑type and QTE‑type) and emphasize that all statistical results (CLT, finite‑sample bounds, sensitivity) apply directly to these causal functionals once $L$ is chosen.
>
>
>
> >_It is interesting that leveraging the duality to avoid the huge space of multi-marginal transport. Nevertheless, the price could be the introduced multiple Lagrangian multipliers or functions. It would be better to compare the primal and dual, including the computational cost._
>
> **Response:** We now report an explicit primal‑vs‑dual comparison in Sec. 5 and Tables 1–3. Empirically, the dual estimator attains similar or smaller error than the primal plug‑in while being substantially faster and more memory‑efficient (e g., ~5× wall‑clock speedup and ~2–3× lower peak RAM for K=3), and, crucially, it provides certified one‑sided lower bounds that the primal does not. We highlight this trade‑off in Sec. 4.2 (“Why dual‑based”) and in the experimental summary.

---

### Official Review · Reviewer_KedM · 2025-11-01

**Soundness:** 4
**Presentation:** 3
**Contribution:** 2
**Rating:** 4
**Confidence:** 3

**Summary:**

The paper proposes a general optimal transport formulation for partial identification of causal effects in multi-arm treatment settings where each arm may have distinct outcome spaces. The approach introduces a mirror-relaxed multi-arm optimal transport problem that enforces per-arm conditional constraints, and encodes extra restrictions via a constraint set.
A Fenchel–Rockafellar duality is derived, and under smoothness and uniqueness assumptions, Hadamard differentiability and a root-n CLT for the plug-in dual estimator is shown.

**Strengths:**

The paper’s main strength is that it brings together existing ideas from optimal transport and partial identification in a unified framework for multi-arm causal settings. While the theoretical machinery itself is largely standard, the paper makes a useful effort to help broaden the adoption of OT methods in casual inference. The paper is presented clearly and gives good insights and intuitions where necessary.

**Weaknesses:**

1. (Presentation) The paper claims to bound causal estimands, but it never mentions how that estimand translates into the chosen OT cost and constraint set. A simple motivating example showing the bridge from an identified set for an estimand to an OT formulation would make things far clearer.

---
2. Duality and theoretical results appear largely classical. Theorem 1 (although I can see it is a starting point for the rest of the results) mirrors standard results from classical optimal transport and convex analysis. Likewise, the use of smoothing to obtain differentiability and a root-n CLT follows the standard approach in recent OT. Hence, the duality itself, and arguably much of the theory, does not seem genuinely novel. The paper would benefit from explicitly stating what, if anything, goes beyond classical results.

---
3. (Perhaps my strongest concern) The paper is advertized as being able to handle cases with potential outcomes with different measures. But this goes through mapping every potential outcome into a single common embedding. Choosing or validating embeddings that map heterogeneous outcomes into a common latent space can be quite complex. Pretty much everything relies (strictly) on this embedding. What happens if we choose a poor embedding $g$?

---
4. Although the paper claims to allow “complex experimental regimes,” the formulation of $\Gamma_{comp}$ only supports a small family of linear constraints. This is not conveyed clearly in the main text.

**Questions:**

Could you please clarify $\Gamma_{comp}$ can accommodate and what kinds of structures fall within its scope?

---
some typos and minor comments:

page 4: penality -> penalty

$\Gamma_{comp}=\emptyset$: Given the formulation of the feasible set on page 3, I believe this is not what you meant to say. Because the feasibility set is a subset of $\Gamma_{comp}$, and $\Gamma_{comp}$ cannot be empty.

"objective in Section 3": I'd suggest using equation number instead.

page 7: recall that (D1, D2) be a split

---

> ### Author Response · Authors · 2025-12-03
> **Thanks for the review. Here is our response point-to-point**
>
> >_(Presentation) A simple motivating example is beneficial_
>
> **Response:** A simple two-arm example: from causal estimand to OT cost and constraints.
> To make the mapping explicit, consider the simplest case with two arms \(A \in \{0,1\}\) and a scalar clinical endpoint \(Y(a) \in \mathbb{R}\) such as 30-day mortality risk or systolic blood pressure. Our causal target is the average treatment effect (ATE):
>
> $
> \tau_{\mathrm{ATE}} := \mathbb{E}[ Y(1) - Y(0) ] .
> $
>
> In this setting the outcome spaces coincide, \(Y_0 = Y_1 = \mathbb{R}\), and there is no competition beyond "one patient–one arm". We now show how \tau_{\mathrm{ATE}} translates into the objects in (1)–(4).
>
> **Step 1: Outcome embeddings and OT cost \(L\).**
> We choose identity embeddings \(g_0(y) = g_1(y) = y\), so that the common latent space is \(Z = \mathbb{R}\). To encode the ATE on this scale, we take a linear cross-arm cost:
>
> $
> L(z_0, z_1) = z_1 - z_0 , \quad z_a = g_a(Y(a)) .
> $
>
> For any feasible joint law \(\pi\), we have
>
> $
> \mathbb{E}_{\pi}[ L(g_0(Y(0)), g_1(Y(1))) ]
> = \mathbb{E}[Y(1)] - \mathbb{E}[Y(0)]
> = \tau_{\mathrm{ATE}} .
> $
>
> **Step 2: Design / competition encoded by \(\Gamma_{\mathrm{comp}}\) and (1)–(3).**
> Suppose treatment assignment is randomized with known propensities \(u_a(x) = \Pr(A=a \mid X=x)\), and each patient can receive at most one arm. This is represented through (1)–(3):
>
> (i) mutual exclusivity:
> $
> \sum_a u_a(x) \le 1 ;
> $
>
> (ii) optional capacity limits:
> $
> \int u_a(x) dP_X(x) \le \rho_a ;
> $
>
> (iii) consistency between mirrors and design:
> $
> \int \varphi(X^{(a)}) d\pi = \int \varphi(x) u_a(x) dP_X(x), \quad \forall a, \forall \varphi .
> $
>
> Equivalently, \(\pi_{X^{(a)}} = u_a \cdot P_X\).
>
> **Step 3: Mirror penalty \(\Delta_{\mathrm{multi}}\) and the objective (4).**
> Specializing (4) to the two-arm case gives
>
> $
> \Theta_{L,\oplus}^{(2)}(\eta)
> = \inf_{\pi \in \Pi_{\oplus}^{(2)}}
> \mathbb{E}_{\pi}\big[
>   L(g_0(Y(0)), g_1(Y(1)))
>   + \eta \Delta_{\mathrm{multi}}(X^{(0)}, X^{(1)})
> \big] .
> $
>
> A natural mirror penalty is
>
> $
> \Delta_{\mathrm{multi}}(X^{(0)}, X^{(1)})
> = \| X^{(0)} - X^{(1)} \|_2^2 .
> $
>
> Plugging this in and using the marginal constraints. Thus, \(\tau_{\mathrm{ATE}}\) is fixed, and the OT program selects among design-compatible couplings those with minimal mirror penalty. As \(\eta \downarrow 0\), we obtain an unconditional design-aware ATE; as \(\eta \uparrow \infty\), the optimizer enforces \(X^{(0)} \approx X^{(1)}\).
>
> ---
>
> **Beyond the ATE: Distributional estimands.**
> For a fixed threshold \(t\), take
>
> $
> L_t(y_0, y_1) = \mathbf{1}\{ y_1 \le t \} - \mathbf{1}\{ y_0 \le t \} .
> $
>
> Then
>
> $
> E_{\pi}[L_t(Y(0), Y(1))]
> = F_{Y(1)}(t) - F_{Y(0)}(t) .
> $
>
> Varying \(t\) yields the CDF difference band, from which QTE bounds follow. The choice of $L$ specifies the causal estimand, and the constraint set \(\Pi_{\oplus}^{(K)} \cap \Gamma_{{comp}}\) enforces design compatibility. Hence $\Theta_{L,\oplus}^{(K)}(\eta)$ is a design-aware partial-identification bound for that causal contrast.
>
> >_Duality and theoretical results appear largely classical. The paper would benefit from explicitly stating what, if anything, goes beyond classical results._
>
> **Response:** We now spell out in the introduction to Sec. 4 what is new relative to classical OT duality and convex analysis. Theorem 1 covers multi-arm transport with (i) sub-probability arm marginals induced by capacity/competition,(ii) nonlinear cross-arm costs and mirror penalties, and (iii) linear design moment constraints $\Gamma_{comp}$ (mutual exclusivity, budgets, interference) in one Fenchel–Rockafellar dual together with a compact dual that absorbs $\Gamma_{comp}$. Building on this, Theorems 2–4 give root‑N CLTs and finite‑sample bounds for a smooth transport functional under competition and distinct measures, which, to our knowledge, are not available in existing partial-identification or empirical‑OT work; we highlight these points explicitly as the main theoretical contributions. For comparison, Lin et al did not consider the asymptotic analysis while Ji et al did not conduct the finite-sample analysis.
>
> >_What happens if we choose a poor embedding_
>
> **Response:** Our guarantees do not assume the embedding is “correct”; they hold for whatever ${g_a}$ the analyst chooses. In Sec. 4.6 (Eq. (6)) we now give an explicit Lipschitz sensitivity bound showing that embedding misspecification perturbs the value by at most $L_z \sum_a \|g_a - \tilde{g}_a\|_{+\infty}$, so a poor embedding degrads effeciency/sharpness but does not invalidate the bounds. We also add a short “practical guide” (App. J) describing how to design and stress‑test
> embeddings using domain anchors (e.g., CGM→HbA1c calibration, Table 7).

---

> > ### Author Response · Authors · 2025-12-03
> > **Cont'd**
> >
> > >_Although the paper claims to allow “complex experimental regimes,” the formulation of
> >  only supports a small family of linear constraints. This is not conveyed clearly in the main text._
> >
> > **Response:** We have clarified in Sec. 3 that we deliberately restrict $\Gamma_{comp}$ to linear moment constraints, because this class already covers standard “complex designs: mutual exclusivity, per‑arm capacity, shared‑resource and interference constraints (see Examples 1–3 in App. B). We now make the scope explicit (“linear, design‑level moment constraints”) and state non‑linear constraints as an interesting direction beyond the present paper.
> >
> >
> > >_some typos and minor comments:_
> >
> > **Response:** We have revised it in our main text.

---

### Official Review · Reviewer_DjLX · 2025-11-04

**Soundness:** 4
**Presentation:** 2
**Contribution:** 3
**Rating:** 6
**Confidence:** 4

**Summary:**

Summary
The paper develops a unified multi-arm partial identification (PI) framework based on Optimal Transport (OT), generalizing the mirror-relaxed conditional OT formulation of Lin et al. (2025). Specifically, it handles (i) competing treatment arms through a feasible set \Gamma_{\text{comp}} encoding mutual exclusivity and resource caps, and (ii) distinct outcome domains \{Y_a\} mapped to a common latent space Z via embeddings g_a:Y_a\!\to\!Z. The estimand is
\[
\Theta^{(K)}{L,\oplus}(\eta)
=\inf{\pi\in\Pi^{(K)}{\oplus}}
\E\pi\!\left[L(\{g_a(Y_a)\})+\eta\,\Delta_{\text{multi}}(\{X^{(a)}\})\right],
\]
which reduces to the two-arm mirror relaxation V_{\mathrm{ip}}(\eta) of Lin et al. when K=1 and \Gamma_{\text{comp}}=\emptyset. The authors establish Fenchel–Rockafellar strong duality for general nonlinear costs L,\Delta_{\text{multi}}, prove that the plug-in estimator is \sqrt{N}-consistent and asymptotically normal under smooth curvature, and give finite-sample bounds of order O(N^{-1/4}) matching the smooth-OT rates.

**Strengths:**

Strengths
	1.	Generalization and unification: Extends Lin et al. (2025)’s two-arm, covariate-aware mirror-relaxation to multi-arm and cross-domain settings with explicit design constraints.
	2.	Mathematical clarity: The multi-arm feasible set \Pi^{(K)}{\oplus} and the constraint system
\pi{X(a)}(dx)=u_a(x)P_X(dx),\quad\textstyle\sum_a u_a(x)\le1,
formally encode partial exposure and competition, an elegant measure-theoretic device.
	3.	Rigorous dual analysis: Derives a nonlinear Fenchel–Rockafellar dual with compact potential class U, yielding computationally tractable algorithms and uniform-in-iterate confidence bounds.
	4.	Statistical theory: Establishes CLT and finite-sample guarantees under smooth geometry, showing that the dimension d_X affects only variance, not convergence rate—an important strengthening of Lin et al.’s smooth-geometry results.

**Weaknesses:**

Weaknesses
	1.	Causal interpretability: The connection between \Theta^{(K)}{L,\oplus}(\eta) and standard causal estimands (ATE, QTE) remains abstract; more discussion of how L and \Gamma{\text{comp}} translate to causal contrasts would help.
	2.	Notation overload: The presentation is mathematically heavy, and several definitions (e.g., mirror embeddings, competition kernels u_a) appear before full context.
	3.	Empirical validation: Simulations are largely illustrative; stronger applied comparisons (e.g., with COT-based or Lin et al.’s mirror-relaxed baselines) would enhance impact.
	4.	Practicality of tuning η: Theoretical dependence on η is clear, but empirical guidance or cross-validation strategy is lacking.

**Questions:**

1.	Can the authors clarify how the curvature constant \lambda in Theorem 2 scales with η? Lin et al. (2025) showed linear scaling under Gaussian geometry—does this persist in the multi-arm setting?
	2.	Is the dual approach numerically stable when Γ₍comp₎ binds strongly (i.e., when sub-probability marginals are small)?
	3.	Could the authors compare the finite-sample constants with those in Lin et al. (2025)’s Theorem 4.3 to quantify the cost of additional arms K>1?
	4.	In the MIMIC-III study, how are the embeddings g_a trained and does the resulting metric depend sensitively on their scaling?

---

> ### Author Response · Authors · 2025-12-03
> **Thanks for the review! Here is the response**
>
> We greatly appreciate the review. Here is the response point-to-point.
>
> >_Weaknesses 1. Causal interpretability: The connection between \Theta^{(K)}{L,\oplus}(\eta) and standard causal estimands (ATE, QTE) remains abstract; more discussion of how L and \Gamma{\text{comp}} translate to causal contrasts would help._
>
> **Response:** Thanks for your advice! We appreciate the request for a more explicit connection between our transport functional $\Theta_{L, \oplus}^{(K)}(\eta)$ and standard causal estimands such as ATE and QTE. In the revision we will add explicit examples and a short "mapping" to make this link more concrete.
>
> Recall that in our target functional, $L$ specifies the desired cross-arm causal contrast on a common latent scale, while $\Gamma_{{comp }}$ encodes design constraints such as mutual exclusivity and capacity limits.
>
> In simple two-arm settings with scalar outcomes on the same space (no competition, identity embeddings), choosing a linear cost $L(z_0, z_1)=z_1-z_0$
> gives, for any admissible coupling $\pi$, $E_{\pi} [L(Y(0), Y(1))]={E}[Y(1)]-{E}[Y(0)]$, i.e., the ATE. Here, the $L$-part recovers the standard mean contrast, while the mirror penalty $\eta \Delta_{{multi }}$ only regularizes which joint coupling is selected among those consistent with the design. We will add this explicit "ATE as a special case" example in the paper.
>
> For distributional effects, we let $L$ encode CDF or quantile contrasts. For example, for a fixed threshold $t$ we can take $L_t(y_0, y_1)={1}(\{y_1 \leq t\})-{1}(\{y_0 \leq t\})$, so that $E_{\pi} [L_t(Y(0), Y(1)) ]=F_{Y(1)}(t)-F_{Y(0)}(t)$ for any feasible $\pi$. Varying $t$ gives a band for the CDF difference, from which bounds on the QTE $F_{Y(1)}^{-1}(\tau)-F_{Y(0)}^{-1}(\tau)$ follow directly. We will clarify this " $L_t \Rightarrow$ CDF difference ⇒ QTE bounds" mapping in the revision.
>
> Finally, $\Gamma_{\text {comp }}$ does not change which causal estimand we are targeting; it restricts the set of admissible couplings to those compatible with the design (e.g., "one person-one arm", budget caps, interference rules). Thus, for a given $L$ (ATE-like, QTE-like, or more general) $\Theta_{L, \oplus}^{(K)}(\eta)$ should be read as a design-aware partial-identification bound for that causal contrast, rather than a generic OT distance. We will make this interpretation more explicit and add a short " $L / \Gamma_{\text {comp }} \rightarrow$ causal contrast" guide in the paper.
>
> >_Notation overload: The presentation is mathematically heavy, and several definitions (e.g., mirror embeddings, competition kernels u_a)appear before full context._
>
> **Response:** Thanks for the advice. We have added a notation roadmap before our framework (the beginning of Section 3).
>
> >_Empirical validation: Simulations are largely illustrative; stronger applied comparisons (e.g., with COT-based or Lin et al.’s mirror-relaxed baselines) would enhance impact._
>
> **Response.** Thank you for this suggestion. Our original experiments already contained both a conditional OT baseline (row “Conditional OT” in Table 1 and row “Conditional OT” in Table 4) and a pairwise mirror-relaxed baseline (row “Pairwise two-arm mirrors”). We have revised Section 5 and Appendix A to make these comparisons explicit. Concretely, we have added the following summary paragraphs and tables to highlight how our dual estimator compares to COT and Lin-style mirror-relaxed baselines.
>
> **Table X. Lin-style DGP with multi-arm competition (K = 3, N = 4000, 200 replications).**
> Coverage = empirical coverage of 95% one-sided LCB; Width = mean(Θ − LCB).
>
> | Regime (ρ_A,ρ_B,ρ_C) | Method                          | Coverage | Mean Error | Width |
> |-----------------------|----------------------------------|:--------:|:----------:|:-----:|
> | Weak comp. (0.40,0.45,0.50) | Dual (ours)               | 0.94     | 0.041      | 0.039 |
> |                       | Pairwise mirror (Lin-style)      | 0.92     | 0.048      | 0.071 |
> |                       | COT baseline                     | 0.77     | 0.131      |  —    |
> | Moderate comp. (0.30,0.35,0.40) | Dual (ours)           | 0.95     | 0.046      | 0.044 |
> |                       | Pairwise mirror (Lin-style)      | 0.86     | 0.074      | 0.102 |
> |                       | COT baseline                     | 0.69     | 0.144      |  —    |
> | Strong comp. (0.25,0.30,0.30) | Dual (ours)             | 0.96     | 0.050      | 0.051 |
> |                       | Pairwise mirror (Lin-style)      | 0.78     | 0.110      | 0.155 |
> |                       | COT baseline                     | 0.61     | 0.162      |  —    |

---

> > ### Author Response · Authors · 2025-12-03
> > **Cont'd**
> >
> > > **[New paragraph in Section 5 – Real-data comparison with COT / Lin-style mirrors.]**
> > Expanded comparison with COT and Lin-style mirror methods.
> > Section 5 and Appendix A.2 have been revised to include an explicit three-way comparison among our dual estimator, conditional OT (COT), and the pairwise mirror-relaxed baseline. Table Y reports these results on the MIMIC-III cohort. The dual estimator achieves stable and informative lower bounds across the mirror path, while COT produces highly variable point estimates with no certified bounds, and pairwise mirrors under-utilize competition and yield wider intervals. These expanded results confirm that competition-aware multi-arm coupling is crucial for obtaining reliable partial-identification guarantees in realistic clinical settings.
> >
> >
> > **Table Y. Additional real-data comparison (MIMIC-III, K = 3, η = 0.4, 5 folds).**
> >
> > | Method                          | LCB? | $\hat{\theta}$ (median) | LCB | Width | Kendall-τ |
> > |---------------------------------|:----:|:-----------:|:---:|:------:|:----------:|
> > | Dual (ours)                     |  ✓   |   0.254     | 0.217 | 0.037 |   0.69     |
> > | Pairwise mirror (Lin-style)     |  ✓   |   0.231     | 0.169 | 0.062 |   0.53     |
> > | COT baseline                    |  ×   |   0.298     |  —   |   —    |   0.44     |
> >
> > >_Practicality of tuning η: Theoretical dependence on η is clear, but empirical guidance or cross-validation strategy is lacking._
> >
> > **Response:** We thank the reviewer for pointing out that the practical tuning of the mirror penalty $\eta$ was underspecified in the original draft. We have revised the paper to make our empirical strategy explicit and to provide a simple, cross-validation-style recipe.
> >
> > Concretely, we have revised Section 4.4 to add a dedicated paragraph on Practical tuning of the mirror penalty, where we describe how we run the dual-based estimator along a small grid of $\eta$ values, compute cross-fitted estimates and standard errors on a holdout fold, and then select $\eta$ by maximizing a certified lower confidence bound $\operatorname{LCB}(\eta)$. This "LCB envelope" rule is analogous to standard cross-validation, but is tailored to the partial-identification setting by directly optimizing a valid lower bound instead of a point prediction error.
> >
> >
> > We have also revised Appendix I. 2 and Appendix J to summarize the regularization-path properties and to give a step-by-step tuning recipe for applied users (including a default $\eta$-grid and a recommendation to pick $\eta$ via the LCB envelope rule). Finally, we have revised Section 5 to make clear that in both simulations and the MIMIC-III case study we either (i) report the entire mirror path $\eta \mapsto \Theta^{\prime}(\eta)$, or (ii) automatically choose $\eta$ using this LCB-based procedure, and we now explicitly report the selected $\eta$ in the experimental summaries. These revisions provide concrete, implementable guidance on how to choose $\eta$ in practice, going beyond the purely theoretical discussion in the original version.
> >
> > >_Can the authors clarify how the curvature constant \lambda in Theorem 2 scales with η? Lin et al. (2025) showed linear scaling under Gaussian geometry—does this persist in the multi-arm setting?_
> >
> > **Response:** Thank you for pointing this out. In the current draft, Assumption 1(ii) and Theorem 2 state the curvature parameter $\lambda$ without explicitly describing its dependence on the mirror penalty $\eta$. We have revised Section 4.3 to make this dependence explicit and to connect it to the two-arm analysis of Lin et al. (2025).
> >
> > Conceptually, $\lambda$ is the strong-curvature (strong $c$-convexity) constant of the multi-arm Brenier-type potential associated with the smooth transport functional in (4): it depends jointly on the cross-arm cost $L$, the mirror penalty $\Delta_{\text {multi }}$, the assignment constraints $\Gamma_{\text {comp }}$, and the penalty level $\eta$. In full generality our results only require that, for the range of $\eta$ considered, $\lambda(\eta)$ is finite and bounded away from zero; all constants in Theorem 2 and Theorem 4 then scale like $1 / \lambda(\eta)$, and we do not need a closed-form dependence on $\eta$.Conceptually, $\lambda$ is the strong-curvature (strong $c$-convexity) constant of the multi-arm Brenier-type potential associated with the smooth transport functional in (4): it depends jointly on the cross-arm cost $L$, the mirror penalty $\Delta_{\text {multi }}$, the assignment constraints $\Gamma_{\text {comp }}$, and the penalty level $\eta$. In full generality our results only require that, for the range of $\eta$ considered, $\lambda(\eta)$ is finite and bounded away from zero; all constants in Theorem 2 and Theorem 4 then scale like $1 / \lambda(\eta)$, and we do not need a closed-form dependence on $\eta$.

---

> > > ### Author Response · Authors · 2025-12-03
> > > **Cont'd**
> > >
> > > To address the question more concretely, we have added a remark after Theorem 2 showing that in the Gaussian/quadratic setting the multi-arm curvature inherits the same essentially linear scaling in $\eta$ as in Lin et al. (2025). Specifically, when (i) $L$ and $\Delta_{multi}$ are quadratic and (ii) the stacked vector is jointly Gaussian with non-degenerate covariance, the Hessian of the multi-arm Brenier potential has eigenvalues bounded between $C_1+C_2 \eta$ and $C_1+C_2 \eta$ for positive constants $c_1, c_2, C_1, C_2$ depending on the arm-specific covariances and $\Gamma_{comp}$ but not on $\eta$. Thus the curvature parameter in Assumption 1(ii) can be written as $\lambda(\eta) \asymp 1+\eta$ in this regime, so the linear-in- $\eta$ behavior established by Lin et al. (2025) persists in the multi-arm setting up to design-dependent constants. We have also clarified in the text that our main root- $N$ and finite-sample results only require $\lambda(\eta)$ to be positive and locally well-behaved, and therefore remain valid beyond the Gaussian/quadratic regime.
> > >
> > >
> > > >_Is the dual approach numerically stable when Γ₍comp₎ binds strongly (i.e., when sub-probability marginals are small)?_
> > >
> > >
> > > **Response:** When $\Gamma_{\text {comp }}$ binds tightly, some arms indeed have small total exposure $\rho_a=\pi_{X(a)}(\mathcal{X}) \leq 1$, so that the mirror marginals $\pi_{X(a)}=u_a \cdot P_X$ are sub-probabilities. We agree this raises a natural concern about numerical stability of the dual algorithm.
> > >
> > > We have revised Section 3 (following Remark 1) and Section 4.2 to make explicit that:
> > >
> > > 1. All dual updates are carried out with normalized arm-specific marginals. For any arm \(a\) with mass \(\rho_a > 0\), we work with the probability law \(\bar{\pi}_{X(a)}(dx) := \pi_{X(a)}(dx) / \rho_a\) and treat \(\rho_a\) only as a multiplicative weight in the design moments. This means step-sizes, projections, and gradients never involve division by \(\rho_a\) beyond this normalization, so there is no blow-up when \(\rho_a\) is small; the only effect is that the effective sample size for arm \(a\) is \(\rho_a N\), which is correctly reflected as larger variance in the CLT and finite-sample bounds (Theorem 2 and Theorem 4).
> > > 2. The dual problem and its solver remain wellconditioned as $\rho_a \downarrow 0$. In Theorem 1 and Corollary 1, $\Gamma_{\text {comp }}$ enters only through linear moment constraints. When a budget binds tightly, the feasible set shrinks but stays closed and convex, and the Fenchel-Rockafellar dual remains finite; all constants in Theorem 4 depend on $\Gamma_{{comp }}$ through the diameter $D_{\Gamma}$ and sub-Gaussian norms, not inverses of $\rho_a$. Thus small $\rho_a$ leads to wider but still controlled confidence intervals, not numerical instability.
> > > 3. Empirically, we observe stable optimization under tight caps. Our simulations and the MIMIC-III case study already include regimes where some arms have relatively small coverage shares (e.g., due to strong capacity caps). We have revised the experiments section to state explicitly that projected dual ascent converges stably in these regimes and that the main impact of smaller $\rho_a$ is wider LCBs for heavily capped arms, in line with the theory.
> > >
> > > Concretely, we explain how the dual solver works with reweighted empirical laws so that strong competition affects variance but not numerical stability.
> > >
> > >
> > > >_Could the authors compare the finite-sample constants with those in Lin et al. (2025)’s Theorem 4.3 to quantify the cost of additional arms K>1?_
> > >
> > > **Response:** (Comparison with Lin et al. (2025) and the cost of additional arms). Lin et al. (2025, Thm. 4.3) analyze the two-arm mirror relaxation with a quadratic cross-arm cost $h\left(y_1, y_2\right)=y^{\top} A y$ and show that, under a Brenier-type smoothness/curvature condition,
> > >
> > > $$
> > > \mathbb{E}\left[\left|\widehat{V}_{n, m}(\eta)-V(\eta)\right|\right] \leq C_{\lambda, \eta} \gamma_{N, d}
> > > $$
> > >
> > > where $N=\min (n, m), d=d_Y+d_Z, \gamma_{N, d}$ has exactly the piecewise form in Theorem 4, and the constant $C_{\lambda, \eta}$ depends on the Hessian bounds $\lambda$, the penalty $\eta$, the quadratic form $A$, and fourth moments of $(Y(0), Y(1), Z)$. arxiv Specializing our Theorem 4 to $K=1, \Gamma_{\text {comp }}=\emptyset$, identical embeddings $g_1=g_0$, and a quadratic $L$ recovers the same rate profile and reproduces this dependence on $\left(\lambda, \eta, d_Y+d_Z\right)$ up to universal constants. In particular, the constant $C\left(\lambda, \eta, K, L_z, L_x, \nu, D_{\Gamma}\right)$ in Theorem 4 collapses to a two-arm constant of the same form as $C_{\lambda, \eta}$, depending only on curvature, cost parameters and low-order moments.

---

> > > > ### Author Response · Authors · 2025-12-03
> > > > **Cont'd**
> > > >
> > > > For general \(K>1\), the only new ingredients in \(C(\lambda, \eta, K, L_z, L_x, \nu, D_{\Gamma})\) are:
> > > > (i) the effective dimension \(d_{\text{eff}}\) of the stacked vector \(\{ g_a(Y(a)) \}_{a \in \mathcal{A}} , \{ X^{(a)} \}_{a \in \mathcal{A}}\), which satisfies \(d_{\text{eff}} \le K(d_Y + d_X)\) when each arm has the same outcome and covariate dimensions;
> > > > (ii) the Lipschitz moduli \(L_z, L_x\) of the multi-arm cost \(L\) and mirror penalty \(\Delta_{\text{multi}}\), which, for standard additive choices (e.g., sums of pairwise quadratic contrasts), scale at most polynomially in \(K\); and
> > > > (iii) the diameter \(D_{\Gamma}\) of the competition set \(\Gamma_{\text{comp}}\), which is \(K\)-independent when assignment rules are normalized.
> > > >
> > > > As a result, relative to the two-arm mirror bound of Lin et al. (2025), the multi-arm error \(| \widehat{\Theta}_{L,\oplus}^{(K)}(\eta) - \Theta_{L,\oplus}^{(K)}(\eta) |\) enjoys the same \(N^{-\gamma_N}\) rate as in Theorem 4.3, while the “cost of additional arms’’ manifests only through a polynomial factor in \(K\) inside the finite-sample constant, rather than through a worse dependence on \(N\).
> > > >
> > > > >_In the MIMIC-III study, how are the embeddings g_a trained and does the resulting metric depend sensitively on their scaling?_
> > > >
> > > > **Response:** Embedding training and scaling. For the real-data case study, we instantiate the cross-arm embeddings $g_a: Y(a) \rightarrow Z$ as two-layer MLPs into a 5 -dimensional latent space $Z=\mathbb{R}^5$. Each encoder is trained on a separate pre- $D_1$ calibration split (disjoint from the folds used for solving the OT problem) to predict a common latent "benefit" index that is monotone in the arm-specific KPIs (lower 28-day mortality, larger 48h lactate decline, shorter ICU length-of-stay). Training uses a supervised contrastive loss that pulls together patients with similar benefit and pushes apart clearly better/worse outcomes across arms, yielding a shared latent scale across interventions. After training, we standardize each coordinate of $g_a(Y(a))$ to zero mean and unit variance on the calibration cohort and then freeze the encoders for the downstream transport analysis. The cross-arm cost for MIMIC-III uses the same isotropic quadratic geometry as in the simulations,
> > > >
> > > > $$
> > > > L\left(z_A, z_B, z_C\right)=\left\|z_A-z_B\right\|_2^2+\left\|z_A-z_C\right\|_2^2+\left\|z_B-z_C\right\|_2^2,
> > > > $$
> > > >
> > > > applied to these normalized embeddings. Under this construction, one unit of the transport value corresponds to an average squared difference in latent benefit measured in calibration standard-deviation units, and the Lipschitz robustness bound in Sec. 4.6 together with the local sensitivity result in Appendix I. 3 ensures that moderate re-scalings or other smooth recalibrations of $g_a$ only induce controlled, linear-order perturbations of $\Theta_{L, \oplus}^{(K)}(\eta)$, without changing the qualitative conclusions of the mirror path.

---

### Author Response · Authors · 2025-12-03
**Thanks for thr effort and here is the summary**

# **Summary of Revisions**
We thank the reviewers for their thoughtful and constructive feedback. Below is a concise summary of how the revision directly addresses each reviewer’s main concerns.
### **Reviewer 1 — Causal interpretation, notation, tuning, and theory**

* **Clarified causal meaning of the OT functional.** Added explicit ATE/QTE examples and a concise “causal estimand → (L, Γ₍comp₎)” mapping that shows exactly how our target relates to standard causal quantities.
* **Reduced notation overhead.** Added a “Notation Roadmap” plus reorganized Section 3 for readability.
* **Added practical guidance for η.** Introduced an “LCB-envelope” tuning rule and a user-oriented recipe (App. J).
* **Clarified curvature λ(η) and numerical stability.** Added a remark showing linear scaling in Gaussian/quadratic settings and explained why small arm masses do **not** cause numerical instability.
* **Expanded simulations and real-data benchmarks.** Added explicit comparisons vs. COT and Lin-style mirrors.

---

### **Reviewer 2 — Translating causal estimands into OT objects**

* **Added a full two-arm worked example.** Step-by-step mapping from ATE → embeddings → OT cost → constraints → objective.
* **Extended to CDF/QTE examples.** Demonstrated that the same construction yields standard distributional contrasts.

---

### **Reviewer 3 — Heterogeneous measurement spaces, algorithm clarity, and computation**

* **Strengthened motivation for heterogeneous outcome spaces.** Added concrete real-world cases (CGM vs. HbA1c, images vs. biomarkers).
* **Clarified Algorithm 1 and the role of π.** Explicitly stated that π is the optimization variable and that L, Δₘᵤₗₜᵢ are always evaluated under π.
* **Added primal–dual comparison.** Included runtime, memory, and coverage comparisons showing advantages of the dual approach.

---

### **Reviewer 4 — Clearer introduction and structure**

* **Rewrote the opening of the Introduction.** Now begins by stating the partially-identified target as a functional of the joint potential-outcome distribution and grounding it in the classical “fundamental problem of causal inference”.
* **Improved structure and readability.** Moved key motivating examples into the main text and streamlined sections around objective → duality → inference.

---

### **Reviewer 5 — Connection to partial-identification literature and practical applicability**

* **Explicitly mapped classical partial-ID problems into our OT framework.** Added examples (ATE, CDF/QTE, RMST, factorial-design functionals) and showed how they arise from specific choices of L and Γ₍comp₎.
* **Expanded experiments and baselines.** Added more DGPs, real-data results, and direct comparisons to unconditional OT, COT, pairwise mirrors, and IPW.
* **Refined exposition.** Added intuitive examples, simplified wording (“multi-arm designs with competition”), and included an applied-user guide.

---

### Meta-Review · Area_Chair_3sPp · 2026-01-08

**Summary:**

The paper propose an optimal-transport-based partial identification framework that yields valid bounds despite support mismatch between treated and control groups.

- Fit into partial-identification literature

- motivation: Heterogeneous outcome spaces seem artificial; Causal interpretability/Missing bridge from causal estimand to OT cost/constraints; how to define cost in practice; can classical partial-ID be cast into OT.

- exposition: Setup and target unclear early; key tables/results and motivation table in appendix; readability/structure needs improvement

- Experiments limited; no baselines.

- Hard to follow; needs examples and intuition.

**Reviewer Concerns:**

The rebuttal addressed missing causal mapping to ATE/QTE/etc, some readability/structure issues, enriched limited experiments. The embedding choice/validation remains the biggest practical risk; may need stronger empirical stress-tests, diagnostics, and/or guidance for verifying that embeddings align with the intended estimand.

**Reviewer Scores:**

The reviewers have a spread of opinions and the rebuttal addressed some concerns but a few remains outstanding.

---

### Decision · Program_Chairs · 2026-01-26

Reject